# Template and target-site recognition by human LINE-1 in retrotransposition

Akanksha Thawani[1,2✉], Alfredo Jose Florez Ariza[1,3], Eva Nogales[1,2,4,5✉] & Kathleen Collins[1,2✉]

The long interspersed element-1 (LINE-1, hereafter L1) retrotransposon has generated nearly one-third of the human genome and serves as an active source of genetic diversity and human disease[1]. L1 spreads through a mechanism termed target-primed reverse transcription, in which the encoded enzyme (ORF2p) nicks the target DNA to prime reverse transcription of its own or non-self RNAs[2]. Here we purified full-length L1 ORF2p and biochemically reconstituted robust target-primed reverse transcription with template RNA and target-site DNA. We report cryo-electron microscopy structures of the complete human L1 ORF2p bound to structured template RNAs and initiating cDNA synthesis. The template polyadenosine tract is recognized in a sequence-specific manner by five distinct domains. Among them, an RNA-binding domain bends the template backbone to allow engagement of an RNA hairpin stem with the L1 ORF2p C-terminal segment. Moreover, structure and biochemical reconstitutions demonstrate an unexpected target-site requirement: L1 ORF2p relies on upstream single-stranded DNA to position the adjacent duplex in the endonuclease active site for nicking of the longer DNA strand, with a single nick generating a staggered DNA break. Our research provides insights into the mechanism of ongoing transposition in the human genome and informs the engineering of retrotransposon proteins for gene therapy.

Non-long-terminal-repeat (non-LTR) retrotransposons are mobile genetic elements in the human genome that are recognized as drivers of genome expansion and evolution[1]. The human genome has one autonomously active retrotransposon from the LINE family. Human L1 is present in an estimated 80–100 transposition-competent copies[3] that are sources of genetic diversity and ongoing somatic mosaicism[4], and contribute to more than 100 known human disease cases[5,6]. Bicistronic L1 encodes an ORF1 protein that binds to RNA[7], and an enzymatic ORF2 protein that has endonuclease (EN) and reverse transcriptase (RT) activities[8,9] (Fig. 1a). New L1 insertions initiate by target-primed reverse transcription (TPRT), in which target-site nicking creates a primer for cDNA synthesis directly into the genome[2,8,10,11]. L1 ORF2p has generated more than 30% of the human genome through transposition and pseudogene synthesis[12]. Current efforts that seek to limit human disease by controlling L1 mobility[13], and to exploit non-LTR retrotransposons and other RTs for genome engineering[14–17], provide an increasingly compelling demand for mechanistic understanding of TPRT and stable cDNA incorporation into the genome. However, much remains unclear, in large part owing to experimental difficulties in L1 ORF2p biochemical reconstitution and structural analyses.

The purification of active L1 ORF2p has been challenging due to the scarcity of L1 ribonucleoproteins (RNPs) in cells, as well as the heterogeneous association of L1 ORF2p with L1 and other RNAs and many directly or indirectly interacting proteins[18–21]. Consequently, biochemical assays for L1 activity have been limited, most relying on the cellular assembly of

an L1 ORF2p RNP[22,23]. Among the questions that remain to be addressed, understanding how L1 ORF2p recognizes template RNAs to initiate TPRT is particularly critical (Fig. 1b). The prevailing model, termed *cis*-preference, proposes that L1 ORF2p co-translationally engages the polyadenosine (poly(A)) tail of its encoding transcript to promote selective binding and cDNA insertion of the L1 mRNA[24–26]. Yet, the most abundant insertions mediated by L1 ORF2p are the non-autonomous short interspersed nuclear elements (SINEs), such as Alu SINEs[24,27,28]. In another outstanding question, how the EN domain of L1 ORF2p selects target sites to nick for TPRT initiation, beyond the short consensus motif TTTTT/AA[8,29–32], remains poorly understood (Fig. 1b). Robust biochemical reconstitutions and structural studies with the purified L1 ORF2p are needed to understand the mechanisms of nucleic acid recognition for TPRT.

## Reconstitution of L1 ORF2p-mediated TPRT

We expressed the full-length L1 ORF2p in insect cells and purified it to relative homogeneity (Extended Data Fig. 1a,b). With an optimal target DNA structure (see below) containing a single TTTTT/AA consensus for genomic L1 insertions[8,29–32], efficient nicking occurred at the intended site, evident by the formation of a 16-nucleotide (nt) nicked product, and TPRT product was synthesized by nick-primed reverse transcription of template RNA (Fig. 1c). We compared template RNAs that are established native substrates of L1 ORF2p, including the L1

[1]California Institute for Quantitative Biosciences (QB3), Berkeley, CA, USA. [2]Department of Molecular and Cell Biology, University of California Berkeley, Berkeley, CA, USA. [3]Biophysics Graduate Group, University of California Berkeley, Berkeley, CA, USA. [4]Howard Hughes Medical Institute, Chevy Chase, MD, USA. [5]Molecular Biophysics and Integrated Bioimaging Division, Lawrence Berkeley National Laboratory, Berkeley, CA, USA. ✉e-mail: athawani@berkeley.edu; enogales@lbl.gov; kcollins@berkeley.edu

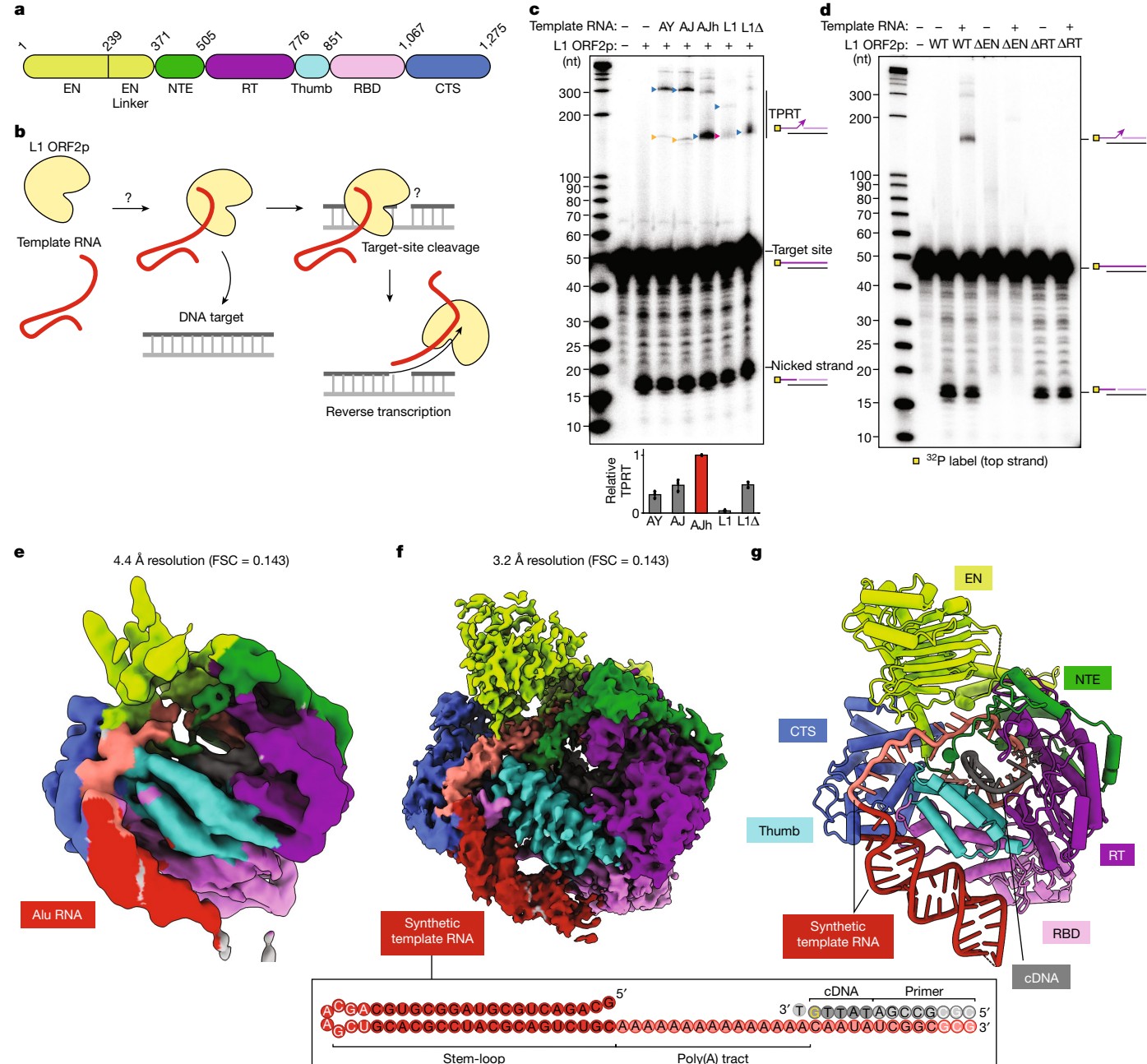

**Fig. 1 | In vitro TPRT activity and cryo-EM structures of human L1 ORF2p RNPs. a**, Domains of human L1 ORF2p. CTS, C-terminal segment; EN, endonuclease; NTE, N-terminal extension; RT, reverse transcriptase; RBD, RNA binding domain. **b**, Schematic of L1 ORF2p-mediated TPRT. **c**, Denaturing gel analysis of TPRT reaction products. The yellow square represents the [32]P-labelled 5′ end of the target DNA strand. The triangles indicate the expected TPRT product for full-length template (blue), incomplete cDNA synthesis (magenta) and possible internal initiation (mustard). Wild-type L1 ORF2p was assayed using different template RNAs with a 25A 3′ end: AY, AluY SINE (307 nt); AJ, AluJ SINE (306 nt); AJh, AluJ half-SINE (141 nt); L1, L1 3′ UTR (231 nt); L1Δ, L1 3′ UTR ΔG-quadruplex (149 nt). Here and for all of the subsequent gels, the DNA ladder length (in nucleotides) is indicated on the left. The experiment was replicated three times. The full-length cDNA product was quantified, normalized to the full-length cDNA product with AJh RNA. The mean ± s.d. of

$n = 3$ biologically independent replicates is displayed below. Here, and for all quantifications, the black dots depict individual data points. **d**, Denaturing gel analysis of TPRT reaction products with wild-type L1 ORF2p and EN-dead (ΔEN) and RT-dead (ΔRT) mutants. AJh 25A (141 nt) was used as the template. The experiment was replicated three times. **e**, Cryo-EM density of L1 ORF2p in a complex with AJh RNA–poly(T) primer in an elongation state, segmented and coloured by domains. FSC, Fourier shell correlation. **f**, Cryo-EM density of L1 ORF2p in an elongation state with synthetic template RNA and primer extended with cDNA, segmented and coloured by domains. A schematic of the synthetic template RNA and cDNA primer used to obtain the high-resolution cryo-EM structure is shown below, with the cDNA 3′ ddG in yellow and the incoming dTTP that is unable to join the cDNA is also shown; white-coloured nucleotides were not modelled in the structure. **g**, Ribbon diagram of the L1 ORF2p RNP structure derived from **f**, coloured by domain.

3′ UTR and Alu RNAs, each with a 3′ 25A tail (Supplementary Table 1). All Alu RNAs, including the evolutionarily youngest AluY RNA[28], a resurrected AluJ RNA[33,34] and a left-half monomer of the Alu RNA tandem repeat sufficient for genome insertion[33] (AluJ half (AJh)), were

efficiently reverse transcribed from the nicked primer (Fig. 1c (lanes AY, AJ and AJh)). By contrast, TPRT of the L1 3′ UTR RNA resulted in a lower amount of product synthesis, with products predominantly migrating faster than expected for full-length cDNA (Fig. 1c (lane L1)). An L1 3′

UTR template lacking nucleotides 1–78 that form a G-quadruplex[35] gave the expected cDNA length, matching the length of the shorter products from the full-length L1 3′ UTR template (Fig. 1c (lanes L1 and L1Δ)). Neither L1 3′ UTR template supported as much TPRT as Alu RNA (AJh), suggesting that the L1 3′ UTR is a suboptimal template for L1 ORF2. Using the optimal AJh template, we verified that neither a control retroviral RT from Moloney murine leukaemia virus (M-MLV RT) nor an EN-dead L1 ORF2p mutant had nicking or TPRT activities (Fig. 1d and Extended Data Fig. 1c), yet both showed robust RT activity as assayed by primer-extension on an annealed RNA–DNA duplex (Extended Data Fig. 1d,e). By contrast, an RT-dead L1 ORF2p retained target-site nicking but no RT or TPRT activity (Fig. 1d and Extended Data Fig. 1d,e). These controls validate our direct readout of robust L1 ORF2p-mediated TPRT activity, bypassing the PCR-based amplification required previously[10].

## Structure of template-RNA-bound L1 ORF2p

We sought to capture the structure of L1 ORF2p. Although our initial attempts at cryo-electron microscopy (cryo-EM) reconstruction of L1 ORF2p without nucleic acids were unsuccessful, we were able to capture L1 ORF2p engaged with RNA. We imaged L1 ORF2p bound to Alu AJh RNA with a poly(thymidine) (poly(T)) primer base-paired to its 3′ end to mimic the initiation of cDNA synthesis (Fig. 1e). In the resulting 4.4-Å-resolution density map, we could place the predicted AlphaFold model of human L1 ORF2p[36] and further identify extra density consistent with the Alu RNA stem-loop bound on one side of the protein and its 3′ tail in the L1 ORF2p RT core in an orientation that is topologically compatible with the co-binding of the Alu RNA partner, the SRP9/14 heterodimer[33] (Fig. 1e and Extended Data Fig. 2a,c). However, the Alu RNP map had preferred orientation issues and did not have the resolution to visualize amino acid side chains (Extended Data Fig. 2).

We improved the quality and resolution of our density map substantially when we used an L1 ORF2p complex with a synthetic RNA template mimicking Alu RNA features (Fig. 1f (right)), containing a 5′ stem-loop and a 3′ single-stranded region of sufficient length to span the distance between the Alu RNA stem-loop position and the active site of L1 ORF2p seen in our 4.4 Å RNP map. As cellular assays concur that L1 templates require a 3′ poly(A) tract[37,38], we used adenosine in the single-stranded region. We halted elongation after 5 bp of cDNA synthesis with dideoxyguanosine triphosphate (ddGTP) replacing dGTP (Fig. 1f (right)). Using this sample, we obtained the cryo-EM structure of the RNP in a paused elongation state at an overall resolution of 3.2 Å (Fig. 1f, Extended Data Figs. 3 and 4 and Extended Data Table 1). This resolution enabled us to model the entire protein chain and the individual nucleotides, including a dTTP bound as a nucleotide substrate but unable to join the cDNA 3′ end (Fig. 1g and Extended Data Fig. 5a,b). Only 8 nt of template RNA near the loop and 3 bp of RNA–DNA duplex farthest from the active site could not be modelled (Fig. 1f (bottom)).

The L1 ORF2p RT core consists of the palm and fingers (together, the RT domain) in the right-hand architecture shared by many polymerases, followed by the thumb domain and preceded by an N-terminal extension (NTE) domain that was previously noted in L1 ORF2p as the Z domain[39,40], all shared with prokaryotic and eukaryotic retrotransposon RTs[41] (Fig. 1a,f,g). The RT and thumb domains cradle the RNA–DNA duplex emerging from the active site. Preceding the NTE, L1 ORF2p has an N-terminal apurinic/apyrimidinic EN domain fold[42] that is connected to the rest of the protein through a folded domain incorporating the previously noted 'cryptic motif'[39] and hereafter designated EN linker, which packs against an adjacent portion of the NTE. The 209-amino-acid L1 C-terminal segment (CTS), together with the NTE and EN linker domains, create an extended surface of contacts with the poly(A) tract of the template RNA proximal to the active site (Fig. 1f,g; summarized in Fig. 2a). The region between the CTS and thumb, which we labelled as a previously unidentified RNA-binding domain (RBD; Fig. 1a), contacts both the RT-bound template RNA and peripheral RNA

stem-loop (Fig. 1f,g and Extended Data Fig. 5c; summarized in Fig. 2a). The array of protein–RNA interactions bends the template RNA to follow an L-shaped architecture (Fig. 1f,g). Overall, our structure reveals a previously undescribed topology and indicates biochemical roles for the different L1 ORF2p domains.

## Features of the catalytic core

L1 ORF2p RT activity is supported by numerous side-chain interactions with nucleic acids. Of the traceable 11 base pairings, 9 are almost fully enclosed, predominantly by interactions with the RT, thumb and RBD domains (Fig. 2a and Extended Data Fig. 5). The incoming dTTP and the dideoxy-guanosine at the primer 3′ end (ddG-13) are positioned by the canonical FADD active-site motif and by the conserved aromatic residues Phe566 and Phe605 (Extended Data Fig. 5a). The incoming dTTP hydrogen bonds with three RT domain residues, including the Arg531 side chain (Extended Data Fig. 5b). These contacts parallel the configuration of a group II intron RT active site[43–46]. The RNA strand of the heteroduplex exiting the active site contacts residues in the NTE and RT domains, and it also contacts the RBD domain not shared with group II intron RTs (Extended Data Fig. 5c). The cDNA strand has fewer contacts: an electrostatic interaction between the DNA backbone and the side chain of Arg375 in the NTE domain, and several hydrophobic contacts with sugars by thumb and RT domain residues (Extended Data Fig. 5d). All contacts to nucleic acids in the RT core are sequence non-specific.

## Single-stranded RNA recognition

Side chains across several domains in the protein define the surface for recognition of the 15 nt single-stranded poly(A) tract template (Fig. 2a,b). The EN linker, NTE, RT and thumb domains engage the poly(A) tract proximal to the active site, whereas the CTS domain interacts with the poly(A) tract predominantly adjacent to the stem-loop (Fig. 2a,b). This architecture suggests a 'threshold' model in which a substantial length of 3′ poly(A) would be required for template binding and threading into the active site. To define the poly(A) length for optimal TPRT, we designed and purified AJh RNAs with variable poly(A) tail length and 3′ tail sequences (Supplementary Table 1 and Extended Data Fig. 5e) and used them as templates for TPRT by L1 ORF2p. Templates with 75A, 50A, 25A and 20A were used efficiently, whereas shorter A-tails of 15A, 10A and 5A produced much less or no TPRT product (Fig. 2c). These results agree with our structure-based prediction: a template with 20A, allowing 5A for base-pairing with the nicked primer and at least 15 nt of single-stranded poly(A), can be efficiently used for TPRT initiation, while, for a template with 25A, product synthesis reaches the same level as obtained using templates with longer A tracts (Fig. 2c). Notably, AJh RNA with either 75A or 50A produced a heterogeneous size distribution of TPRT products, with 75A displaying a distinct skew towards a lower length of cDNA product than the expected 200 nt full-length cDNA (Fig. 2c). This heterogeneity suggests that the longer poly(A) tracts exceed the length of single-stranded RNA recognized by L1 ORF2p. Overall, our findings agree with studies showing that the poly(A) tail is required for in vivo mobility of L1[37] or Alu SINEs[38].

Notably, we observed side-chain interactions with A bases distributed across the entire length of poly(A) tract (Fig. 2a), including contacts that sequence-specifically recognize the adenine base (Fig. 2d–f). The A-60 base forms adenine-specific hydrogen bonds with Arg385 and Asn388 of the NTE domain, as well as a hydrophobic contact with Ile517 from the RT domain (Fig. 2d). The A-57 base forms a hydrogen-bond with Lys1236 from the CTS domain and stacks against the Trp365 side chain from the EN linker domain (Fig. 2e). The A-55 base forms hydrogen-bonds with Asn371 and Cys804 from the NTE and thumb domains, respectively, and is caged in a hydrophobic pocket formed by leucine residues from the NTE domain and Phe366 from the EN linker (Fig. 2f). The CTS

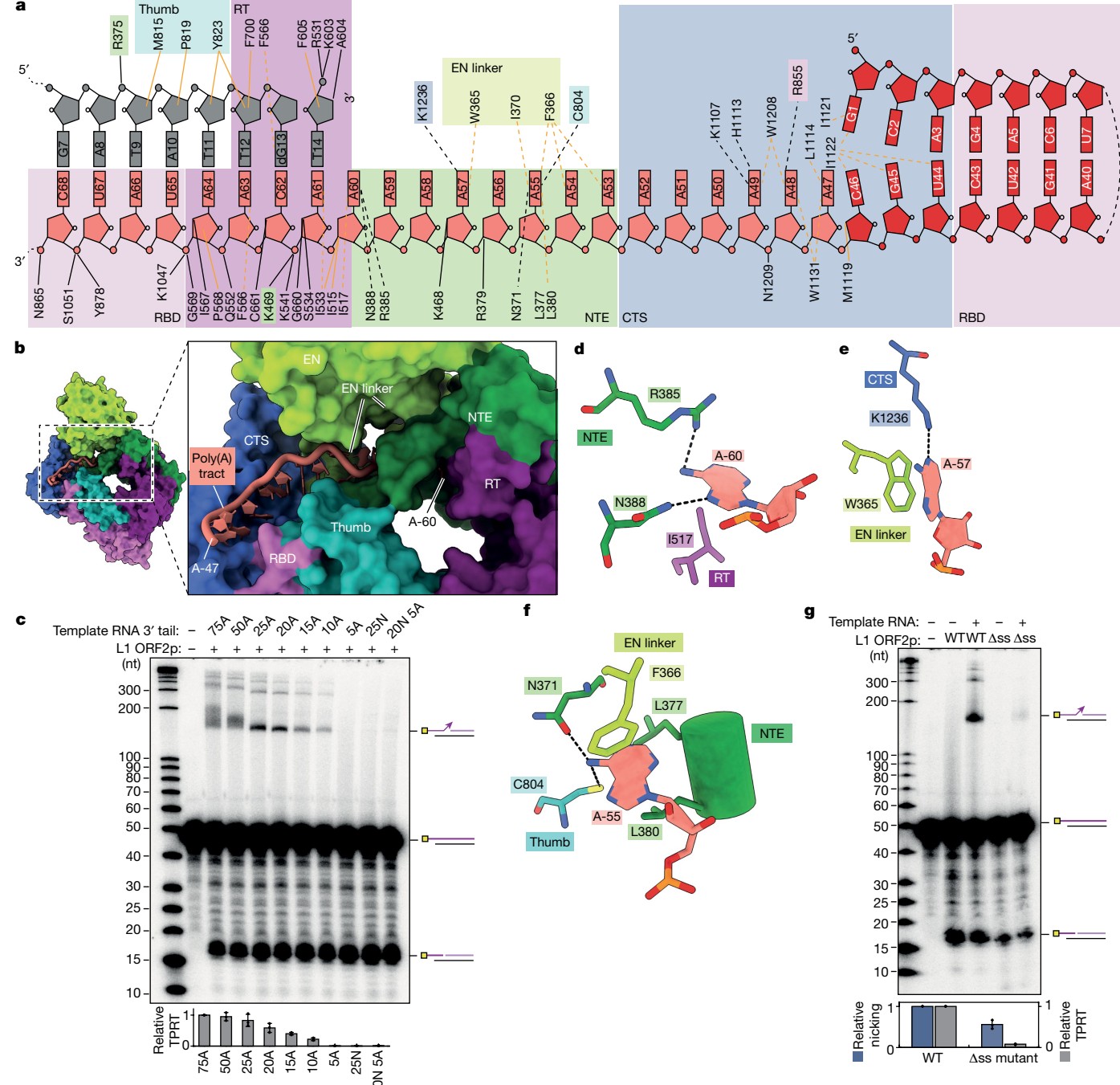

**Fig. 2 | Recognition of the template RNA and its poly(A) tract. a**, Schematic of direct interactions between L1 ORF2p and the template RNA. The black lines denote hydrogen bonds and the mustard lines denote hydrophobic contacts. The dashed lines represent direct contacts with the nucleobases or ribonucleobases. **b**, Recognition of the poly(A) tract by the EN linker, NTE, RT, thumb and CTS domains. **c**, Denaturing gel analysis of TPRT reaction products with AJh template RNAs with differing 3′ poly(A) tail lengths, including 75A (191 nt), 50A (166 nt), 25A (141 nt), 20A (136 nt), 15A (131 nt), 10A (126 nt) or 5A (121 nt), or with a 25N 3′ tail (141 nt) or 20N and 5A nucleotides (20N5A, 141 nt). The 25N sequence is GGTAACGAGAACTGTCATGCACCCC and the 20NA5 sequence is GGTAACGAGAACTGTCATGCAAAAA (Supplementary Table 1). The experiment was replicated three times. Full-length cDNA product was quantified, normalized to the full-length cDNA product with AJh 75A. The mean ± s.d. of $n = 3$ biologically independent replicates is shown below. **d**, Adenine-specific hydrogen bonds between template A-60 and side chains in the NTE and thumb domains, alongside a hydrophobic contact with the RT domain. **e,f**, Hydrogen bonds and hydrophobic interactions between the template A-57 base and side chains in the CTS (**e**) and EN linker domains, and between the A-55 base and EN linker, thumb and NTE domain residues (**f**). A heteroatom representation (red, oxygen; blue, nitrogen) is shown. **g**, Denaturing gel analysis of TPRT reaction products with wild-type or single-stranded RNA binding mutant (Δss) L1 ORF2p using AJh 25A template RNA. The experiment was replicated three times. The full-length cDNA was quantified as the TPRT product, and the nicked product at the expected size was quantified independently. Relative EN nicking and TPRT in the +RNA lanes were normalized to the wild-type L1 ORF2p. The mean ± s.d. of $n = 3$ biologically independent replicates is shown below.

domain also contributes to adenine-specific recognition (see below). To investigate the dependence of TPRT on the single-stranded poly(A) sequence, we generated AJh-based RNA templates terminating in a

25 nt sequence with mixed base composition (25N) or 20N with 3′ 5A to retain template-primer base pairing (Fig. 2, Supplementary Table 1 and Extended Data Fig. 5e). Neither template supported the TPRT activity

of L1 ORF2p (Fig. 2c). Further intrigued by the large number of hydrogen bonds with the poly(A) tract, we created mutant L1 ORF2p with alanine substitutions for all eight side chains that make base contacts to single-stranded RNA (Fig. 2a). When assayed for TPRT activity, the L1 ORF2p mutant for single-stranded RNA base interactions (Δss) showed distinctly reduced TPRT while retaining significant EN activity and RT activity when assayed by primer extension (Fig. 2g and Extended Data Fig. 1d,e). We suggest that these contacts contribute to a conformation of L1 ORF2p poised for cDNA synthesis.

## Novel roles for the C-terminal domain

The template RNA stem-loop and poly(A) region distal to the RT active site are predominantly engaged by the CTS domain (Fig. 2a,b). Adjacent to the stem-loop, the A-49 base makes adenine-specific hydrogen bonds with Lys1107 and His1113 in the CTS domain (Fig. 3a). Other CTS domain interactions with the RNA are predominantly hydrophobic, without much sequence specificity, in agreement with previous research[47] (Fig. 2a). This involves aromatic side chains of Trp1208, Trp1131 and His1113 that present stacking opportunities for the RNA bases (Fig. 3b). Notably, our structure captures the CTS domain forcing apart the RNA stem-loop strands at the base of the stem through the steric barrier defined by an α-helix (hereafter, termed insertion helix), which forks the RNA stem. In the structure, the first three stem base pairs are splayed apart (Fig. 3c) concurrent with Ile1121 and Ile1122 of the insertion helix forming hydrophobic interactions with the splayed bases G-1 and C-46 (Ile1121 and Ile1122), G-45 and U-44 (Ile1122) (Fig. 3c). These interactions induce a distortion in RNA conformation away from the canonical A-form helix at the base of the stem (Extended Data Fig. 5f).

To investigate the role of the insertion helix and the entire CTS domain overall, we generated mutants of L1 ORF2p with the entire CTS domain deleted (ΔCTS) or with the insertion helix replaced by negatively charged residues (ΔIH). Both showed notably reduced TPRT activity, and the ΔCTS protein was further compromised for target-site nicking activity (Fig. 3d), suggesting a role of the CTS domain beyond interacting with and unwinding the template RNA. To validate the structural integrity of L1 ORF2p mutants, particularly, without the CTS domain, we verified that the mutant proteins had similar or greater than wild-type RT activity in our primer extension assays (Extended Data Fig. 1d,e).

The entire template RNA stem is nestled into a positively charged surface composed of the CTS, RBD and thumb domains (Fig. 3e and Extended Data Fig. 5g), which engage but do not contort the RNA stem aside from the stem's base (Extended Data Fig. 5g). To investigate the importance of the RNA stem-loop for L1 ORF2p TPRT activity, we generated AJh template RNA variants that differ from native stem structure by increased (AJhm) or decreased (AJh-uf) base pairing (Supplementary Table 1). While removing mismatches did not increase TPRT product, unpairing the stem-loop with mismatches resulted in a modest decrease in TPRT efficiency and a substantial increase in the heterogeneity of TPRT product lengths, in which shorter than full-length cDNA products were generated (Fig. 3f). These results suggest that the stem-loop could contribute to defining where TPRT initiates within the template RNA.

To examine whether other RT families share a CTS-like domain with a similar function, we searched for a homologous structure across the evolutionary tree. Our structure-based search revealed a distant relationship to nucleic acid-interacting motifs in the *Bombyx mori* R2 retrotransposon protein[48,49] and in the human telomerase catalytic core[50] (Extended Data Fig. 6a,b). However, it remains to be determined whether these partial CTS-like motifs share the same function as the CTS domain in L1 ORF2p. By contrast, primary sequence comparison found homology only within the L1 family. L1 enzymes from fish to human show conservation of the overall hydrophobic content of the CTS-domain insertion helix, with L1 ORF2p Ile1122 being replaced only by another hydrophobic residue (Extended Data Fig. 6c).

## Target-site architecture for TPRT

To investigate what structural features may influence recognition and cleavage of target DNA, we superimposed the structure of the L1 EN domain co-crystallized with DNA duplex[51] onto our full-length L1 ORF2p RNP structure (Fig. 4a). We observed that the consensus cleavage-site (TTTTT/AA) is accessible to the EN domain when located close to the 5' end of the DNA duplex (Fig. 4a (top)). Notably, adding extra DNA base pairs upstream (5' of TTTTT) of the consensus cleavage site introduced a steric clash with the L1 ORF2p CTS domain (Fig. 4a (bottom)). We predicted that as little as around 10 upstream base pairs could severely inhibit EN domain engagement with the target site. To test this structure-based prediction, we designed DNA duplexes with the consensus cleavage site positioned at different distances from the edge of the base-paired duplex. TPRT assays revealed a considerable inhibition of EN nicking activity and subsequent TPRT from an upstream duplex region as short as 11 bp, with optimal EN nicking and TPRT for an upstream duplex of around 7–9 bp (Fig. 4b,c). Off-target EN nicking (not at the consensus site) was common for non-optimal target-site duplexes and occurred between pyrimidine and purine nucleotides, in agreement with non-consensus cleavage in cells[8,30] (Extended Data Fig. 7). Consistent with what would be expected from the structure, deletion of the CTS domain of L1 ORF2s (ΔCTS mutant) enabled nicking of DNA substrates with an upstream duplex region greater than 13 bp (Extended Data Fig. 8). Nonetheless, the ΔCTS mutant did not nick all target sites equally (Extended Data Fig. 8), indicating that there are other determinants of efficient nicking beyond the minimal consensus TTTTT/AA.

L1-mediated TPRT in cells is coupled with DNA replication, with preferential EN nicking of the lagging-strand template[30,52]. We therefore hypothesized that an optimal target site could have a 5' single-stranded DNA overhang upstream of the duplex region containing the EN consensus sequence, a design that mimics the lagging-strand template with an Okazaki fragment primer. To test this possibility, we compared EN nicking and TPRT activity using DNA duplexes with different 5' overhang lengths upstream of the consensus target site. We found that the presence of an overhang was strongly stimulatory, with some influence from the overhang nucleotide composition (Fig. 4d). Notably, increasing the upstream overhang length from 9 to 27 nt gave a marked stimulation of nicking efficiency, with two-thirds of the target DNA containing the longest overhang converted into on-target nicked product (Fig. 4e). Consequently, a sixfold increase in the TPRT product was also observed when increasing the overhang length from 9 to 27 nt (Fig. 4e). We conclude that L1-target sites are partial duplex structures with a long single-stranded 5' overhang, with the EN cleavage site positioned on duplex DNA near the single-strand/duplex transition (Fig. 4f). This structure of optimal target-site DNA architecture supports efficient TPRT by L1 ORF2p (Fig. 4f) and explains why previous reconstitutions resulted in low TPRT efficiency[32]. Our results have profound implications for the understanding of L1 and Alu mobility in the human genome.

## Discussion

### Adaptation for nucleic acid recognition

Phylogenetic characterization suggests that a prokaryotic mobile group IIB intron protein gave rise to eukaryotic single-ORF retrotransposons with a domain architecture like the R2 retrotransposon, which, in turn, spawned two-ORF retrotransposons like those in the L1 family[41]. We compared the L1 ORF2p structure and substrate engagement with that of its ancestral group IIB intron from *Thermosynechococcus elongatus*[45], and with the recently reported cryo-EM structure of non-LTR retrotransposon R2 from *B. mori* (R2Bm)[48,49]. Template RNA binds to L1 ORF2p with similar topology to that of group IIB intron RT binding to intron RNA and that of R2Bm binding to target-site DNA upstream of

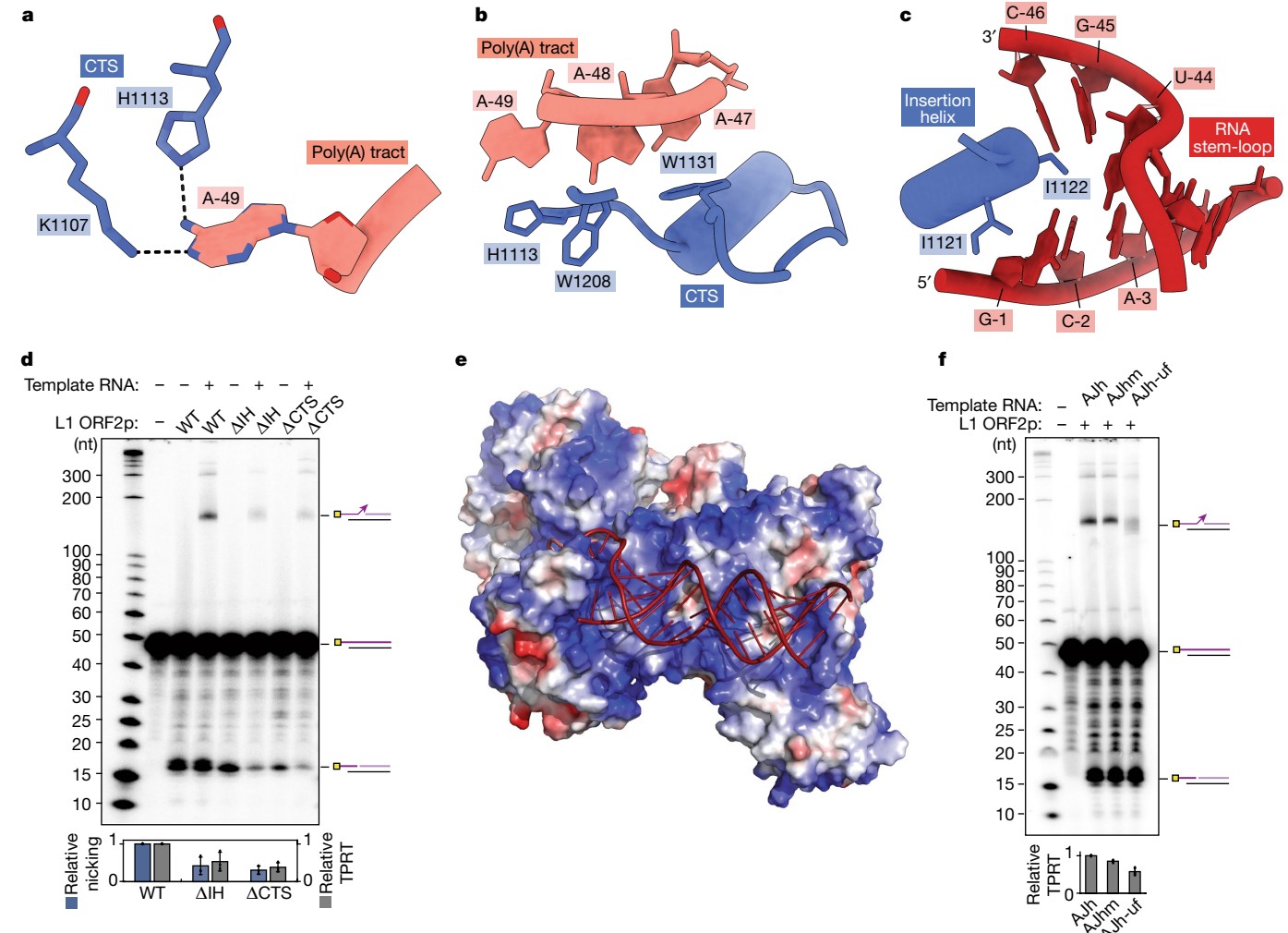

**Fig. 3 | Engagement and unwinding of the template RNA by L1 ORF2p CTS.**
**a**, Base-reading hydrogen bonds between the duplex-proximal poly(A) tract and residues in the CTS. A heteroatom representation (red, oxygen; blue, nitrogen) is shown. **b**, Aromatic side chains from the CTS domain near the 5′ end of the poly(A) tract. **c**, Isoleucine side chains from the CTS insertion helix oblige unwinding of the RNA stem. **d**, Denaturing gel analysis of TPRT reaction products with wild-type L1 ORF2p, ΔIH and ΔCTS mutants was performed using the AJh 25A template RNA (141 nt). The experiment was replicated three independent times. The full-length cDNA was quantified as the TPRT product, and the nicked product at the expected size was quantified independently. Relative EN nicking and TPRT in the +RNA lanes were normalized to the

wild-type L1 ORF2p. The mean ± s.d. of $n = 3$ biologically independent replicates is shown below. **e**, Electrostatic rendering of the surface of L1 ORF2p engaging the RNA stem-loop. Blue corresponds to positively charged surface and red to negatively charged surface. **f**, Denaturing gel analysis of TPRT reaction products with wild-type L1 ORF2p and template RNAs of variable stem-loop structures: AJh, 25A, AluJ half-SINE (141 nt); AJhm, 25A, AluJ half-SINE 25A with reduced stem bulges (142 nt); AJh-uf, 25A, AluJ half-SINE 25A with disrupted stem base-pairing (141 nt). The experiment was replicated three independent times. Full-length cDNA product was quantified as the relative TPRT product, normalized to the full-length cDNA product with AJh 25A. The mean ± s.d. of $n = 3$ biologically independent replicates is shown below.

the nick site (Extended Data Fig. 9). However, and despite their evolutionary relationship, our study highlights major differences between the TPRT strategies of L1 ORF2p and R2Bm proteins. First, while the CTS-like domain of R2Bm melts duplex DNA (Extended Data Fig. 9b), the analogous L1 ORF2p CTS domain can bind to and facilitate unwinding of RNA. Second, the EN domains are in distinct positions relative to their RT cores (Extended Data Fig. 9b). Third, whereas R2Bm engages long duplex DNA with sequence-specific DNA-binding domains, L1 ORF2p has a largely sequence-independent target-site association that relies on limited duplex length 5′ of the target site and a single-stranded DNA overhang.

## Implications for L1 and SINE lifecycles

Together, our structural and biochemical studies reveal insights into the retrotransposition of L1 and SINEs and offer mechanistic rationale for the observed biological properties of L1-mediated genomic insertions

(Fig. 4f). First, the extensive surface of L1 ORF2p dedicated to binding single-stranded poly(A) with adenine-specific contacts favours the use of the poly(A) tract of L1 RNAs or the genome-encoded poly(A) tract of SINEs as initiation sites for cDNA synthesis[27,37,38]. Second, template anchoring to L1 ORF2p by a stem-loop structure can explain how Alu RNAs outcompete the L1 3′ UTR for L1 ORF2p binding[24–27], even if both are associated to the same ribosome, because the L1 3′ UTR lacks a similar stem-loop structure. Third, the long single-stranded DNA upstream of the EN cleavage site required for the activity of L1 ORF2p helps to explain the preference for nicking the lagging DNA strand template at replication forks, and why the chromatin engagement and synthesis of new DNA of L1 ORF2p are coupled with genome replication[30,52].

A complete L1 retrotransposition cycle has been assumed to require nicking of the second strand of a target site before the second-strand synthesis that generates a double-stranded copy of L1 or SINE. The L1 ORF2p target-site architecture, where first-strand cleavage occurs at

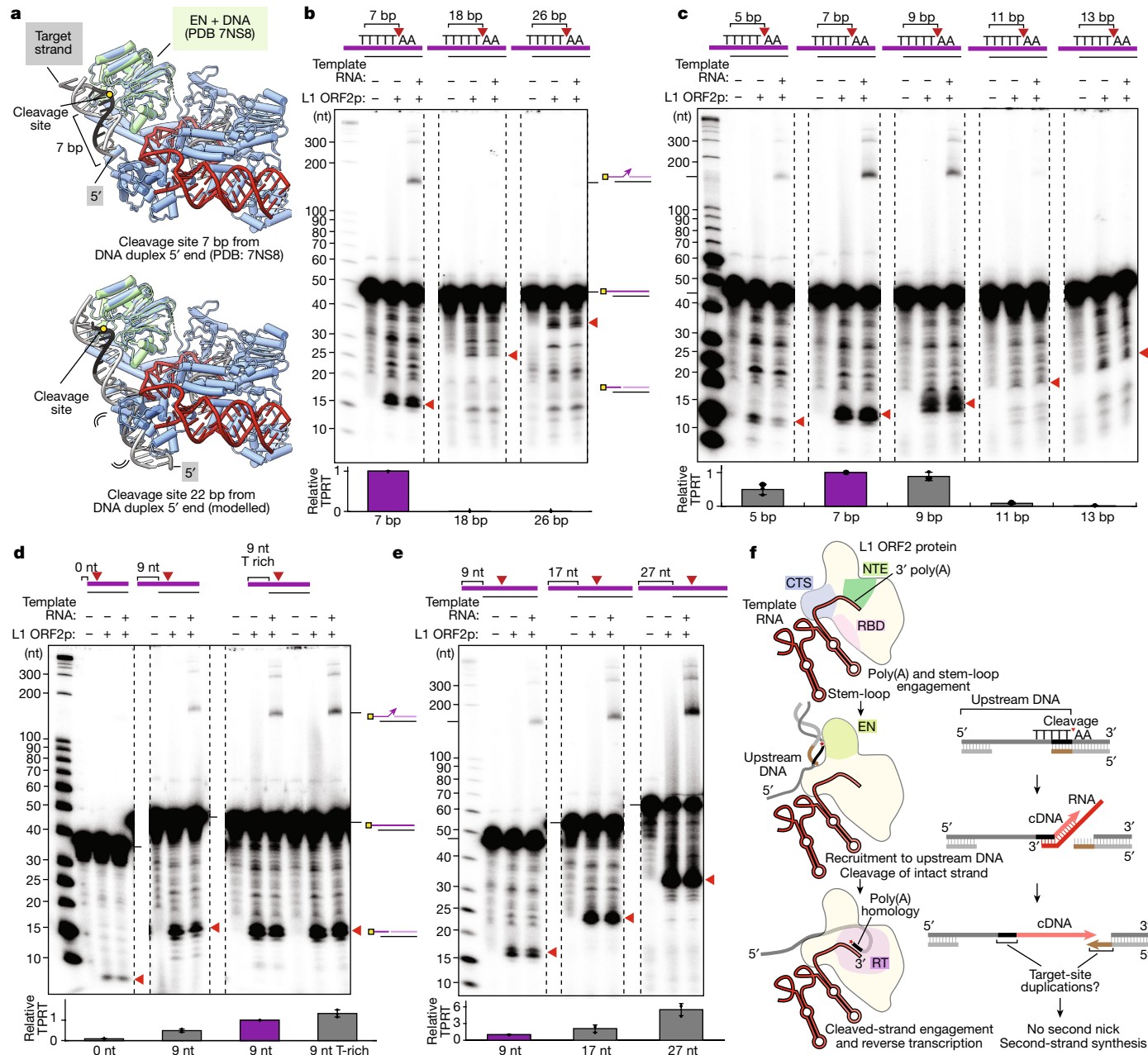

**Fig. 4 | Target-site position and upstream single-stranded DNA determine the efficiency of nicking and TPRT. a**, The full-length L1 ORF2p RNP structure (this study) superposed with a structure of the EN domain: duplex DNA complex (Protein Data Bank (PDB): 78NS). For a cleavage site near the 5′ end of the duplex DNA, there is no steric clash with L1 ORF2p (top). A modelled, longer DNA duplex engaged with the EN domain illustrates the steric clash of upstream duplex DNA with L1 ORF2p (bottom). **b,c**, Denaturing gel analysis of TPRT reaction products using target DNA with a varying cleavage-site position from 7 to 26 bp from the 5′ end of the duplex DNA (**b**) and 5 to 13 bp (**c**). **d,e**, Denaturing gel analysis of TPRT reaction products using target DNA of varying length and sequence of upstream single-stranded DNA. A blunt duplex end and short overhangs with T-rich sequences (**d**), and longer overhang

lengths from 9 to 27 nt (**e**) were used. The red arrowheads in **b**–**e** denote the expected nicked product sizes from cleavage at the consensus target site. The experiments in **b**–**e** were replicated three independent times. The relative amount of full-length cDNA was quantified as the TPRT product. The mean ± s.d. for *n* = 3 replicates is shown. The purple bars in **b**–**e** bar graphs indicate the common DNA target site in all panels, which was used for normalization of the relative TPRT product. The rightmost lanes in **d** use a T-rich overhang with an alternative sequence. AJh 25A (141 nt) was used as the template RNA across **b**–**e**. **f**, Model for the initial stages of template engagement, target-site identification and first-strand synthesis by L1 ORF2p. The overhang single-stranded DNA is drawn near the CTS domain for illustration purposes only.

a limited length of duplex away from a single-strand/duplex transition on the 5′-overhang strand, produces a nick only around 10 bp away from the 5′ overhang. The terminal region of the DNA duplex between the nick and the 5′-overhang would be prone to dissociation, eliminating the need for enzyme-mediated duplex melting or second-strand nicking (Fig. 4f). The target-site DNA architecture also accounts for

sequence duplication surrounding the new L1 insertion, although the observed target-site duplication lengths[8,29] would also depend on other factors, for example, the extent of unpairing of upstream duplex by RPA (replication protein A) from the adjacent single-stranded DNA. L1 ORF2p interaction with factors such as PCNA could facilitate target-site selection[18,19]. The predicted PCNA-interacting protein box motif in L1

ORF2p[18] is located on a highly accessible α-helix of the NTE domain (Extended Data Fig. 10a), and we found that addition of PCNA gives a modest increase in TPRT activity in our biochemical assays, despite the short linear duplex (Extended Data Fig. 10b). Overall, the combination of the target-site structure specificity of L1 ORF2p and its interaction with PCNA can explain preferential insertion into the lagging-strand template behind a replication fork, where there would be an intact leading-strand duplex to support DNA break repair.

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

## Methods

### Protein expression and purification

Full-length human L1 ORF2 DNA was synthesized (Genscript) and cloned into the pFastbac1 vector with His and ZZ-tags. The L1 ORF2p mutation and truncation constructs consisted of the following residues: RT mutant (D702A, D703A), EN mutant (D145A, Y226K)[51], ssRNA (Δss) binding mutant (N371A, R385A, N388A, C804A, R855A, K1107A, H1113A, K1236A), Δinsertion helix (V1117 to K1124 mutated to EDDDDDE), ΔCTS (missing residues 1067–1275). All of the constructs were fully sequenced. The plasmids were transformed into the DH10Bac *E. coli* strain to produce bacmids and transfected into Sf9 cells using the Bac-to-Bac system (Invitrogen). Three rounds of baculoviral expansion were performed and used for infection of Sf9 cells or High Five cells. The insect cells were lysed by sonication and the lysate was clarified by centrifugation at 40,000 rpm in the Ti45 rotor (Beckman Coulter) for 30–45 min. The proteins were purified with the IgG Sepharose resin (Cytiva), eluted by cleavage with TEV protease, followed by a Heparin column (Cytiva) and finally through gel filtration using the Superdex 200 10/300 column (Cytiva). Peak elution fractions were analysed on SDS–PAGE, concentrated, flash-frozen in liquid and stored at −80 °C. Protein concentrations were determined by analysing with Bradford reagent (Bio-Rad) against a known bovine serum albumin standard. Mass spectrometry was performed to verify that the full-length L1 ORF2p protein was obtained.

Human PCNA with N-terminal His-tag was expressed in *E. coli* (Rosetta2 strain) and purified using Ni-NTA-affinity chromatography (Qiagen), followed by a HiTrapQ column (Cytiva) and finally through gel filtration on the Superdex 200 10/300 column (Cytiva). Peak elution fractions were analysed using SDS–PAGE, concentrated, flash-frozen in liquid nitrogen and stored in −80 °C. Protein concentrations were determined by analysis with Bradford reagent (Bio-Rad) against a known bovine serum albumin standard.

### RNA transcription and purification

The sequence of the youngest SINE element, AluY, was PCR-amplified from a parent vector[53] to include the T7 RNA polymerase promoter followed by a 25A sequence. The full-length AluJ SINE element sequence[33] was synthesized (IDT) and PCR-amplified to include the T7 RNA polymerase promoter followed by 25A tail. AluJ half SINE RNA was PCR-amplified to isolate the 5′ folded Alu domain followed by variable poly(A) tail from 75A to 5A, non-A tail or ending in 23A-GC for cryo-EM template. L1 3′ UTR sequence of youngest L1 family, L1.3 (GenBank: L19088.1) was synthesized (IDT). Full-length L1 3′ UTR or a truncation lacking 1–78 nt containing a G-quadruplex were PCR-amplified to include the T7 RNA polymerase promoter followed by 25A sequence as the 3′ end. RNA for in vitro reverse transcription assay was designed to result in minimal secondary structure features; transcription templates were synthesized (IDT) and PCR-amplified. All RNAs were transcribed with T7 RNA polymerase in 40–100 μl reactions using the HiScribe T7 High Yield RNA Synthesis Kit (NEB). For high-resolution structure determination, a synthetic template RNA was generated containing a GC-rich hairpin, a 15A sequence followed by a CAATA sequence for L1 ORF2p to polymerize and trap with a dideoxy-G and an 8 nt (TCG-GCGCG) sequence complementary to the DNA primer (Supplementary Table 1). The DNA template for these RNAs was synthesized as complementary oligonucleotides (IDT) to include the T7 RNA polymerase promoter, sense and antisense strands were annealed by heating to 95 °C and slow cooling to 4 °C, and were then transcribed using T7 RNA polymerase as described above. The in vitro transcription reaction was performed for 5 h at 37 °C. The template DNA was removed with DNase RQ1 (Promega), and the transcribed RNA was separated on a 6–9% denaturing polyacrylamide gel. The RNA band was excised and eluted with RNA elution buffer (300 mM NaCl, 10 mM Tris pH 8, 0.5% SDS, 5 mM EDTA) overnight at 4 °C. The RNA was supplemented with

25 μg glycogen and 300 mM $NH_4OAc$ and further precipitated with ethanol, centrifuged and washed with 70% ethanol. The precipitated RNA was air dried before being dissolved in RNase-free $H_2O$ and supplemented with Ribolock (Thermo Fisher Scientific) for long-term storage at −20 °C.

### Cryo-EM sample preparation and data collection

Preparation of graphene oxide grids was adapted from our previously developed protocol[54]. In brief, Quantifoil Au/Cu R1.2/1.3 grids 200-mesh (Quantifoil, Micro Tools) were cleaned by applying two drops of chloroform, then glow discharged. A total of 4 μl of 1 mg ml$^{-1}$ polyethylenimine HCl MAX Linear Mw 40k (PEI, Polysciences) in 25 mM K-HEPES pH 7.5 was applied to the grids, incubated for 2 min, blotted away, washed twice with $H_2O$ and dried for 15 min on Whatman paper. Graphene oxide (Sigma-Aldrich, 763705) was diluted to 0.2 mg ml$^{-1}$ in $H_2O$, vortexed for 30 s, and precipitated at 1,200*g* for 60 s. A total of 4 μl of supernatant was applied to the PEI treated grids, incubated for 2 min, blotted away, washed twice with 4 μl $H_2O$ each and dried for 15 min on Whatman paper before using for grid preparation.

AluJ half SINE RNA for EM (141 nt) was diluted to 10 μM, then refolded in RNase-free $H_2O$ by heating to 70 °C for 5 min followed by slow cooling to 4 °C for 2 h. A 7 nt DNA primer was added to refolded RNA at a 1.5:1 primer:RNA molar ratio and annealed by heating to 30 °C for 3 min and slow cooling to 4 °C to assemble the RD duplex. Synthetic template RNA (74 nt) was diluted to 10 μM, then refolded in RNase-free $H_2O$ by heating to 90 °C for 3 min and snap-cooling to 4 °C. An 8 nt DNA primer was added to the refolded RNA at a 1.5:1 primer:RNA molar ratio and annealed by heating to 45 °C for 3 min and snap-cooling to 4 °C to assemble the RD duplex. The cryo-EM sample was prepared by diluting wild-type L1 ORF2p to 600 nM concentration in cryo-EM buffer (30 mM K-HEPES pH 7.9, 150 mM KCl, 10 mM $MgCl_2$, 5 mM EGTA, 1 mM DTT). Assembled RD duplex was added to L1 ORF2p at a 2:1 RD duplex:protein molar ratio. For synthetic template RNA, dNTPs were added to the reaction to a final concentration of 1 mM dTTP, 1 mM dATP and 1 mM ddGTP to trap the L1 ORF2p-mediated reverse transcription reaction. For SINE RNA, 1 mM dideoxyTTP (ddTTP) was added. The assembled reaction was incubated at 37 °C for 30 s to allow nucleic acid binding and complementary DNA synthesis. BS3 (4 mM; Thermo Fisher Scientific) was added to the reaction to cross-link the sample on ice for 5 min. A total of 4 μl of the sample was applied to the graphene-oxide-coated grid, incubated for 90 s at room temperature and then washed with cryo-EM buffer. The grid was then blotted for 6 s with a blot force of 5 at 20 °C in 100% humidity and vitrified by plunging into liquid ethane using the Vitrobot Mark IV (Thermo Fisher Scientific) system.

For the L1 ORF2p-Alu RNP, micrographs were collected on the Titan Krios microscope (Thermo Fisher Scientific) operated at 300 keV and equipped with a K3 Summit direct electron detector (Gatan). In total, 23,878 videos were recorded using the program SerialEM at a nominal magnification of ×105,000 in super-resolution mode (super-resolution pixel size of 0.405 Å per pixel) and with a defocus range of −1.5 μm to −2.5 μm. The electron exposure was about 50 e$^-$ Å$^{-2}$. Each video stack contained 50 frames. For the L1 ORF2p-synthetic template RNP, the initial reconstruction was obtained from datasets collected on the Talos Arctica microscope. In total, 11,711 videos were recorded at a nominal magnification of ×45,000 in super-resolution mode (super-resolution pixel size of 0.4495 Å per pixel) and with a defocus range of −1.2 μm to −2.5 μm. The electron exposure was about 50 e$^-$ Å$^{-2}$. Each video stack contained 50 frames. For the final reconstruction of the L1 ORF2p-synthetic template RNP, we collected a large dataset on the Titan Krios G3i (Thermo Fisher Scientific) system operated at 300 keV and equipped with a K3 Summit direct electron detector (Gatan) and an energy filter with a slit width of 20 eV. A total of 23,874 videos was recorded at a nominal magnification of ×105,000 in super-resolution mode (super-resolution pixel size of 0.405 Å per pixel), with a defocus

range of −1.0 μm to −2.5 μm. The electron exposure was about 50 e⁻ Å⁻². Each video stack contained 50 frames.

## Cryo-EM data processing

Cryo-EM data processing workflows are outlined in Extended Data Figs. 2 and 3. All video frames were motion-corrected using Motion-Cor2[55,56] in RELION v.3.1.1 and the corresponding super-resolution pixel size was binned 2× during this process. Contrast transfer function (CTF) parameters for each micrograph were estimated using CTFFIND (v.4.1)[57]. For the L1 ORF2p-synthetic template RNP, a subset of micrographs was selected, and around 2,000 particles were manually picked and inspected to train a Cryolo model using Cryolo (v.1.7.6)[58]. The trained models were used to predict particle locations on the entire dataset, for both the initial dataset acquired with a Talos Arctica and the final dataset acquired with the Titan Krios. The particle picks from the Talos Arctica session were imported to cryoSPARC (v.3)[59] to sort particles by 2D classification. A total of 238,798 particles from the initial dataset acquired using the Talos Arctica system were imported back to RELION and a 3D initial model was generated. After 3D classification of this dataset, class 1, containing 89,150 particles with apparent RNA density, was further processed to produce a 4.2 Å reconstruction. For the final Titan Krios dataset, 786,013 particles, obtained after Cryolo picking and 2D classification with cryoSPARC v.3, were imported back to RELION and binned by 2. The 4.2 Å reconstruction from the Talos Arctica dataset was filtered to 25 Å and used as the initial model for a first round of 3D classification. A subset of 222,012 particles displaying a clearer RNA density was selected, re-extracted with no binning and refined to 3.3 Å. RNA-focused 3D classification without alignment was then performed and one class that displays the most complete RNA density, containing 120,397 particles, was selected. Particle polishing and CTF refinement was performed on this subset, followed by focused classification without alignment on the poly(A) tract RNA. The final reconstruction was obtained at a nominal resolution of 3.2 Å from 111,564 particles. The cryo-EM map was sharpened with post-processing in RELION for model building and display in the figures.

For the L1 ORF2p–Alu RNP complex, the motion-corrected micrographs were imported to cryoSPARC, 13 million particles were picked with a blob picker and sorted with 2D classification down to 399,535 particles, which were then imported to RELION v.3.1.1 for further processing. A subset of these particles was used to generate an initial 3D model. 3D classification was performed with the entire set of particles into three classes. A subset of 155,822 particles displaying a clear density of the EN domain and Alu RNA stem-loop and 5′ fold was selected and refined to 4.4 Å.

## Model building and refinement

Model building was initiated by rigid-body fitting the AlphaFold[36] model of human L1 ORF2p into the final 3.3 Å cryo-EM density map using UCSF ChimeraX[60]. The EN domain was removed at this point due to the lower resolution in that part of the density map. The L1 ORF2p protein was first manually inspected in COOT[61] to correct the amino acid sequence and then processed for real-space refinement in PHE-NIX[62]. Amino acid side chains were manually inspected in COOT and modified when needed before another round of real-space refinement in PHENIX. Nucleic acid was built using a difference density map generated from the cryo-EM density map with the protein density subtracted. The core RNA–DNA duplex from a yeast RNA Pol III structure (PDB: 5FJ8) and dsRNA from a *Drosophila* Dicer-2 structure (PDB: 7W0C) were first manually docked into the cryo-EM map using UCSF ChimeraX. The L1 ORF2 RNP was then manually rebuilt in COOT using the nucleic acid difference map and the correct RNA and DNA sequences bound to the protein core and the dsRNA sequence bound to L1 ORF2p. We modelled the canonical (A-form) dsRNA bound to L1 ORF2p, but we are unable to rule out other helical forms in all or part of this segment due to the lower resolution of this region of the density map. The single-stranded

RNA was built de novo in COOT using the nucleic acid difference map. The model was corrected to include the ddG in the terminating DNA polymer obtained from PDB 1QSS, and the following unincorporated dTTP obtained from PDB 1CR1. Both were docked into the density map using UCSF Chimera and manually rebuilt with the corresponding DNA chain in COOT. The model was processed for global refinement using iterative rounds of real-space refinements in PHENIX with rotamer and Ramachandran restraints. For ddG, ligand restraints were generated in PHENIX using the eLBOW tool. For the dTTP, ligand restraints were obtained from the PDB. PHENIX refinements were performed with these input restraints. At this point, the EN domain from the AlphaFold model of human L1 ORF2p was manually docked in UCSF Chimera and merged into the model with COOT. The complete model was then processed for final real-space refinement and validation in PHENIX. Model building and validation statistics are listed in Extended Data Table 1.

## In vitro RT reactions

For RT assays, the DNA primer was 5′-labelled with ³²P γ-ATP (Perkin Elmer) using T4 PNK (NEB). Unlabelled nucleotide was removed using a spin column (Cytiva). Primer was annealed to the RT template RNA at 1:1 concentration by heating to 75 °C for 3 min and slow cooling to 4 °C for 1 h. RT reactions were assembled on ice in a volume of 20 μl with final concentrations of 25 mM Tris-HCl pH 7.5, 75 mM KCl, 35 mM NaCl, 5 mM MgCl₂, 10 mM DTT, 2% PEG-6K, 100 nM RNA–DNA duplex, 0.1 U μl⁻¹ M-MLV RT (Promega) or 100 nM L1 ORF2p wild-type or mutant protein, 1 mM dNTPs. RT reactions were incubated at 37 °C. The 4.5 μl reaction was withdrawn at 0, 1, 5 and 20 min and mixed with 100 μl of stop solution (50 mM Tris-HCl pH 7.5, 20 mM EDTA, 0.2% SDS). Nucleic acid was purified with 1 volume (100 μl) of phenol–chloroform–isoamyl alcohol and precipitated with 3 volumes of ethanol. The samples were then pelleted at about 18,000g for 20 min at room temperature, washed with 7 volumes of 70% ethanol and pelleted again at about 18,000g for 3 min. The pellet was air-dried, resuspended in 5 μl water and supplemented with 7 μl formamide loading dye (95% deionized formamide, 0.025% (w/v) bromophenol blue, 0.025% (w/v) xylene cyanol, 5 mM EDTA pH 8.0). The sample was heated to 95 °C for 3 min then placed onto ice before loading the sample onto a 7–8% urea–PAGE gel. After electrophoresis, the gel was dried, exposed to a phosphoimaging screen and imaged using the Typhoon Trio (Cytiva) system. To quantitatively compare the RT activity of enzymes, we measured the gel intensity of the full-length cDNA band for all enzymes used at various timepoints using ImageJ. The reaction product generated by M-MLV RT at 5 min was used to normalize each intensity measurement before combining datapoints from three separate repetitions of the RT assay. The mean intensity and its s.d. are plotted for each enzyme at each timepoint in Extended Data Fig. 1e.

## In vitro TPRT reactions

The target DNA site was synthesized (IDT) to have 3′ phosphorylation modification on both the top and bottom strands to block direct extension of the 3′ ends by L1 ORF2p. The target DNA strands were gel-purified with denaturing urea–PAGE (Supplementary Table 1), with the top strand containing the cleavage (TTTTTAA) sequence. The top strand was 5′ labelled with ³²P γ-ATP (Perkin Elmer) using T4 PNK (NEB). Unlabelled nucleotide was removed using a spin column (Cytiva). The two strands were annealed at an equimolar ratio by heating to 95 °C and slow cooling to 4 °C over 1.5 h. The template RNA was independently refolded by melting at 70 °C for 5 min and snap-cooling to 4 °C before assembling the reaction. TPRT reactions were assembled in a volume of 10 μl with final concentrations of 25 mM Tris-HCl pH 7.5, 75 mM KCl, 35 mM NaCl, 5 mM MgCl₂, 10 mM DTT, 2% PEG-6K, 1 mM dNTPs, 50 nM annealed DNA duplex, 50 nM template RNA, 0.4 U μl⁻¹ M-MLV RT (Promega), 200 nM L1 ORF2p wild-type or mutant proteins. Buffer or 200 nM PCNA was added in addition to L1 ORF2p at a 1:1 molar ratio in Extended Data Fig. 10b. TPRT reactions were incubated at 37 °C for

30 min and mixed with 90 μl of stop solution (50 mM Tris-HCl pH 7.5, 20 mM EDTA, 0.2% SDS). Nucleic acid was purified with 1 volume (100 μl) of phenol–chloroform–isoamyl alcohol and precipitated with 3 volumes of ethanol. The samples were then pelleted at around 18,000g for 15 min at room temperature, washed with 7 volumes of 70% ethanol and pelleted again at about 18,000g for 3 min. The pellet was air-dried resuspended in 5 μl water and supplemented with 7 μl formamide loading dye (95% deionized formamide, 0.025% (w/v) bromophenol blue, 0.025% (w/v) xylene cyanol, 5 mM EDTA pH 8.0). The sample was heated to 95 °C for 3 min then placed onto ice before loading the sample onto a 9% urea–PAGE gel. After electrophoresis, the gel was dried, exposed to a phosphoimaging screen and imaged using the Typhoon Trio (Cytiva) system. To quantitatively compare the EN nicking and TPRT activity across distinct target sites (Fig. 4b–e), distinct template RNAs (Figs. 2c and 3f), protein mutations (Figs. 2g and 3d) or with addition of co-factors (Extended Data Fig. 10b), we measured the gel intensity of the full-length TPRT product with ImageJ. The relative TPRT product was measured by dividing the total TPRT product generated with each template RNA, target site or protein mutation by the total product for the condition used for the normalization, highlighted in each figure legend. The relative EN nicking activity was measured by dividing the total nicked target generated with each protein site by the total nicked target for the condition used for the normalization, highlighted in each figure legend. The experiment and analyses were repeated three independent times and the resulting average and its s.d. is plotted in the bar graphs below each gel.

## Bioinformatics analysis

Structure-based search for L1 ORF2p CTS homologues was performed by isolating the coordinates for the CTS and comparing against 3D structures using the DALI server[63]. Two hits for RTs included the insect non-LTR retroelement (PDB: 8GH6) and human TERT (PDB: 7BG9). The CTS was aligned with these coordinates using the MatchMaker tool in ChimeraX and displayed in Extended Data Fig. 6.

The L1 ORF2p family of protein sequences was collected from a recent study[18] and by searching for similar proteins in the UniProt database[64]. In total, 14 full-length sequences were aligned using the Multiple Sequence Comparison by Log-Expectation (MUSCLE) tool in SnapGene v.6.0 (www.snapgene.com). Local alignments near the region of interest are displayed in Extended Data Fig. 6c and the corresponding GenBank accession number or UniProt ID for each sequence is listed.

## Comparison with R2 RT and group II intron RT

*B. mori* R2 RT (PDB: 8GH6) and the *T. elongatus* group IIB intron RT (PDB: 6ME0) were aligned with human L1 ORF2p protein chain using the MatchMaker tool in UCSF ChimeraX.

## Reporting summary

Further information on research design is available in the Nature Portfolio Reporting Summary linked to this article.

## Data availability

The 3.2 Å cryo-EM map reported here has been deposited at the Electron Microscopy Data Bank (EMD-42637) and the corresponding atomic model has been deposited at the PDB (8UW3). All of the other datasets generated and analysed during the current study are available from the corresponding authors on request.

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

**Acknowledgements** We thank the members of the Nogales and Collins laboratories, especially L. Ferguson for help with sequencing analysis, and Z. Yang and B. Van Treeck for discussions; D. Toso and R. Thakkar at the Cal-Cryo EM facility at QB3-Berkeley for help with EM data acquisition; P. Tobias and K. Stine for computing support; and A. Killilea at the UC Berkeley MCB Department cell culture facility. This work was supported by the Damon Runyon Postdoctoral Fellowship DRG-2437-21 (to A.T.), the National Institutes of Health (NIH) grant R35-GM127018 (to E.N.) and the National Institutes of Health grant DP1 HL156819 (to K.C.). E.N. is a Howard Hughes Medical Institute (HHMI) Investigator. C. Bustamante provided funding for A.J.F.A. through NIH grant R01GM032543 and HHMI.

**Author contributions** A.T., E.N. and K.C. conceived the project. A.T. collected and analysed the EM data, performed manual model building and refinements, analysed the structures and performed biochemical assays. A.J.F.A. obtained the initial L1 ORF2p structure and provided advice on single-particle analysis and model building. A.T. wrote the paper with input and revisions from all of the authors.

**Competing interests** K.C. is an equity holder and scientific advisor for Addition Therapeutics, which uses a different retrotransposon protein for gene therapy technology. UC Berkeley has filed a provisional patent application on L1 ORF2p RNP engineering with all of the authors listed as inventors.

**Additional information**
**Correspondence and requests for materials** should be addressed to Akanksha Thawani, Eva Nogales or Kathleen Collins.

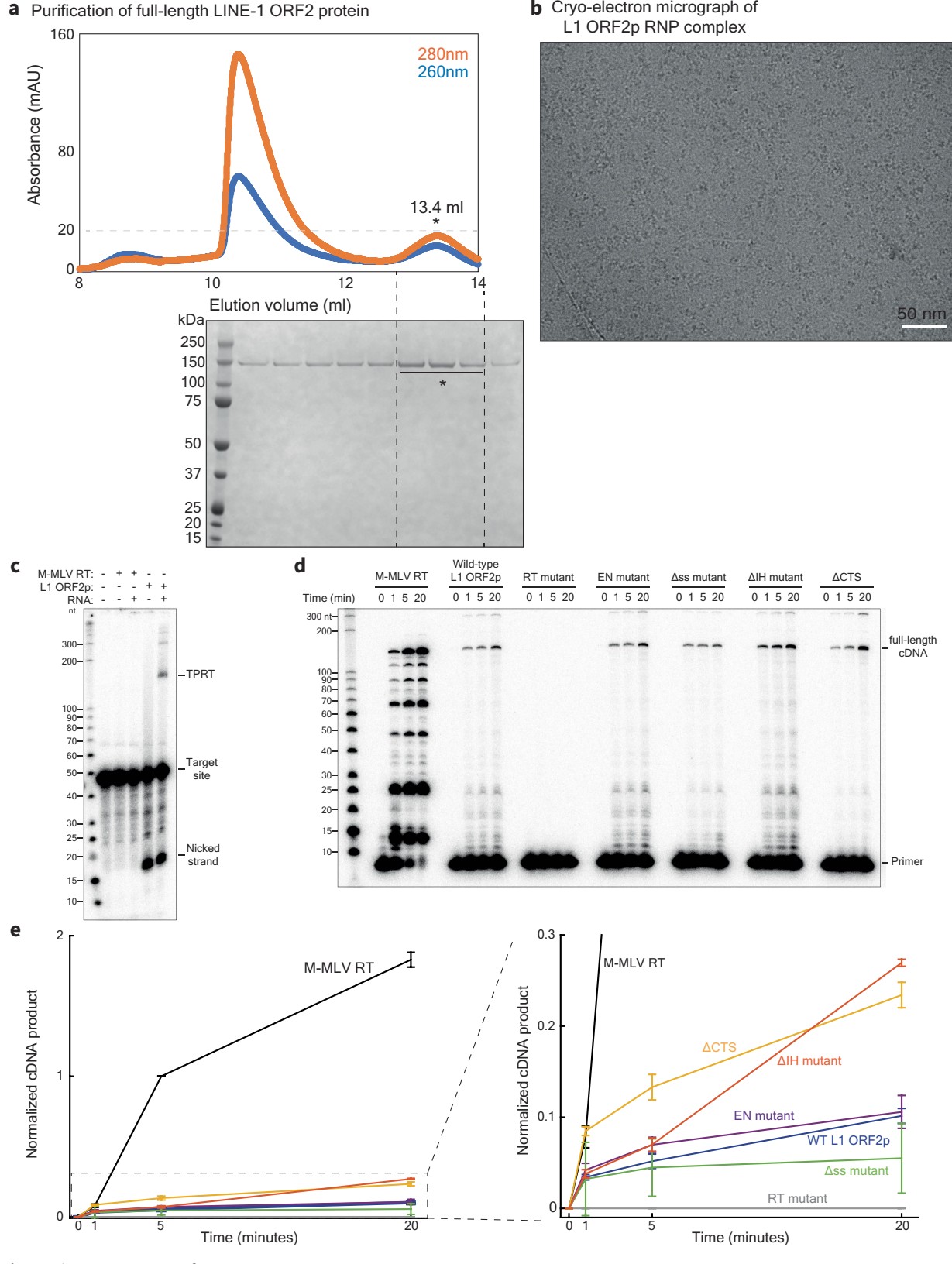

**a** Purification of full-length LINE-1 ORF2 protein

**b** Cryo-electron micrograph of L1 ORF2p RNP complex

**Extended Data Fig. 1** | See next page for caption.

**Extended Data Fig. 1 | Purification, electron microscopy and reverse transcriptase activity of human L1 ORF2p and mutants.** (a) Size exclusion chromatogram (top) and SDS-PAGE of the final step of L1 ORF2p purification stained with Coomassie dye (bottom). The experiment was replicated more than 10 independent times. (b) Cryo-electron micrograph of L1 ORF2p-RNP complex. The experiment was replicated more than 10 independent times. (c) Denaturing gel analysis of TPRT reaction products with M-MLV RT (negative control) and wild-type L1 ORF2p using AJh 25 A as the template RNA (141nt). The experiment was replicated 3 independent times. (d) Denaturing gel analysis of the amount of reverse transcribed product with RT template RNA (129 nt), base-paired at its 3′ end to a 9 nt primer, after 0, 1, 5 and 20 minutes by M-MLV RT, wild-type L1 ORF2p and L1 ORF2p mutants. RT mutant is RT-dead, and EN mutant is EN-dead. (e) Intensity of full-length cDNA product was quantified and plotted across time for all proteins. The experiment in (d) was replicated 3 independent times, cDNA product was normalized by the cDNA product generated by M-MLV RT at 5 min, and the mean and standard deviation across three repeats are plotted.

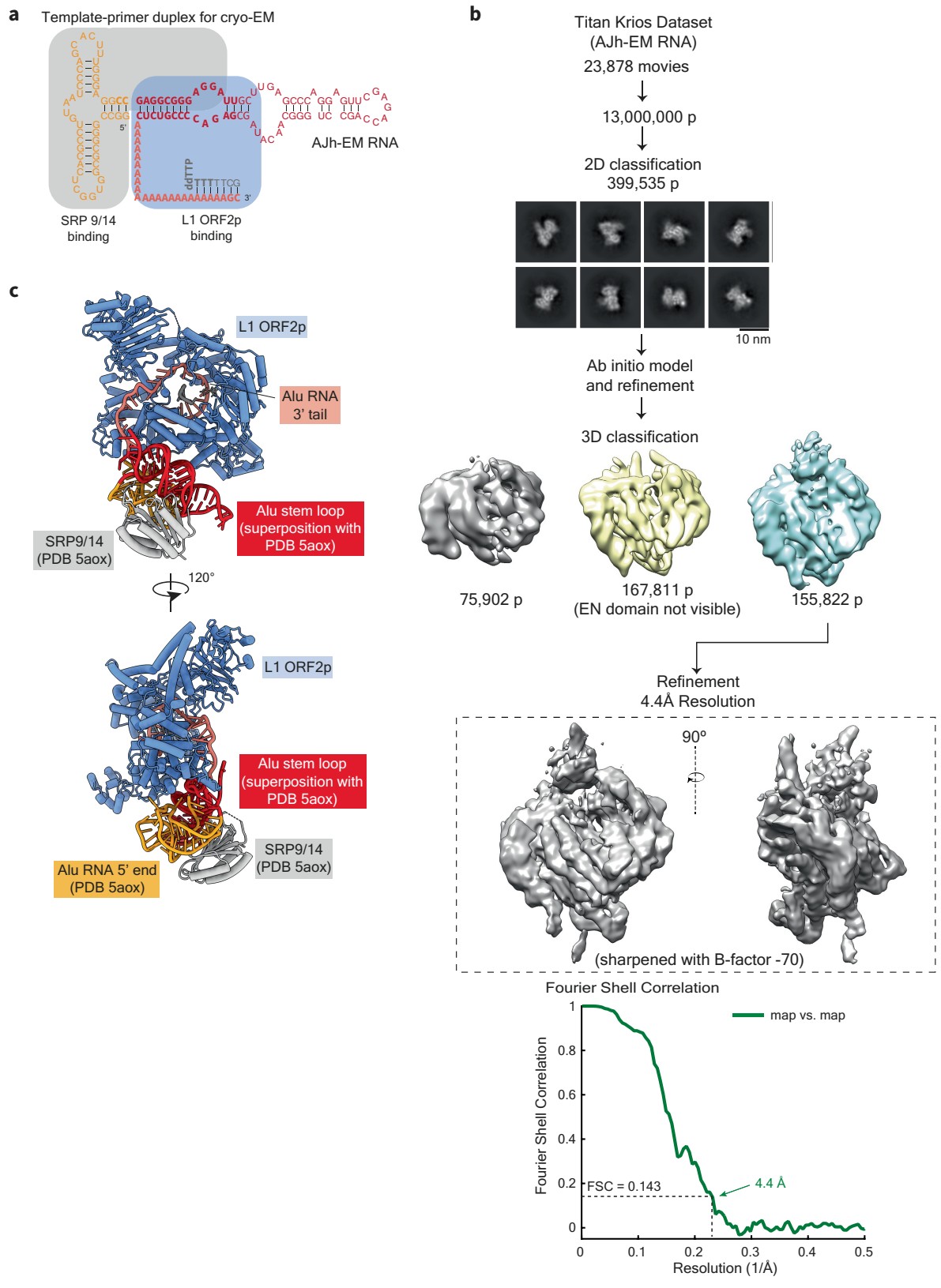

**Extended Data Fig. 2 | Cryo-EM of L1 ORF2p RNP with Alu RNA.** (a) Secondary structure schematic of AluJ half-SINE (AJh-EM) RNA and the DNA primer extended by the addition of dideoxy-TTP used for cryo-EM. Bold nts denote the RNA and DNA bases visible in the cryo-EM density map. RNA regions that bind SRP9/14 and L1 ORF2p are denoted with grey and blue shading, respectively. (b) Cryo-EM data processing pipeline for L1 ORF2p in complex with the Alu RNA and base-paired primer. A final cryo-EM density map at 4.4 angstrom resolution and the corresponding FSC curve are displayed. (c) SRP9/14 bound to AJh RNA (PDB 5AOX[33]) was superimposed with L1 ORF2p-Alu RNA structure using the common RNA stem between the two structures, showing engagement of the SRP proteins to a distinct Alu RNA domain.

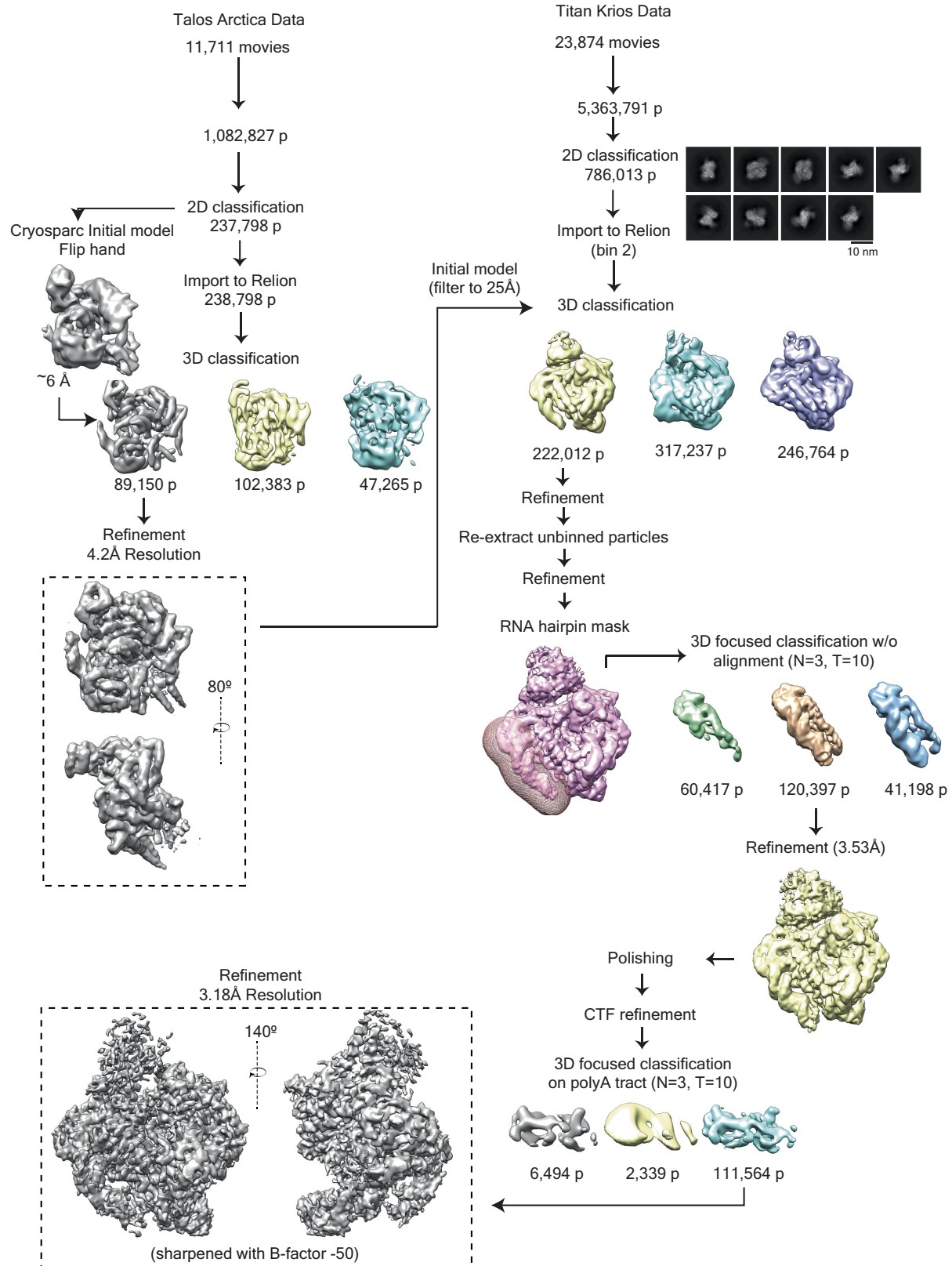

**Extended Data Fig. 3 | Cryo-EM data processing for L1 ORF2p RNP complex bound to synthetic template RNA.** Summary of single particle analysis pipeline leading to the reconstruction of the L1 ORF2p RNP engaged with the synthetic template RNA described in Figs. 1–4 of this paper.

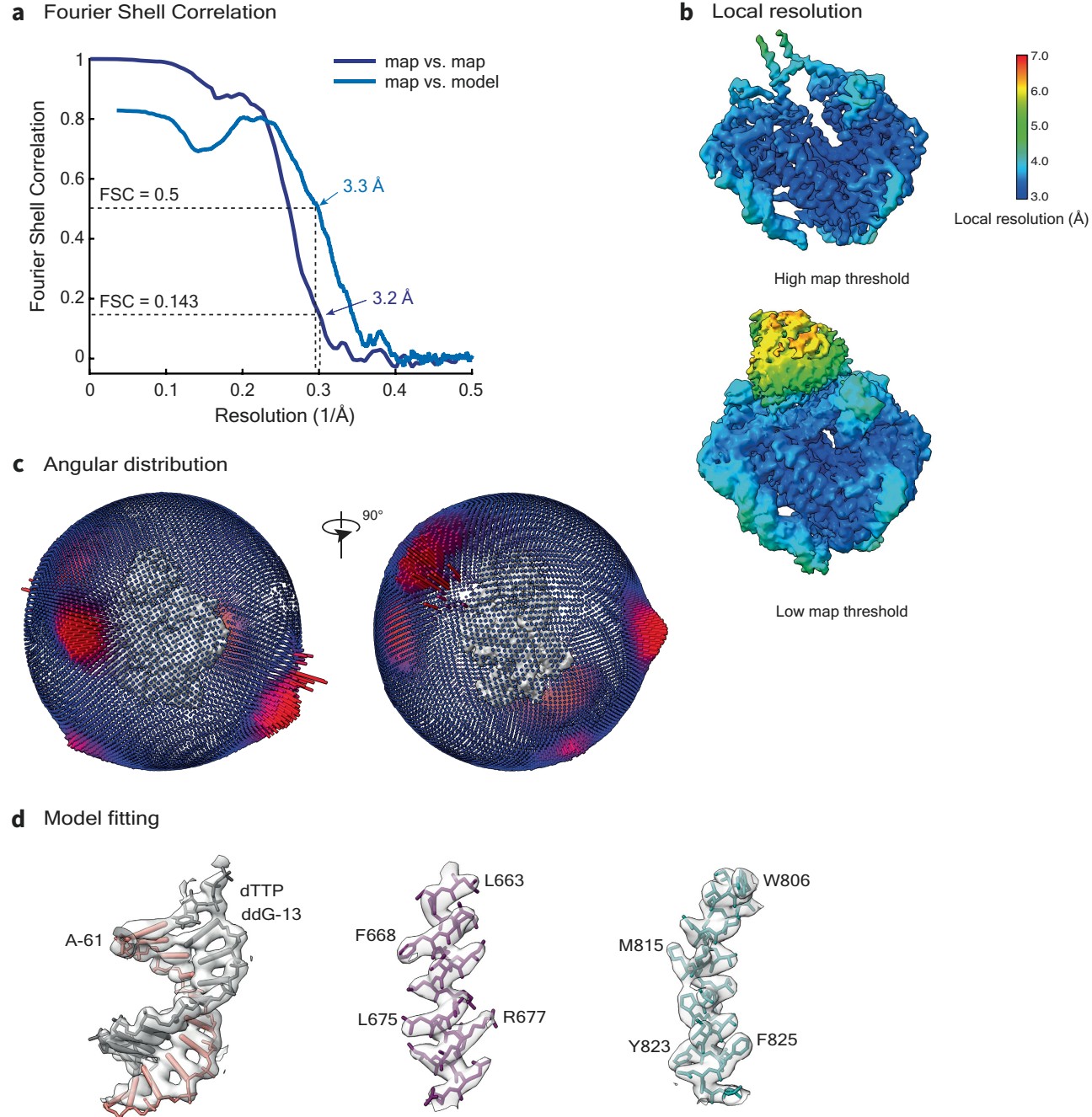

**a** Fourier Shell Correlation

**b** Local resolution

High map threshold

Low map threshold

**c** Angular distribution

**d** Model fitting

**Extended Data Fig. 4 | Resolution estimation.** (a) Gold-standard FSC curve for the L1 ORF2p RNP map, and map versus model FSC obtained from the final model after validation in Phenix (b) Unsharpened density map obtained from analysis in Extended Data Fig. 4 was coloured by local resolution as estimated by Relion 3.1. (c) Particle orientation distribution in the final reconstruction. (d) Representative map densities with atomic models for regions of RNA-DNA duplex and protein.

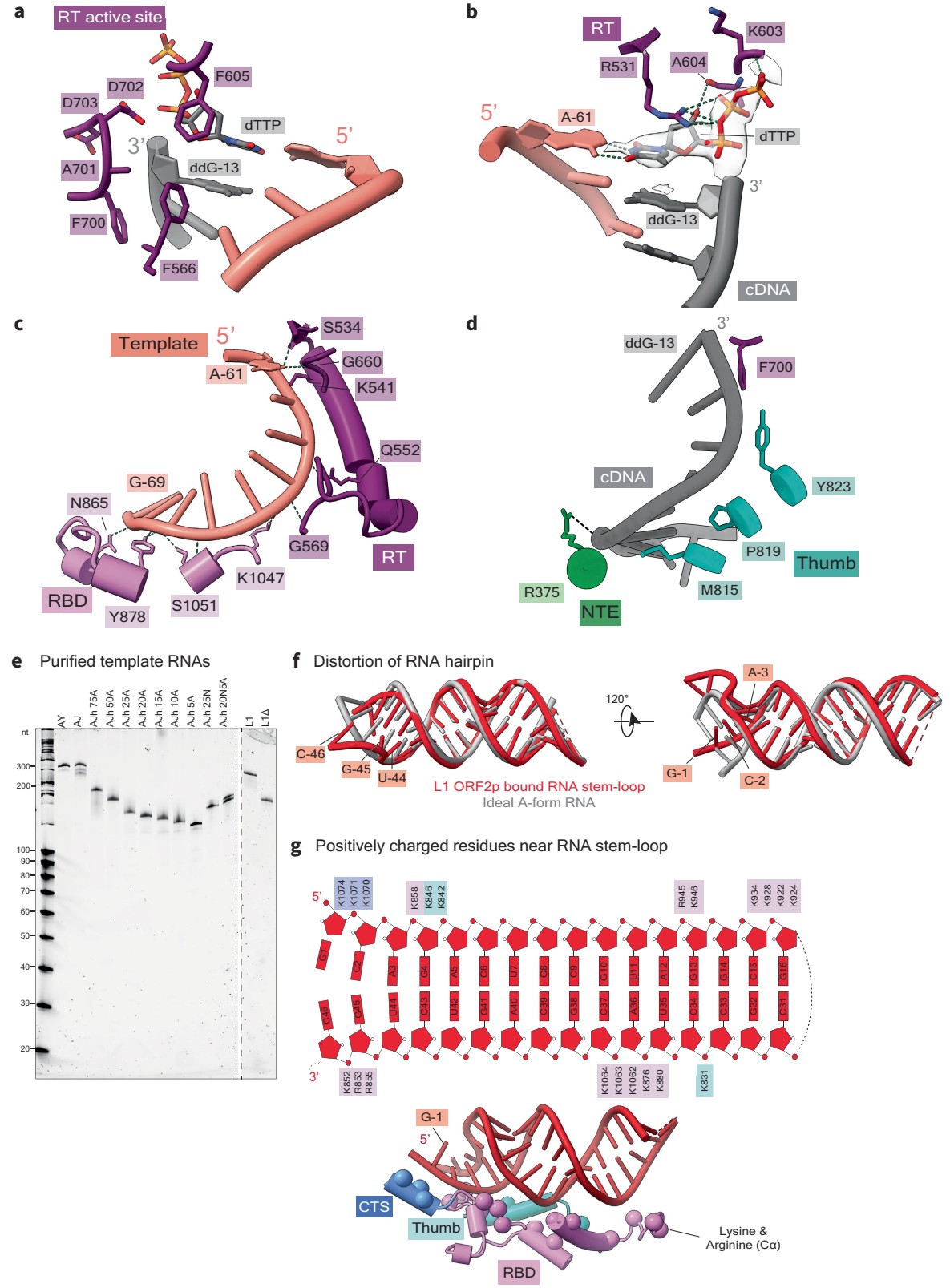

**Extended Data Fig. 5 | Active site conformation and supporting data for investigation of the poly(A) tract and stem-loop engagement.** (a) RT active site residues involved in hydrophobic interactions with the DNA and incoming dTTP are shown relative to the metal-binding aspartic acid side chains of the active site (D702 and D703). (b) Hydrogen bonding interactions with the incoming dTTP. Density of the EM map for the dTTP is displayed. (c) Interactions of RT and RBD domain side chains with the duplex region of the template RNA. (d) NTE, Thumb and RT domain residues interacting with the DNA primer and the cDNA, including a cDNA hydrogen bond with Arg375 of the NTE. (e) Denaturing gel and SYBR-Gold staining of purified RNAs used in the TPRT assays in Fig. 1c and Fig. 2c, showing their integrity and migration as expected. (f) Deviation of the RNA stem-loop from a canonical A-form helix at the end contacted by the CTS insertion helix. (g) Schematic of positively charged L1 ORF2p residues surrounding the RNA stem-loop. Cα positions of all lysines and arginines near the RNA stem-loop from the CTS, RBD and Thumb domain are displayed (bottom).

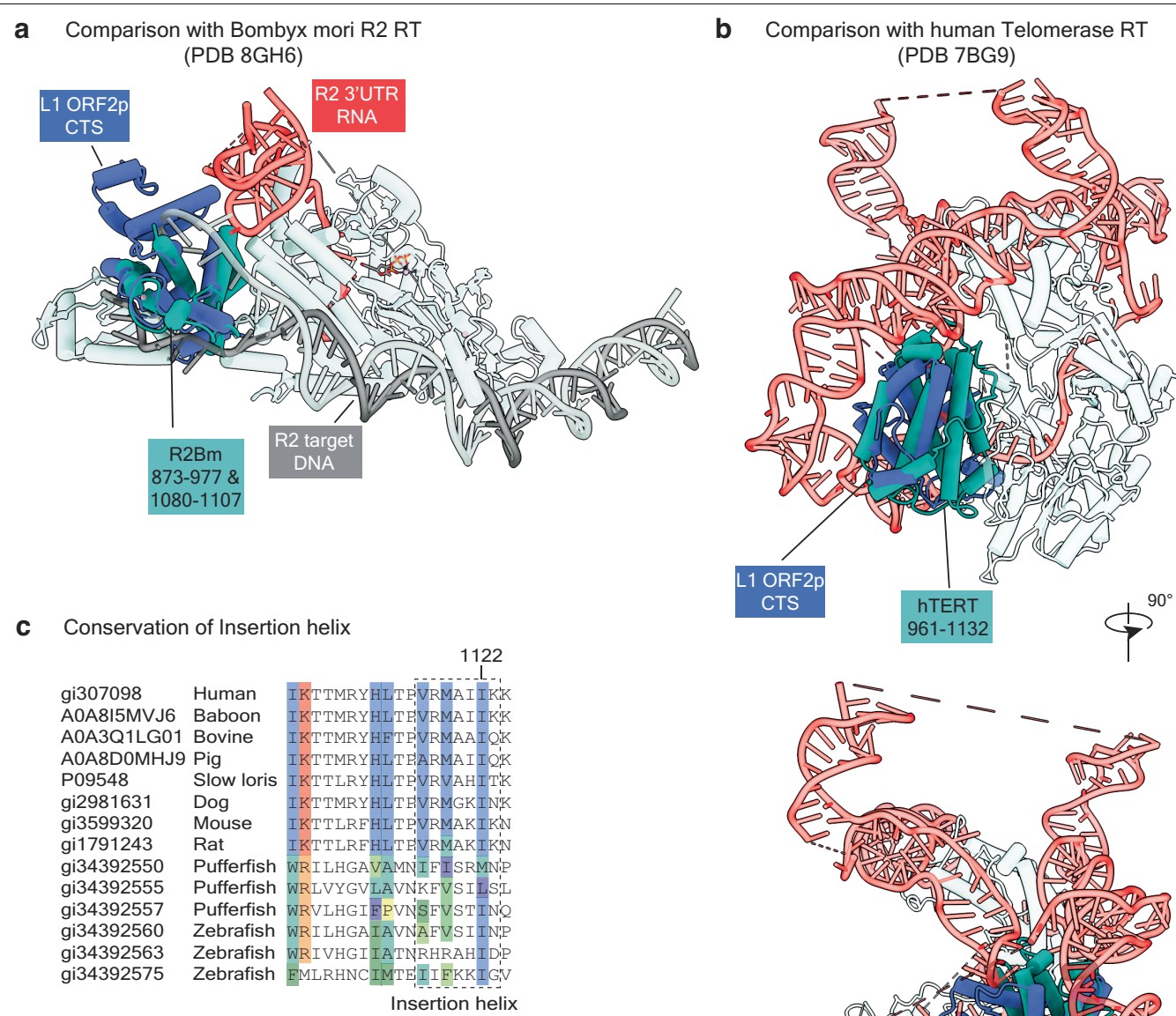

**a**  Comparison with Bombyx mori R2 RT
(PDB 8GH6)

**L1 ORF2p CTS**

**R2 3'UTR RNA**

**R2Bm 873-977 & 1080-1107**

**R2 target DNA**

**b**  Comparison with human Telomerase RT
(PDB 7BG9)

**L1 ORF2p CTS**

**hTERT 961-1132**

90°

**hTR RNA**

**c**  Conservation of Insertion helix

```
                              1122
gi307098     Human       IKTTMRYHLTPVRMAIIKK
A0A8I5MVJ6   Baboon      IKTTMRYHLTPVRMAIIKK
A0A3Q1LG01   Bovine      IKTTMRYHFTPVRMAAIQK
A0A8D0MHJ9   Pig         IKTTMRYHLTPARMAIIQK
P09548       Slow loris  IKTTLRYHLTPVRVAHITK
gi2981631    Dog         IKTTMRYHLTPVRMGKINK
gi3599320    Mouse       IKTTLRFHLTPVRMAKIKN
gi1791243    Rat         IKTTLRFHLTPVRMAKIKN
gi34392550   Pufferfish  WRILHGAVAMNIFISRMNP
gi34392555   Pufferfish  WRLVYGVLAVNKFVSILSL
gi34392557   Pufferfish  WRVLHGIFPVNSFVSTINQ
gi34392560   Zebrafish   WRILHGAIAVNAFVSIINP
gi34392563   Zebrafish   WRIVHGIIATNRHRAHIDP
gi34392575   Zebrafish   FMLRHNCIMTEIIFKKIGV
```
Insertion helix

**Extended Data Fig. 6 | Structure and sequence-based bioinformatics analysis on L1 ORF2p CTS domain.** (a-b) Comparison of the L1 ORF2p CTS domain structure and CTS-like structures in the *Bombyx mori* R2 enzyme in (a) and in human telomerase reverse transcriptase in (b). (c) Sequence conservation for the insertion helix showing that Ile1122 is highly conserved across predicted proteins from the L1 family. The IDs gi*xxx* represent Genbank accession codes and A*xxxx* and P*xxxx* are Uniprot IDs.

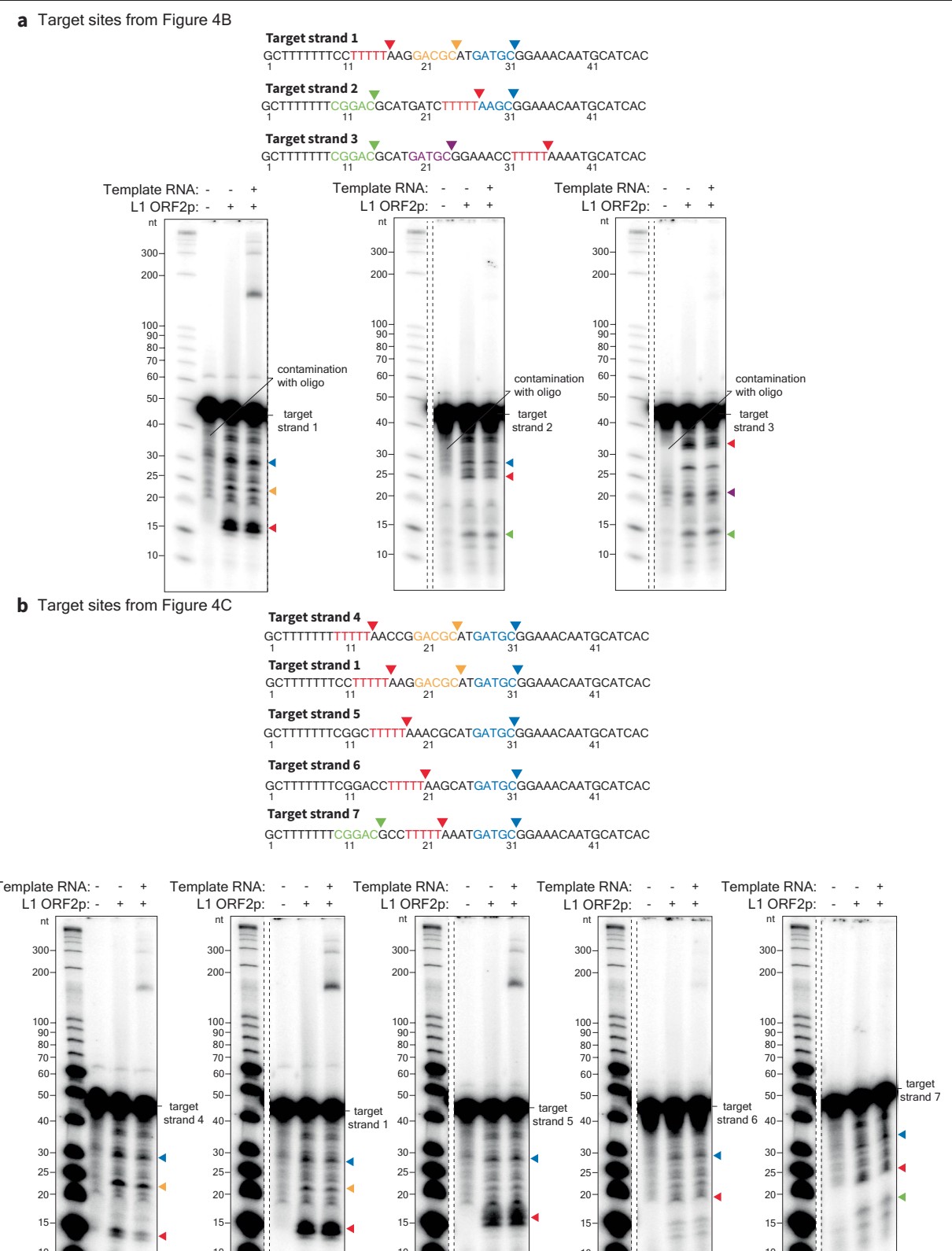

**Extended Data Fig. 7 | Analysis of off-target cleavage by L1 ORF2p.** (a-b) The target DNA sequences from Fig. 4b, c are indicated in (a) and (b), respectively, and L1 ORF2p cleavage products are matched to sequence using different colours of arrowhead. The green, yellow, and blue shades represent off-target cleavage products, while red represent on-target cleavage. The annotated off-target cleavage products are consistent with the L1 ORF2p cleavage site analysis described in a previous work[30].

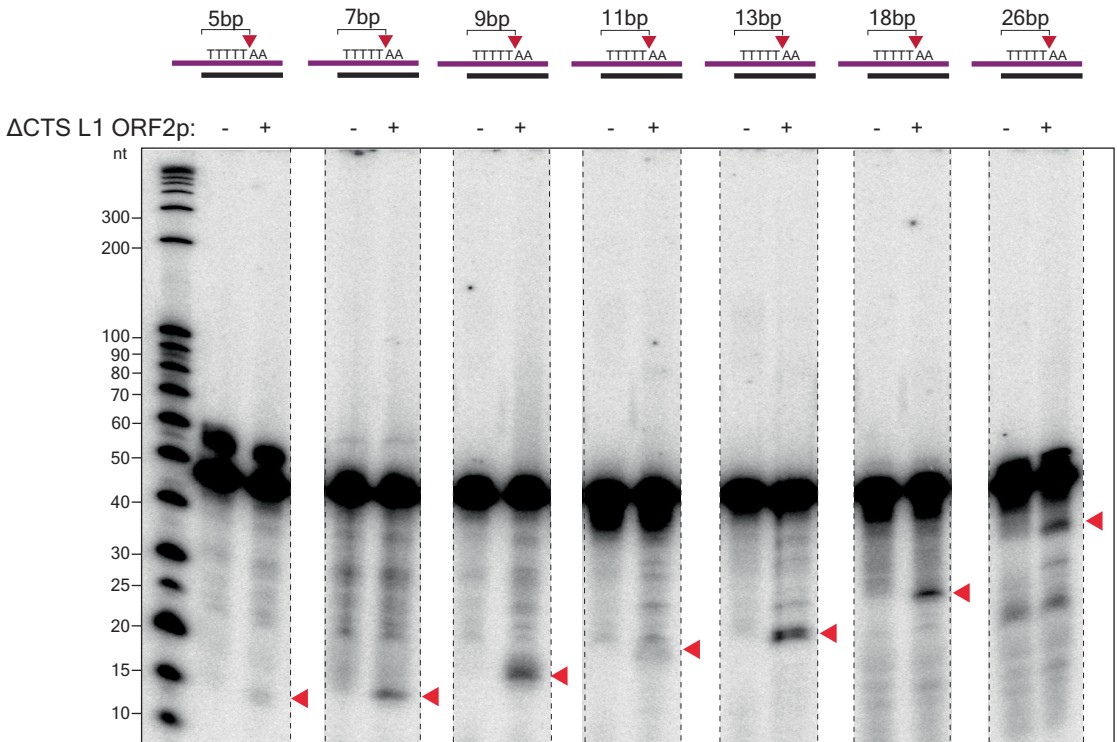

**Extended Data Fig. 8 | Analysis of target cleavage by ΔCTS L1 ORF2p.** Denaturing gel analysis of EN cleavage products using target DNA with varying position of the cleavage site varied between 7 and 26 bp from the 5′ end of the duplex DNA, as denoted in the schematics above each set of lanes. Expected nicked product size from cleavage at the consensus target site is denoted with a red arrowhead. The experiment was replicated three times.

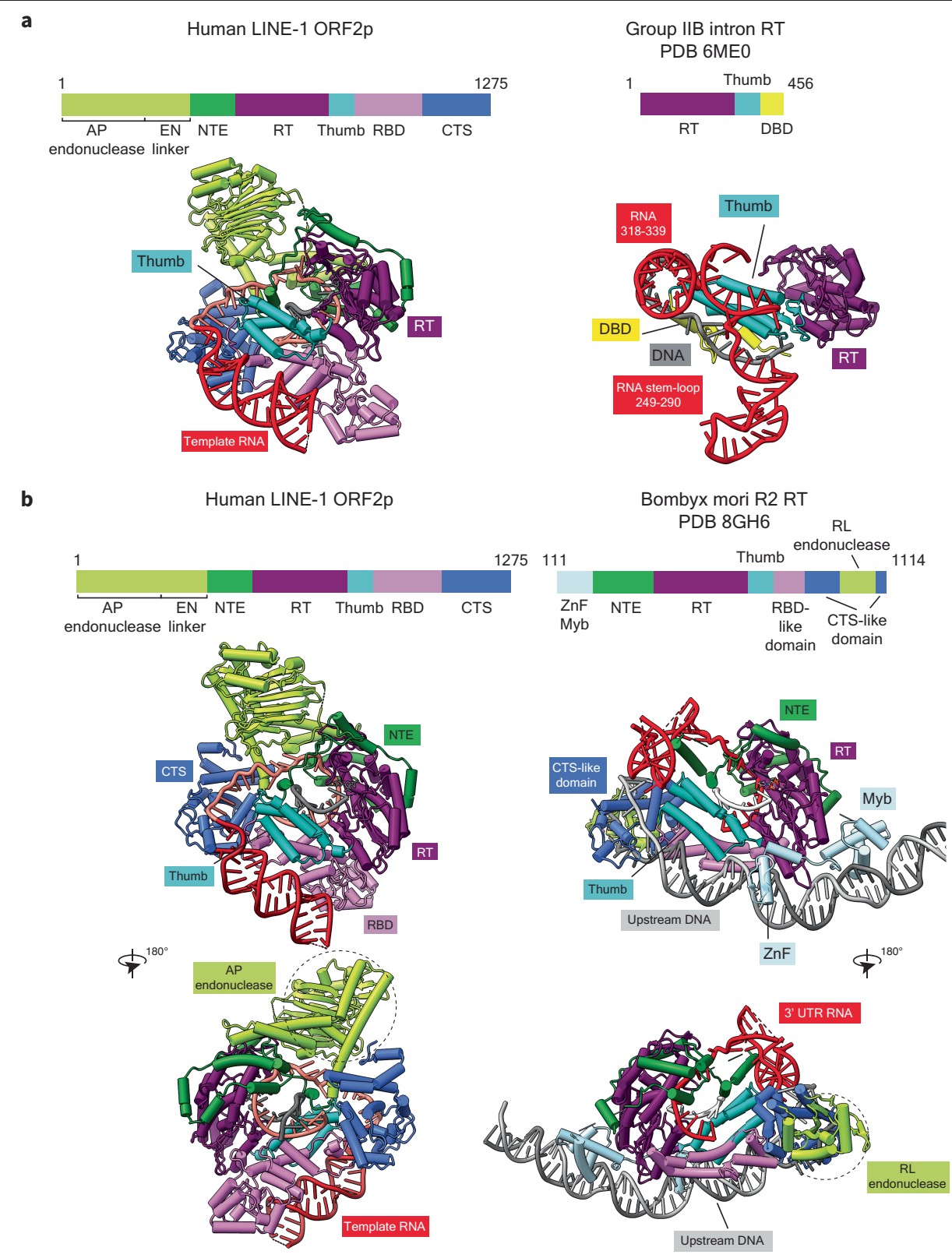

**Extended Data Fig. 9 | Comparison between the L1 ORF2p RNP and related structures.** (a) Comparison to target DNA-engaged Group IIB intron RNP structure with the RT protein bound to intron RNA (PDB 6ME0)[45]. The RT domains are coloured to directly compare with L1 ORF2p. The DNA is coloured grey, the intron RNA is coloured red. DBD, DNA binding domain, coloured yellow. For clarity, only the regions of intron RNA, DNA and the RT protein that have an equivalent in the L1 ORF2p RNP structure are displayed. (b) Comparison with the R2Bm TPRT complex (PDB 8GH6)[48]. R2Bm domains are coloured to directly compare with human L1 ORF2p. The downstream DNA was removed for clarity, while the upstream DNA is coloured grey, and the RNA coloured red for comparison with L1 ORF2p. RL, restriction enzyme like; ZnF, zinc finger domain for DNA binding; Myb, Myb domain for DNA binding.

**a**

PCNA interaction domain

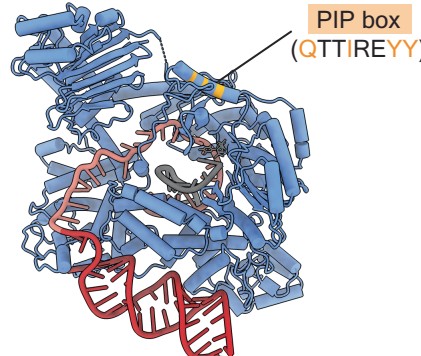

PIP box
(QTTIREYY)

Orientation of PCNA relative to L1 ORF2p
(modeled)

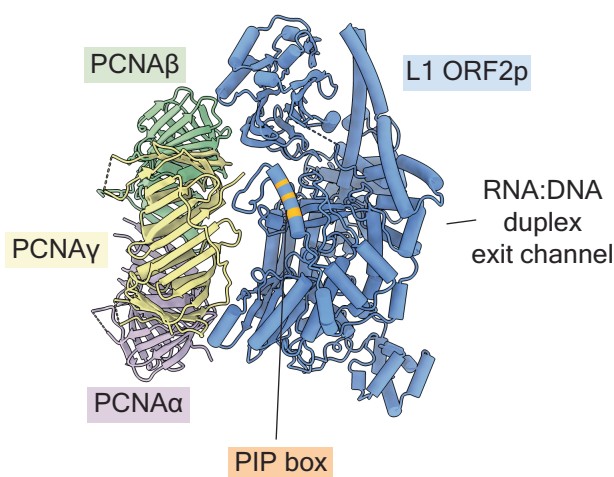

PCNAβ

PCNAγ

PCNAα

L1 ORF2p

RNA:DNA
duplex
exit channel

PIP box

**b**

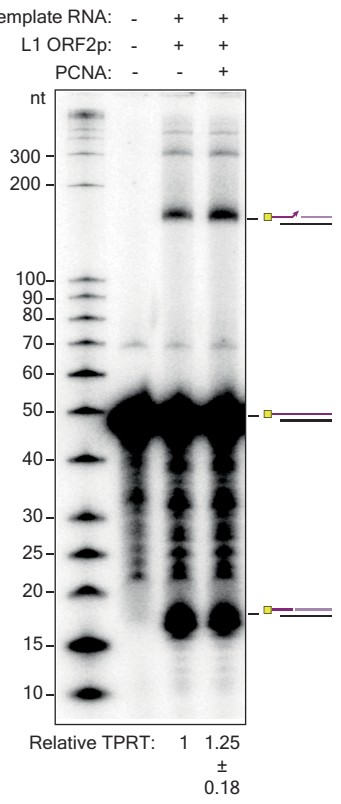

Template RNA:  −  +  +
L1 ORF2p:  −  +  +
PCNA:  −  −  +

nt

300 −
200 −

100 −
90 −
80 −
70 −
60 −

50 −

40 −

30 −

25 −

20 −

15 −

10 −

Relative TPRT:   1   1.25
±
0.18

**Extended Data Fig. 10 | Proposed configuration of PCNA interaction with L1 ORF2p.** (a) Top panel: the predicted PCNA interaction domain (PIP box)[18] within the NTE domain is highlighted. Bottom panel: putative orientation of the PCNA trimer and L1 ORF2p based on existing structures of PCNA with PIP-box containing protein complexes (PDB 7NV0). Based on the superposition of the PIP box, PCNA would be expected to interact near the face of L1 ORF2p for entry of nucleic acids (DNA and RNA), not near the exit channel of the product duplex. (b) Denaturing gel analysis of TPRT reactions with L1 ORF2p in the presence of equimolar PCNA with AJh 25 A (141 nt) as the template RNA. The experiment was replicated three times. Full-length cDNA product was quantified as the relative TPRT product, normalized by the full-length cDNA product without PCNA, and its mean and standard deviation of error across three replicates are displayed below.

**Extended Data Table 1 | Cryo-EM data collection, refinement and validation statistics**

| | Hs L1ORF2p RNP (EMD-42637) (PDB 8UW3) |
|---|---|
| **Data collection and processing** | |
| Magnification | 105,000 |
| Voltage (kV) | 300 |
| Electron exposure (e–/Å$^2$) | 50 |
| Defocus range (μm) | -1.0 to -2.5 |
| Pixel size (Å) | 0.81 |
| Symmetry imposed | *C1* |
| Initial particle images (no.) | 786,083 |
| Final particle images (no.) | 111,564 |
| Map resolution (Å) | 3.2 |
| FSC threshold | 0.143 |
| Map resolution range (Å) | 3.0 to 6.6 |
| | |
| **Refinement** | |
| Initial model used (PDB code) | none (generated in AlphaFold) |
| Model resolution (Å) | 3.3 |
| FSC threshold | 0.5 |
| Map sharpening *B* factor (Å$^2$) | -50 |
| Model composition | |
| Non-hydrogen atoms | 12,012 |
| Protein residues | 1,265 |
| Nucleic acid atoms | 73 |
| Ligands | 1 (dTTP) |
| *B* factors (Å$^2$) | |
| Protein | 162.81 |
| Nucleotide | 90.65 |
| Ligand | 80.32 |
| R.m.s. deviations | |
| Bond lengths (Å) | 0.004 |
| Bond angles (°) | 0.596 |
| Validation | |
| MolProbity score | 2.14 |
| Clashscore | 8.53 |
| Poor rotamers (%) | 2.51 |
| Ramachandran plot | |
| Favored (%) | 94.53 |
| Allowed (%) | 5.39 |
| Disallowed (%) | 0.08 |

# Reporting Summary

## Statistics

For all statistical analyses, confirm that the following items are present in the figure legend, table legend, main text, or Methods section.

| n/a | Confirmed | |
|---|---|---|
| ☐ | ☒ | The exact sample size (*n*) for each experimental group/condition, given as a discrete number and unit of measurement |
| ☐ | ☒ | A statement on whether measurements were taken from distinct samples or whether the same sample was measured repeatedly |
| ☒ | ☐ | The statistical test(s) used AND whether they are one- or two-sided *Only common tests should be described solely by name; describe more complex techniques in the Methods section.* |
| ☒ | ☐ | A description of all covariates tested |
| ☒ | ☐ | A description of any assumptions or corrections, such as tests of normality and adjustment for multiple comparisons |
| ☐ | ☒ | A full description of the statistical parameters including central tendency (e.g. means) or other basic estimates (e.g. regression coefficient) AND variation (e.g. standard deviation) or associated estimates of uncertainty (e.g. confidence intervals) |
| ☒ | ☐ | For null hypothesis testing, the test statistic (e.g. *F*, *t*, *r*) with confidence intervals, effect sizes, degrees of freedom and *P* value noted *Give P values as exact values whenever suitable.* |
| ☒ | ☐ | For Bayesian analysis, information on the choice of priors and Markov chain Monte Carlo settings |
| ☒ | ☐ | For hierarchical and complex designs, identification of the appropriate level for tests and full reporting of outcomes |
| ☒ | ☐ | Estimates of effect sizes (e.g. Cohen's *d*, Pearson's *r*), indicating how they were calculated |

*Our web collection on statistics for biologists contains articles on many of the points above.*

## Software and code

Policy information about availability of computer code

| | |
|---|---|
| Data collection | Serial EM 4-0-20 for cryo-EM data collection |
| Data analysis | The EM softwares used: Relion 3.1.1, cryoSPARC v.3, cryoSPARC v.4, Cryolo 1.7.6. Structures were built using Coot 0.8.9, Chimera 1.14, ChimeraX 1.3, Phenix 1.20, Pymol 2.5.4. Gels were analyzed using ImageJ (Fiji) 2.1.0 |

For manuscripts utilizing custom algorithms or software that are central to the research but not yet described in published literature, software must be made available to editors and reviewers. We strongly encourage code deposition in a community repository (e.g. GitHub). See the Nature Portfolio guidelines for submitting code & software for further information.

## Data

Policy information about availability of data

All manuscripts must include a data availability statement. This statement should provide the following information, where applicable:
- Accession codes, unique identifiers, or web links for publicly available datasets
- A description of any restrictions on data availability
- For clinical datasets or third party data, please ensure that the statement adheres to our policy

The described cryo-EM maps and coordinate files were deposited in the Electron Microscopy Data Bank (EMDB) with accession code EMD-42637 and in Protein Data Bank (PDB) with accession code PDB 8UW3. All other datasets, reagents or resources generated during this study are available upon request from the corresponding authors.

# Research involving human participants, their data, or biological material

Policy information about studies with human participants or human data. See also policy information about sex, gender (identity/presentation), and sexual orientation and race, ethnicity and racism.

| | |
|---|---|
| Reporting on sex and gender | n/a |
| Reporting on race, ethnicity, or other socially relevant groupings | n/a |
| Population characteristics | n/a |
| Recruitment | n/a |
| Ethics oversight | n/a |

Note that full information on the approval of the study protocol must also be provided in the manuscript.

# Field-specific reporting

Please select the one below that is the best fit for your research. If you are not sure, read the appropriate sections before making your selection.

☒ Life sciences  ☐ Behavioural & social sciences  ☐ Ecological, evolutionary & environmental sciences

For a reference copy of the document with all sections, see nature.com/documents/nr-reporting-summary-flat.pdf

# Life sciences study design

All studies must disclose on these points even when the disclosure is negative.

| | |
|---|---|
| Sample size | In total 23,874 microscope raw movies collected from two different grid preparations were used for data processing of the highest resolution structure, sufficient to provide a high resolution structure. This data size was determined in order to reconstruct a high-resolution cryo-EM map for structure determination were obtained (at around 3 angstrom resolution). For low resolution, Alu structure 23,878 microscope raw movies collected from one grid preparations were used for data processing to yield a structure where protein and RNA densities could be clearly fitted at around 4 angstrom resolution. For biochemical assays, at least three independent biological replicates were performed, as recommended and as is the standard in similar works. |
| Data exclusions | Poor resolution data was excluded from cryo-EM analysis through 2D classifications and 3D classifications. This is standard step in single-particle cryo-EM analysis workflow and necessary to obtain highest resolution structures. |
| Replication | All biochemical experiments were repeated in three or more independent replicates, specified within the figure legends for individual experiments. All replicates which showed similar results. Data from all replicates were pooled for quantification and reported in bar graphs |
| Randomization | In the Fourier shell correlation (FSC) measurement step of the Relion 3.1 data processing pipeline, data were randomly divided into two halves resulting in two independently determined 3D volumes that were used for the FSC calculation. |
| Blinding | Data division in the FSC calculation step is a computer-based, unbiased process. Individual processing of different datasets collected from different human heart samples gave rise to the same 3D structures. |

# Reporting for specific materials, systems and methods

We require information from authors about some types of materials, experimental systems and methods used in many studies. Here, indicate whether each material, system or method listed is relevant to your study. If you are not sure if a list item applies to your research, read the appropriate section before selecting a response.

## Materials & experimental systems

| n/a | Involved in the study |
|---|---|
| ☒ | ☐ Antibodies |
| ☐ | ☒ Eukaryotic cell lines |
| ☒ | ☐ Palaeontology and archaeology |
| ☒ | ☐ Animals and other organisms |
| ☒ | ☐ Clinical data |
| ☒ | ☐ Dual use research of concern |
| ☒ | ☐ Plants |

## Methods

| n/a | Involved in the study |
|---|---|
| ☒ | ☐ ChIP-seq |
| ☒ | ☐ Flow cytometry |
| ☒ | ☐ MRI-based neuroimaging |

# Eukaryotic cell lines

Policy information about cell lines and Sex and Gender in Research

| | |
|---|---|
| Cell line source(s) | SF9 cell line for baculovirus generation. SF9 and High5 cells lines for protein production. The SF9 and High5 cells were obtained from Invitrogen, ThermoFisher. |
| Authentication | No authentication of cell lines was performed as they were purchased from reliable commercial sources. |
| Mycoplasma contamination | Cells were tested for mycoplasma contamination and were found to be negative. Cell lines were monitored for doubling time and correct morphology. |
| Commonly misidentified lines (See ICLAC register) | No misidentified cell lines were used in this work. |

# Plants

| | |
|---|---|
| Seed stocks | n/a |
| Novel plant genotypes | n/a |
| Authentication | n/a |

