## [Peer Review File · Nature]

Manuscript Title: Template and target site recognition by human LINE-1 in retrotransposition

Redactions – unpublished data

Reviewer Comments & Author Rebuttals

Reviewer Reports on the Initial Version:

Referees' comments:

Referee #1:

Summary

The authors purified the LINE-1 (L1) ORF2-encoded protein (ORF2p). They next: (1) biochemically reconstituted the initial steps of target site primed reverse transcription (TPRT); and (2) report a low resolution (9-10 Å) cryo-electron microscopy structure of ORF2p recognizing a template RNA used to initiate cDNA synthesis. The authors then used this information, as well as alpha-fold ORF2p prediction models, to gain insight into how ORF2p recognizes the L1 RNA poly(A) tail and assess the function of unassigned and/or ill-defined conserved amino acids within ORF2p function in target site primed reverse transcription (TPRT).

Evaluation

Overall, the paper represents an important advance, as it provides structural insight for how L1 ORF2p participates in the initial steps of TPRT. However, it is somewhat disappointing that the manuscript focuses more on an EN-independent overhang-primed reverse transcriptase (OPRT) assay rather than the partially reconstituted TPRT assay. Moreover, although nicely extending previous analyses, the study can be criticized as only providing a limited amount of "new" biological insights into L1 biology. Below, I have included comments meant to help the authors improve their study.

Major Comments

Fig. 1: It would be useful to use EN-deficient and RT-deficient mutants in the TPRT assay depicted in Fig. 1c (e.g., such as the mutants used in Fig. 2e). There are many bands in Fig. 1c that require explanation; can the authors comment on how they characterized these products? Comparing WT, EN-, and RT- L1 ORF2p mutants in the TPRT assay should allow more careful characterization of products. In sum, thoroughly defining products in the RT and TPRT assays would be a welcome addition to the paper.

Fig. 1: Did the authors ever try to detect second-strand cleavage activity in their TPRT assay? Previous studies detected L1 EN activity and first strand cDNA synthesis (e.g., PMID: 12411507 and earlier references), which raises the question "What have we really learned in this paper about TPRT that significantly extends previous studies?".

Fig. 1: A question not fully resolved in this study is why the L1 EN activity is so weak on double strand DNA substrates (Fig. 1c). For example, as mentioned in the discussion, although the L1 EN target site

"consensus" sequence is 5'-TTTTT/AA-3', many L1 EN cleavage sites contain at least one C to T substitution (Flasch et al., 2019). Do T to C changes affect L1 EN cleavage?

Fig. 2: It is a bit disappointing that the authors examined most of their mutants in the OPRT rather than the TPRT assay (see Fig. 4g,h). For example, did the authors examine CTS and/or other mutants in the L1 TPRT assay? Comparing mutants in both the TPRT and OPRT assays would be a welcome addition to the paper. As is, these results can be criticized as simply extending previous analyses.

Fig. 2: Can the authors comment on whether the overhang primed reverse transcriptase (OPRT) reaction in Fig. 2e is similar in concept to previously reported examples of EN-independent L1 retrotransposition (see PMID: 12006980, PMID: 17344853, and PMID: 21940498). If they are similar (as they appear to be) they may wish to cite some of the above papers.

Figs 2 and 4: Can the authors state how they characterized the presumptive template jumping products? Can they examine whether mutations in the L1 RT (similar to one previously reported from R2Bm and group II introns) affect template jumping? Again, these analyses would speak to product characterization.

Line 247: I am not sure what the authors mean by "...shorter than expected poly(A) lengths...". In general, new (and de novo) L1 insertions have much longer poly(A) tails than genomic L1s. Please clarify the meaning of what is written.

Minor Comments

Title: I am not a fan of the term "retrotransposase." Why not stick with the accepted nomenclature and say "human LINE-1 ORF2p."

Line 45: In general, this paper is referenced in an impressive and scholarly manner. However, several groups (in addition to Taylor et al., 2013) demonstrated that cellular proteins interact with L1 RNPs. The authors may wish to cite the first reference on the topic (PMID: 23749060) and at a later date in (PMID: 25951186). The use of L1s containing "tags" to detect ORF1p, ORF2p, and L1 RNA originally were reported in (PMID: 20147320 and PMID: 20949108). Of course, I realize references are limited and it is up to the authors to choose if they want to cite any of the above studies in addition to Taylor et al. (2013).

Referee #2:

In this manuscript, Thawani et al. solved the structure of the ORF2p protein of human L1 non-LTR-retrotransposon by cryo-EM. This is a notable achievement as the full-length protein has been notoriously difficult to express in large quantities. The work is very timely as other related structures from *Bombyx mori* R2 retrotransposons have been recently released (Wilkinson et al., *Science*, 2023; Deng et al., *bioRxiv*, 2023) and L1 ORF2p domain organization and properties considerably differ from R2 and are representative of different clades of non-LTR-retrotransposons.

Obtaining the structure of L1 ORF2p is a significant advance for the field and beyond. Unfortunately, the study does not provide solid new information related to the mechanism of L1 replication, known as target-primed reverse transcription, nor does it explain the unique hallmarks of L1 compared to R2. Such features include, for instance, the preferential reverse transcription of its own RNA (cis-preference) or other species such as Alu RNAs, coordination between the initial cleavage and the subsequent reverse transcription steps, second-strand cleavage, etc. In addition, the structure is only minimally exploited to explain the large body of genetic and biochemical work achieved these last decades (mutagenesis experiments, RT substrate preference, etc). Finally, the large emphasis on the hairpin in the template RNA is exaggerated at this stage. Added to minimize heterogeneity in cryo-EM, its biological relevance has not been demonstrated and

is likely limited by the observation that long-used L1 reporter constructs (such as those described in Moran et al., Cell, 1996) do not contain the native L1 3' UTR, which was shown not to be required for retrotransposition.

Pros:

- The structure of full-length L1 ORF2p at high-resolution is novel and could help guiding future experiments.

Cons:

- The obtained structure has been minimally exploited and offers little insight into the mechanism of L1 replication or, more generally, into L1 biology.

- How template selection is achieved remains an unsolved problem, contrary to what is claimed.

- From a technical perspective, the biochemical assays are not sufficiently characterized.

Major issues:

1. RT assays: The RT assays are not well characterized and sufficiently described. Information related to primer and template of each assay is spread in the text/figure/Table S1, which makes it difficult to follow. It is not just a question of presentation. Taking Fig. 1b as an example (but other similar assays have similar issues throughout the manuscript), two types of templates are used according to Table S1: a 240-nt RNA (L1 3' UTR) and a 64-nt RNA (hairpin RNA). However, the cDNAs produced are much shorter: approximately 180 nt and 55 nt, respectively. For the long RNA template, a product labeled as template jump seems to exhibit the correct size of 240 nt. Moreover, in reactions obtained with a given template, faint bands corresponding to the size of the main product obtained with the other template are observed. Thus, the exact nature of the observed products is undetermined and needs to be more carefully characterized. From Table S1, it also seems that RT or TPRT assays with the template used for cryo-EM were not done or shown.

2. RT assays: Enzymatic activities should be recorded in conditions of initial velocity and quantified to provide meaningful comparisons.

3. Primer models: The authors do not link the choice of template and primers in their assay with previous work. Assays with primers formed by a hairpin and 3' overhang, identical to the so-called "overhang-primed reverse transcription" (OPRT) assay, have been previously developed and extensively used to show that a 4-nt TTTT overhang is suboptimal for priming (Monot et al., PLoS Genet., 2013). Comparing optimal and non-optimal sets of primers in the context of L1 ORF2p structure could reveal how L1 RT copes with the variety of cleaved substrates (produced by L1 EN) and ultimately contribute to integration site preference.

4. Template selection: The authors' claim that their "functional assays together with (their) structure provide insight into how 3' polyA tract is recognized and required for the mobilization of L1 and SINEs in vivo" is greatly exaggerated. First, the hypothesis that sequence-specificity for the poly(A) can explain template choice is hard to reconcile with the fact that all cellular mRNA species contain a poly(A) tail, too. If some level of specificity comes from specific contacts between the poly(A) tail and amino-acids outside of the RT domain as suggested, a structural comparison should have been made with non-poly(A) template, and more importantly with template:primer pairs that more accurately mimic L1 initiation of reverse transcription (here the structure somehow corresponds to an elongating enzyme with an A-rich template as the heteroduplex region of the template:primer substrate is not poly(A):oligo(dT) which would more accurately represents the initiation step).

5. Template selection: Moreover, the 22-nt poly(A) stretch used in the templates appears arbitrary and not consistent with the effect of p(A) length on retrotransposition of Alu sequences (which have a DNA encoded p(A) tract that can be experimentally controlled; see Dewannieux et al., *Genomics*, 2005).

6. RNA hairpin in the template: The significance of the RNA hairpin-related experiments as a whole is unclear, given that a role in L1 biology has not been demonstrated, and somehow contradicts the observation that the 3' UTR is not required for retrotransposition (Moran et al., *Cell*, 1996). A stem-loop in L1 3' UTR was computationally predicted in Grechishnikova et al. (*BMC Genomics*, 2016), but again never shown to play any role in retrotransposition. Sequencing products obtained with the different mutants and template:primer variants would provide stronger evidence to support a potential role on defining the site of initiation, and should be compared with retrotransposition assays in cell culture with or without such a hairpin.

7. CTS role: The proposed role of the insertion helix on processivity should be tested by proper processivity assays (using trapping nucleic acids). An additional concern related to Fig. 4g is that the primer hairpin appears significantly shortened in presence of the enzyme with the helix deletion. As the 3' overhang is necessary for priming, it is very likely that the absence of product is due to a terminal degradation of the primer and its inability to anneal to the template. Finally, the hypothesized role of the CTS on stabilizing or promoting primer:template heteroduplex could be tested directly using quantitative molecular interaction methods and CTS mutants, which would provide more strength to the predictions made by the authors.

Other points:

8. Does the so-called Zn-knuckle in the C-terminal region of ORF2p bind Zn?

9. L1 poly(A) tail is likely bound in vivo by PABPC1 (Dai et al., *MCB*, 2012). Can the authors determine (using AlphaFold models) whether such a binding would prevent reverse transcription or not, and/or might require prior displacement of PABPC1?

10. L. 66: TPRT was biochemically reconstituted by Cost et al. (*EMBO J.*, 2002)

11. Although there is no clear consensus on the naming of the different domains of ORF2p, the authors introduce new ones, such as AP-EN (a more common usage is EN or APE), CTS (instead of CTD), etc. Former studies have introduced different names for the EN linker, too (cryptic sequence and Z domain; cf Christian et al., *NAR*, 2016). At minima, I would suggest displaying these alternative names in Fig. 1a so that the reader can easily relate the present work to previous studies.

12. The term 'retrotransposase' is not commonly used in the field (a simple PubMed search returns only four articles). If the authors want to introduce a new name, they should indicate this more explicitly and provide a clear definition. Otherwise, they should stick to ORF2p.

13. L. 259: When referring to L1 target site, Jurka (*PNAS*, 1997) and Sultana et al. (*Mol. Cell*, 2019) should be cited, too. As for "primer pairing with the template", please cite Kulpa et al. (*NSMB*, 2006), Monot et al. (*PLoS Genet.*, 2013), and Kopera et al. (*PNAS*, 2011). It would be also worth mentioning and discussing how the structure of ORF2p may explain that terminal nucleotides can be mismatched but still elongated, unlike most DNA polymerases (these articles).

14. L. 268-269: It is unclear how "the structural and biochemical characterization in this work informs template design for productive trans assembly of RNPs from the L1 family, removing the requirement for co-translational RNP assembly", particularly for gene editing purpose, as assembly was done in vitro and any template:primer pair with sufficient base-pair complementarity appears to be successfully extended under these experimental conditions.

Referee #3:

Thawani et al. have solved the first exciting structure of a LINE-1 retroelement complex that is responsible for comprising ~50% of the human genome. They have captured the LINE-1 RT in the process of cDNA synthesis using an RNA template. They show that the polyA tail of the RNA template is recognized by accessory domains outside of the RT core. In addition, a C-terminal domain was found to be responsible for disrupting RNA secondary structure in the template to promote processivity. This work represents the first high-resolution structure of a retroelement to give insight into cDNA synthesis during TPRT in humans. This work is a major advancement for the field and provides an initial framework for repurposing these mobile genetic elements for genome editing.

The following major concerns/revisions should be first be addressed:

1. The authors used a bacterial group IIC intron retroelement as their model for a phylogenetic comparison with the LINE-1 RT. This is a suboptimal comparison as IIC RTs are only found in bacteria. The L1 RT is more directly related to the group IIB intron retroelements, which are found in both bacteria and eukaryotes. Therefore, it is more likely that the IIB lineage evolved into LINE-1 elements, rather than the IIC class, which hit a dead end in prokaryotes. In this regard, the first structure of group II intron retroelement inserting into DNA was solved by Haack et al. (2019) (ref. below) of the IIB class, which predates the group IIC intron retroelement structure solved by Chung et al. (2022).

D.B. Haack et al., Cryo-EM Structures of a Group II Intron Reverse Splicing into DNA. Cell 178, 612-623.e612 (2019).

2. Ext. Data Fig. 6: The authors developed an assay for testing the importance of the recognition of the polyA tract for initiating cDNA synthesis. They used a DNA duplex with a polyT overhang to anneal a polyA-containing RNA template with the overhang providing the 3'-OH to prime cDNA synthesis. They show that mutating the polyA sequence into a GC-rich sequence in the RNA template does not result in cDNA synthesis. What is concerning is that the DNA overhang region is also simultaneously changed to a complementary GC-rich sequence, yet no evidence was provided to show that these regions anneal with each other. The GC-rich sequences in both the RNA template and DNA overhang could potentially form secondary structures that preclude annealing of the two regions. This is especially a concern since no high temperature annealing step was performed. This is puzzling since this is typically done for in vitro annealing of nucleic acids. We propose the following experiments to verify this central conclusion:

A. A control experiment should be performed using a native gel assay to confirm that this expected annealing is actually happening without a high temperature annealing step. A native gel would show a mobility shift upon formation of the complex. In addition, the positive control reaction involving a primer DNA annealed to an unstructured RNA did involve a high temperature annealing step (75° C) and was successful for cDNA synthesis. Since this is the main biochemical assay to verify a central finding of the paper, it is important to repeat these experiments with an annealing step included for the experimental samples as well.

B. Additionally, to show that the lack of cDNA synthesis in the absence of the polyT sequence is due to specific recognition of the L1 RT and not due to lack of RNA/DNA annealing, a commercially available RT such as Superscript III should be used as a positive control. If cDNA synthesis is observed using Superscript III with the GC-rich sample, then this would be sufficient to support the claim that annealing is occurring and that polyA recognition is important for L1 RT functionality.

3. Many of the major conclusions/experiments regarding the L1 RT need to include a control RT such as

Superscript III throughout the manuscript. It is difficult to assess the validity of the assays without a control RT. For example, it is possible that many of the properties observed for the L1 RT would also be observed with any general RT with unusual primer/template combinations. This is also true for Ext. Data Fig. 7a. In this assay, the length of the polyA tail on the RNA template was varied. Shortening this polyA tail will bring the structured GC-rich RNA hairpin closer to the RT active site and likely result in steric hindrance resulting in a lack of RT activity. However, this could happen with any general RT for cDNA synthesis. Therefore, a positive control RT would provide evidence for the assertion that this only occurs for the L1 RT.

4. To my knowledge, the C-terminus segment (CTS) that separates the secondary structure in the RNA template is the first description of such a domain in an RT. It is possible that a similar domain may exist in all RTs as their function generally requires them to disrupt secondary structure of the RNA template to be threaded through the active site. It may not have been seen before because no one determined the structure of an RT actively involved in cDNA synthesis in the presence of a highly structured RNA template. The presence of a homologous domain in other RTs should be verified through a bioinformatics search. It would be great if the authors could comment on this.

Minor revisions:

1. p. 5, line 137: This sentence refers to residue 468 as an arginine; however it is shown in Ext. Data Fig. 5b as "K468", which is a lysine.

2. p. 6, line 190: This sentence states that Ile1121 forms a hydrophobic with G1 of the RNA hairpin structure. However, Fig. 4d depicts the side chain of Ile1121 pointing away from G1. Therefore, it does not seem to be forming a hydrophobic interaction with the nucleobase of G1.

Author Rebuttals to Initial Comments:

Response to Reviewers' Comments

We appreciated the thoughtful comments of the reviewers on the original version of this manuscript. Although all reviewers recognized our structure as a major advance, they expressed that the manuscript did not go far enough in providing new insights into the mechanism of TPRT by LINE-

1. We took this feedback seriously and have made dramatic improvements in our biochemical reconstitutions as we gained unexpected new insights about L1ORF2 TPRT mechanism. Below we first summarize the key improvements and additions in the revised manuscript, then address each set of comments in detail. We thank you in advance for re-examining the manuscript.

In human cells, the majority of L1ORF2 TPRT events insert new Alu repeats into the genome as the 300 bp Alu repeats constitute about the same fraction of the genome as L1 (~10%). Therefore, to reconstitute biochemically robust TPRT, we used a native template of L1ORF2, the Alu RNA, for both our assays and to produce a new cryo-EM structure (see below). All new structure/function assays in this revised manuscript were done with this physiological L1ORF2 template, which we show supports more robust *in vitro* TPRT than the L1 3'UTR (Figure 1C). We benchmarked our improved TPRT assay using wild-type, EN-dead, and RT-dead versions of L1ORF2 protein in parallel. TPRT using Alu RNA yields complete cDNA synthesis products, so the TPRT product profiles in the revised manuscript lead to clearer interpretation. As these data supersede the previous overhang-primed reverse transcription (OPRT), we have replaced all OPRT assays with *bona fide* TPRT assays, which gave us more information about the cleavage and TRPT activities of L1ORF2.

New to the revised manuscript, we used TPRT assays to deepen the understanding of L1ORF2's target-site selection. The results were illuminating, so we combined the previous figures into revised Figures 1-3 and dedicated Figure 4 to these new findings. We show that target-site DNA strand architecture is

a major determinant of EN cleavage efficiency. Our model emerged from close examination of our L1ORF2 RNP structure. Modeling an EN-domain-bound target-site duplex with ≥ 10 bp 5' of the nick leads to a steric clash with non-EN domains of L1ORF2. TPRT activity increased substantially when the consensus target site had 9 bp or less upstream DNA duplex, and it skyrocketed when we added an upstream 5' single-stranded DNA overhang. This biochemically determined optimal target site for purified L1ORF2 RNP is abundant in cells as the lagging strand behind a DNA replication fork, where L1 actually prefers to insert (Flasch et al. 2019). Finally, we solved a new cryo-EM structure of L1ORF2 bound to an Alu RNA (a transposition-competent Alu repeat unit) at 4.4 Å resolution (Figure 1E). This map compares remarkably well to our original structure with the synthetic template RNA at 3.2 Å resolution, including the positioning of the Alu RNA stem-loop and polyA tail (compare Figure 1E and 1F). This comparison validates the design of the synthetic template RNA that gave us the high-resolution structure. Together, our results reveal that direct recognition by L1ORF2 of both RNA structure and sequence, as well as recognition of a specific target DNA architecture, play important roles determining the landscape of new L1-mediated insertions

The authors purified the LINE-1 (L1) ORF2-encoded protein (ORF2p). They next: (1) biochemically reconstituted the initial steps of target site primed reverse transcription (TPRT); and (2) report a low resolution (9-10 Å) cryo-electron microscopy structure of ORF2p recognizing a template RNA used to initiate cDNA synthesis. The authors then used this information, as well as alpha-fold ORF2p prediction models, to gain insight into how ORF2p recognizes the L1 RNA poly(A) tail and assess the function of unassigned and/or ill-defined conserved amino acids within ORF2p function in target site primed reverse transcription (TPRT).

Overall, the paper represents an important advance, as it provides structural insight for how L1 ORF2p participates in the initial steps of TPRT. However, it is somewhat disappointing that the manuscript focuses more on an EN-independent overhang-primed reverse transcriptase (OPRT) assay rather than the partially reconstituted TPRT assay. Moreover, although nicely extending previous analyses, the study can be criticized as only providing a limited amount of “new” biological insights into L1 biology. Below, I have included comments meant to help the authors improve their study.

We thank the reviewer for recognizing the breakthrough in our work and providing critical feedback that helped us as we delved deeper in our biochemical assays to gain more insights about L1ORF2 TPRT mechanism. We substituted all OPRT assays with robust TPRT assays, gaining information about target-site cleavage as well as TPRT. Building on our original submission, the revised manuscript better demonstrates how L1ORF2 recognizes RNA template and discovers the long- missing DNA target site *architecture* that supports robust cleavage and TPRT.

We also want to clarify the resolution estimates of our cryo-EM data. Our original cryo-EM map of L1ORF2 engaged in cDNA synthesis on a synthetic template RNA has a resolution of 3.2 Å that allows us to visualize the amino acid side chains (Figure 1F). The revised manuscript includes a second cryo-EM structure of L1ORF2 engaged with a physiological Alu RNA template at 4.4 Å resolution (Figure 1E). We have added this information to the figure panels for clarity. The AlphaFold model was used as a starting point for model building, a standard practice in the structural biology, but the final refined coordinates we report are markedly different from these initial coordinates.

Fig. 1: It would be useful to use EN-deficient and RT-deficient mutants in the TPRT assay depicted in Fig. 1c (e.g., such as the mutants used in Fig. 2e). There are many bands in Fig. 1c that require explanation; can the authors comment on how they characterized these products? Comparing WT, EN-, and RT- L1 ORF2p mutants in the TPRT assay should allow more careful characterization of products. In sum, thoroughly defining products in the RT and TPRT assays would be a welcome addition to the paper.

We appreciate this request and are excited to present a greatly expanded biochemical characterization of TPRT by L1ORF2 in the revised manuscript. Revised Figure 1D compares wild-type, EN-dead, and RT-dead proteins for their cleavage and TPRT activities. Gel bands below the target site appear in reactions of wild-type or RT-dead protein but not EN-dead protein, and therefore are products of on-target or off-target cleavage. Our target site optimization drastically improved on-target cleavage, making the interpretation of the product profile more straightforward now. For sub-optimal target sites, off-target cleavage positions are mapped to DNA sequence in Extended Data Figure 9. Gel bands above the target site represent cDNA product from the TPRT reaction, which in the revised

manuscript are easier to interpret due to the use of an Alu template RNA that is efficiently copied into full-length cDNA. Products longer than one complete template copying likely arise by template jumping, but because we did not investigate these products further, we removed the “template jump” label for long products.

Fig. 1: Did the authors ever try to detect second-strand cleavage activity in their TPRT assay? Previous studies detected L1 EN activity and first strand cDNA synthesis (e.g., PMID: 12411507 and earlier references), which raises the question “What have we really learned in this paper about TPRT that significantly extends previous studies?”

This is an important point, where we would like to emphasize the major technical limitations of the previous proof-of-principle TPRT by L1ORF2 (Cost et al., EMBO J 2002) and several conflicting results reported. Weak EN and RT activities were observed using a plasmid DNA substrate, and the low signal necessitated that the authors use PCR amplification as a read-out of the desired TPRT products. From their analysis, the authors reported several findings that have been contradicted by the biological requirements for L1 insertion, including (1) similar TPRT activity of wild-type and EN-dead L1ORF2 (Fig. 2C of Cost et al.), (2) no TPRT requirement for RNA polyA tail (Fig. 4B of Cost et al.), and (3) EN nicking activity observed only upon proteolytic removal of L1 EN domain from the remainder of protein (Fig. 5 of Cost et al.). These reports are in contradiction with the known biological requirements for L1 insertion, including the demonstrated role of EN activity and of the RNA template polyA tail (Moran et al., Cell 1996; Doucet et al., Mol Cell 2015). In the revised manuscript, we have emphasized that our L1ORF2 assays give new insights about TPRT mechanism that are clearly not simple replicates of previous findings. Our data highlight the role of EN activity (Figures 1 and 4) and the template RNA polyA tail (Figure 2), and they provide an unexpected biochemical mechanism that can account for L1ORF2 TPRT preference in cells for the lagging strand template behind a replication fork (Flasch et al., Cell 2019) (Figure 4).

We appreciate the experiment recommendation and have indeed assayed for it but did not observe a distinct second-strand cleavage or second-strand synthesis product *in vitro* (see Reviewer Figure 1). Our new insights about L1ORF2 target site preference suggests that the second strand cleavage may not occur or be necessary for full gene integration (Figure 4). Our structure and added biochemical assays (Figure 4) show that first-strand cleavage occurs on a DNA duplex near a single-strand/double-strand DNA junction. Thus, nicking effectively produces a staggered double-stranded break, removing the requirement of second-strand nicking by L1ORF2 and allowing a higherfidelity host polymerase to synthesize the second strand.

Fig. 1: A question not fully resolved in this study is why the L1 EN activity is so weak on double strand DNA substrates (Fig. 1c). For example, as mentioned in the discussion, although the L1 EN target site “consensus” sequence is 5'-TTTTT/AA-3', many L1 EN cleavage sites contain at least one C to T substitution (Flasch et al., 2019). Do T to C changes affect L1 EN cleavage?

This is an excellent question and one that puzzled us too, as did the fact that the relatively abundant off-target cleavage products would be unproductive for TPRT. Our revised TPRT assays demonstrate that target DNA architecture, not just sequence, is a major determinant of EN cleavage efficiency. This hypothesis emerged from closer analysis of the L1ORF2 RNP structure. In the revised manuscript, we combined the previous Figures into revised Figures 1-3 to dedicate Figure 4 to this new ground. Briefly, for efficient EN cleavage and TPRT, the nick site needs to be positioned less than 9bp from the 5' end of the DNA duplex with an upstream single-stranded DNA overhang (Figure 4B-E). Otherwise, target site duplex binding would lead to a steric clash with the non-EN domains of

full-length L1ORF2 (Figure 4A). Therefore, the target sites utilized in our original manuscript and in previous work, which were annealed DNA duplexes or plasmids, would have had compromised binding affinity for the EN domain due to a steric clash with non-EN domains. The steric clash accounts for the apparently low EN activity of full-length L1ORF2 versus the EN domain proteolytically removed from the full-length protein (Cost et al., EMBO J 2002).

We find that the T to C substitution(s) in the EN cleavage site that the reviewer notes to be tolerated in cells (Flasch et al. 2019) are also tolerated in DNA substrates for *in vitro* TPRT (see Reviewer Figure 2). Although EN activity is not compromised, TPRT activity is reduced, likely from the compromised base-pairing of primer and RNA template.

Fig. 2: It is a bit disappointing that the authors examined most of their mutants in the OPRT rather than the TPRT assay (see Fig. 4g,h). For example, did the authors examine CTS and/or other mutants in the L1 TPRT assay? Comparing mutants in both the TPRT and OPRT assays would be a welcome addition to the paper. As is, these results can be criticized as simply extending previous analyses.

We thank the reviewer for this prompt, which made us take the extra time and effort to purify and characterize the mutant L1ORF2 proteins using our improved TPRT assay in parallel to a standard RT primer-extension assay with annealed primer-template (Figures 1D, 2G, 3F and Extended Data Figure 2B-C). TPRT activity of the Δ Insertion helix and Δ CTS mutants was considerably lower than the wild-type enzyme (Figure 3F), with the Δ CTS mutant notably displaying particularly low EN cleavage activity (Figure 3F). However, both of these proteins showed robust RT activity in a primer-extension assay, equal or exceeding the RT activity of wild-type L1ORF2 (Extended Data Figure 2B- C). These findings support a role for the CTS in TPRT-specific coordination of nucleic acid interactions with L1ORF2.

Fig. 2: Can the authors comment on whether the overhang primed reverse transcriptase (OPRT) reaction in Fig. 2e is similar in concept to previously reported examples of EN-independent L1 retrotransposition (see PMID: 12006980, PMID: 17344853, and PMID: 21940498). If they are similar (as they appear to be) they may wish to cite some of the above papers.

We thank the reviewer for this question and citation suggestions. Our OPRT assays were superseded by the new, high efficiency TPRT assays, with the latter providing more meaningful data. As a result, we have only included the new TPRT data and no longer discuss the OPRT assay in the manuscript.

Figs 2 and 4: Can the authors state how they characterized the presumptive template jumping products? Can they examine whether mutations in the L1 RT (similar to one previously reported from R2Bm and group II introns) affect template jumping? Again, these analyses would speak to product characterization.

We appreciate this question. In earlier studies not linked to this work, we screened for mutations that would diminish the template jumping activity of the R2 non-LTR retroelement protein, which has an active-site architecture very similar to L1ORF2. We could not achieve a clean separation-of-function and instead, observed a diminished overall RT activity when the template jumping activity was mutated, unlike the studies of Group II intron RT proteins. As a result, we could not produce L1ORF2 mutations that reduce the template jumping without affecting also the overall TPRT activity. Further, in the revised work the template jumping products are much clearer due to robust full-length TPRT cDNA synthesis on the Alu template RNA. Products longer than one cDNA appear in incremental units of template length, as discerned from the new DNA size ladder. This pattern matches the one

expected for products arising from template jumping. Yet, because template jumping as a mechanism is not the focus of our work, and as we did not sequence these products from L1ORF2 TPRT reactions, we have removed the “template jump” label for long products.

Line 247: I am not sure what the authors mean by “...shorter than expected poly(A) lengths...”. In general, new (and de novo) L1 insertions have much longer poly(A) tails than genomic L1s. Please clarify the meaning of what is written.

We have significantly modified the section describing these results (Figure 2C) and have removed this statement from the revised manuscript.

Minor Comments

Title: I am not a fan of the term “retrotransposase.” Why not stick with the accepted nomenclature and say “human LINE-1 ORF2p.”

We agree and have now adopted the accepted nomenclature and removed all instances of the word “retrotransposase” from the revised manuscript.

Line 45: In general, this paper is referenced in an impressive and scholarly manner. However, several groups (in addition to Taylor et al., 2013) demonstrated that cellular proteins interact with L1 RNPs. The authors may wish to cite the first reference on the topic (PMID: 23749060) and at a later date in (PMID: 25951186). The use of L1s containing “tags” to detect ORF1p, ORF2p, and L1 RNA originally were reported in (PMID: 20147320 and PMID: 20949108). Of course, I realize references are limited and it is up to the authors to choose if they want to cite any of the above studies in addition to Taylor et al. (2013).

We thank the reviewer for pointing out these L1ORF1-associated co-factor interaction papers to cite. We have now cited Goodier et al., 2013 and Moldovan et al., 2015 in addition to Taylor et al., 2013 in the appropriate sentence about co-factors associated with L1 RNPs.

In this manuscript, Thawani et al. solved the structure of the ORF2p protein of human L1 non-LTR-retrotransposon by cryo-EM. This is a notable achievement as the full-length protein has been notoriously difficult to express in large quantities. The work is very timely as other related structures from Bombyx mori R2 retrotransposons have been recently released (Wilkinson et al., Science, 2023; Deng et al., bioRxiv, 2023) and L1 ORF2p domain organization and properties considerably differ from R2 and are representative of different clades of non-LTR-retrotransposons.

Obtaining the structure of L1 ORF2p is a significant advance for the field and beyond. Unfortunately, the study does not provide solid new information related to the mechanism of L1 replication, known as target-primed reverse transcription, nor does it explain the unique hallmarks of L1 compared to R2. Such features include, for instance, the preferential reverse transcription of its own RNA (cis-preference) or other species such as Alu RNAs, coordination between the initial cleavage and the subsequent reverse transcription steps, second-strand cleavage, etc. In addition, the structure is only minimally exploited to explain the large body of genetic and biochemical work achieved these last decades (mutagenesis experiments, RT substrate preference, etc). Finally, the large emphasis on the hairpin in the template RNA is exaggerated at this stage. Added to minimize heterogeneity in cryo-EM, its biological relevance has not been demonstrated and is likely limited by the observation that long-used L1 reporter constructs (such as those described in Moran et al., Cell, 1996) do not contain the native L1 3' UTR, which was shown not to be required for retrotransposition.

Pros:

- The structure of full-length L1 ORF2p at high-resolution is novel and could help guiding future experiments.*

Cons:

- The obtained structure has been minimally exploited and offers little insight into the mechanism of L1 replication or, more generally, into L1 biology.*
- How template selection is achieved remains an unsolved problem, contrary to what is claimed.*
- From a technical perspective, the biochemical assays are not sufficiently characterized.*

We thank the reviewer for seeing the potential of our work and highlighting the L1 replication features that remain to be understood, prompting us to improve and expand our experiments. Below we discuss the overall points in the context of our work before addressing specific points in subsequent comments.

1. Cis-preference: While *cis*-preference is one key feature of the L1 replication cycle, our work focused on understanding the principles of template RNA recognition in *trans* (an important *competing* interaction). It is generally acknowledged that far more L1ORF2 genome insertion events are from L1ORF2 use of Alu RNA rather than L1 RNA as template (each is ~10% of the genome, but Alu elements are on the order of 300 bp, so their copy number in the genome is higher than L1 copy number), making Alu RNA a preferred substrate of L1ORF2. The revised manuscript includes a cryo-EM structure of L1ORF2 bound to an Alu RNA and uses an Alu RNA as the *in vitro* TRPT template throughout. We would also like to note that investigating template selection from a *cis*-preference perspective is unfeasible with structural methods, as it would require solving a structure of ribosome engaged, partially translated L1ORF2 protein bound to the L1 mRNA polyA tail.

Nevertheless, we believe that our revised manuscript offers new information to those interested in *cis*-preference. First, we show that L1ORF2 dramatically prefers an Alu RNA over L1 3'UTR RNA as template for TPRT (Figure 1C, compare product in lanes AY, AJ, AJh and L1), indicating that the binding and positioning of the L1 3'UTR by L1ORF2 may be sub-optimal. Second, we show that L1ORF2 often halts cDNA synthesis *in vitro* at a G-quadruplex-forming region within the L1 3'UTR RNA, as assessed by incomplete cDNA synthesis if this region of template is its presence but not in its absence (Figure 1C, lanes L1 and L1Δ). This result suggests that RNA remodeling enzymes could strongly influence the percentage of L1 insertions that are full-length.

2. Validation of the structure including the stem-loop: When considering copy number, Alu RNA seems to be the preferred endogenous template for L1ORF2. To validate our results with this native template for L1ORF2, we solved a new cryo-EM structure of L1ORF2 bound to an Alu RNA repeat unit with polyA:polyT duplex at 4.4 Å resolution (Figure 1E). This map compares remarkably well to our structure with a synthetic template RNA at 3.2 Å resolution, including an identical mode of Alu RNA stem-loop engagement (compare Figure 1E and 1F). As the Alu RNA stem structure is well established for its importance in *in vivo* transposition (Dewannieux et al., Nat Gen 2003; Ahl et al., Mol Cell 2015), these data validate the design of the synthetic RNA template. We also note that the use of synthetic nucleic acids is a common practice for reducing conformational heterogeneity to achieve the highest resolution structure.

To better examine the role of a template RNA stem-loop using biochemical assays, we designed Alu-derived RNA templates with increased stem-loop formation (removing mismatches, AJhm), or reduced stem-loop formation (introducing more mismatches, AJh-uf), compared to wild-type (AJh) (Figure 3F). The TPRT product profiles suggest that the template RNA stem-loop specifies a more homogenous position of cDNA synthesis initiation. This “ruler” function likely ensures that full-length Alu is transposed.

Finally, we revised the manuscript to place more focus on L1ORF2 nucleic acid interactions beyond than with the template RNA hairpin.

3. Improved biochemical analyses and nicking-RT coordination: We are pleased to have greatly improved and expanded the biochemical analysis of L1ORF2. In brief, after discovering unexpected target-site structural features that drastically improve EN and TPRT activities of L1ORF2, we replaced all OPRT assays of the original manuscript with improved TPRT assays, thereby eliminating concerns about interpretation of the OPRT assay. Further, we added more careful and quantified analysis of L1ORF2's RNA template preference (Fig. 1C, 1D, 2C, 2G, 3D, 3F).

Our revised manuscript describes new work to characterize target site specificity. By modeling a previous structure of DNA duplex engaged with the EN domain onto our L1ORF2 RNP structure, we predicted that the EN cleavage site (TTTTT/AA) could have only a limited number of DNA base-pairs on its upstream side (5' of TTTTT/AA) to avoid a steric clash with non-EN domains of L1ORF2 (Figure 4A). When we used a target site with the edge of DNA duplex only 7 bp away (5' of a consensus cleavage motif) and included an upstream single-stranded DNA overhang, target site cleavage and TPRT activity skyrocketed (Figure 4E). These findings contribute biochemical understanding of a mechanism for the L1 preference for insertions to the lagging strand template behind a replication fork (Flasch et al., Cell 2019). To accommodate these findings within the revised manuscript, we combined previous Figures 1-4 into revised Figures 1-3 and added a new Figure 4. The unexpected L1ORF2 target-site structure is very distinct from target site recognition by the R2 retroelement protein, which binds an intact DNA duplex with sequence-specific DNA binding domains (Wilkinson et al., Science 2023).

Finally, we would like to discuss how our work compares with previous genetic and biochemical studies. Previous TPRT proof-of-principle for L1ORF2 (Cost et al., EMBO J 2002) was challenging due to weak EN and RT activities with a plasmid DNA substrate, and the low signal necessitated that the authors use PCR amplification as a read-out of the desired TPRT products. From their non-quantitative analysis, the authors reported several findings that have been contradicted by the biological requirements for L1 insertion, including (1) similar TPRT activity of wild-type and EN-dead L1ORF2 (Fig. 2C of Cost et al.), (2) no TPRT requirement for RNA polyA tail (Fig. 4B of Cost et al.), and (3) EN nicking activity observed only upon proteolytic removal of L1 EN domain from the remainder of protein (Fig. 5 of Cost et al.). In contrast, biological requirements for L1 insertion include a demonstrated role for EN activity and of the RNA template polyA tail (Moran et al., Cell 1996; Doucet et al., Mol Cell 2015). Our data highlight the role of EN activity (Figures 1 and 4) and the template RNA polyA tail (Figure 2). In the manuscript, we have refrained from extensively pointing out contradictions with the prior TPRT study, and instead chose to highlight the novelty of our findings.

4. **Second-strand cleavage:** With our new insights about target-site features, we believe that a second strand cleavage is not necessary for L1 insertion in cells (Figure 4F). The target-site architecture implies that one nick will result in a staggered end DNA break, making second strand cleavage not necessary. In agreement with this expectation, we do not detect second strand cleavage or second strand synthesis activity for L1ORF2 *in vitro*, even when first strand cleavage and TPRT occur efficiently (see Reviewer Figure 1).

Major issues:

1. *RT assays: The RT assays are not well characterized and sufficiently described. Information related to primer and template of each assay is spread in the text/figure/Table S1, which makes it difficult to follow. It is not just a question of presentation. Taking Fig. 1b as an example (but other similar assays have similar issues throughout the manuscript), two types of templates are used according to Table S1: a 240-nt RNA (L1 3' UTR) and a 64-nt RNA (hairpin RNA). However, the cDNAs produced are much shorter: approximately 180 nt and 55 nt, respectively. For the long RNA template, a product labeled as template jump seems to exhibit the correct size of 240 nt. Moreover, in reactions obtained with a given template, faint bands corresponding to the size of the main product obtained with the other template are observed. Thus, the exact nature of the observed products is undetermined and needs to be more carefully characterized. From Table S1, it also seems that RT or TPRT assays with the template used for cryo-EM were not done or shown.*

We understand the concerns raised by the reviewer and have thoroughly addressed them in the revised manuscript. The revised manuscript replaces all OPRT assays with improved TPRT assays. The improved TPRT reaction we developed resulted from our analysis of the cryo-EM structure, which allowed us to predict target site DNA features that would optimize cleavage efficiency. The only RT assays included in the revised manuscript, shown in Extended Data Figure 2B-C, use primer extension after annealing a stable primer-template duplex.

Regarding template RNAs, we changed the default template in the revised manuscript to be from an Alu RNA, a biological L1ORF2 template. Template sequence variants were made in this background. In the revised manuscript, we used a closely spaced DNA ladder in gels that clearly resolve TPRT products so that we could verify that cDNA products run at the expected lengths. We have further denoted the input RNA length in figure legends in addition to Table S1 for a clearer presentation of our data. In the one figure panel where cDNA synthesis products do not entirely match

expectation (Figure 1C), the full-length and incomplete synthesis products are labeled explicitly and described in the text and figure legend. Using L1 3'UTR template (231 nt), incomplete product synthesis correlated with a halt at the start of a G-quadruplex-forming region. Therefore, we assayed a version of L1 3'UTR lacking the G-quadruplex-forming region; this 149 nt template gave only full-length cDNA product, matching the product migration of halted cDNA synthesis on the longer 3'UTR template (see Figure 1C right-most two lanes). From this assay, we concluded that L1 3'UTR may be a sub-optimal template RNA for reconstituted TPRT assays *in vitro*, in comparison to the half Alu RNA competent for genome insertion (141 nt) that gave much more TPRT product (Figure 1C). The latter template RNA, or variants of it, were used for all subsequent experiments. The revised manuscript also describes an added structure of L1ORF2 bound to this RNA template (Figure 1E).

2. RT assays: Enzymatic activities should be recorded in conditions of initial velocity and quantified to provide meaningful comparisons.

We have included an assay that measures RT activity by extension of a template-annealed primer over a time course. In the revised manuscript (Extended Data Figure 2B-C), we compared the amount of product synthesized over time by L1ORF2 wild-type and all its mutants side-by-side.

3. Primer models: The authors do not link the choice of template and primers in their assay with previous work. Assays with primers formed by a hairpin and 3' overhang, identical to the so-called "overhang-primed reverse transcription" (OPRT) assay, have been previously developed and extensively used to show that a 4-nt TTTT overhang is suboptimal for priming (Monot et al., PLoS Genet., 2013). Comparing optimal and non-optimal sets of primers in the context of L1 ORF2p structure could reveal how L1 RT copes with the variety of cleaved substrates (produced by L1 EN) and ultimately contribute to integration site preference.

As the improved TPRT assays superseded and replaced our original OPRT assays, we have omitted the discussion of OPRT assays in the revised manuscript.

4. Template selection: The authors' claim that their "functional assays together with (their) structure provide insight into how 3' polyA tract is recognized and required for the mobilization of L1 and SINEs in vivo" is greatly exaggerated. First, the hypothesis that sequence-specificity for the poly(A) can explain template choice is hard to reconcile with the fact that all cellular mRNA species contain a poly(A) tail, too. If some level of specificity comes from specific contacts between the poly(A) tail and amino-acids outside of the RT domain as suggested, a structural comparison should have been made with non-poly(A) template, and more importantly with template:primer pairs that more accurately mimic L1 initiation of reverse transcription (here the structure somehow corresponds to an elongating enzyme with an A-rich template as the heteroduplex region of the template:primer substrate is not poly(A):oligo(dT) which would more accurately represents the initiation step).

We thank the reviewer for raising this important point. First, we note that the entire length of the single-stranded RNA interacting with L1ORF2 is 100% adenosine. Non-A nucleotides were only used in the region of primer-template duplex to ensure a unique register of cDNA synthesis, where no sequence-specific interactions are believed to occur for L1ORF2 or any related RTs (Stamos et al., Mol Cell 2017). Second, to address the reviewer's concern, in revised manuscript Figure 2C we compare L1ORF2's TPRT using the Alu template RNA with 25A tail versus a version with the 5' 20A replaced with a scrambled mix of nucleotides (20N5A) that allows the terminal 5A to base-pair with primer 5T, or a version with the entire 25A tail replaced with a scrambled mix of nucleotides

(25N). The results show that the primer-pairing 5A are not sufficient to specify a good template RNA, so the polyA stretch that contacts L1ORF2 is critical. Third, to emphasize the biological relevance of the structure, we solved a new cryo-EM structure at 4.4 Å resolution of L1ORF2 with Alu RNA as the bound template (Figure 1E). In that case we utilized a terminal polyA:oligo(dT) primer duplex to address the reviewer's comment, with only two non-A RNA nucleotides at the template 3' end to enable precise annealing of the primer. We observed the same structural features and the same protein-RNA interaction interfaces as we see for L1ORF2 with the synthetic template RNA (compare Figure 1E and 1F), further validating our structural findings. Finally, we emphasize that the polyA sequence specificity we identified is not incompatible with the L1ORF2's *cis*-preference model that the reviewer notes. Instead, several works report that the polyA tail is essential for mobility of L1 RNA (Doucet et al., Mol Cell 2015) and Alu RNAs (Dewannieux et al., Nat Gen 2003; Dewannieux et al., Genomics 2005). Our structure and biochemical reconstitutions identify the molecular basis of this specificity. In summary, the structures and biochemical assays in this work do provide insight into how a template RNA 3' polyA tract is recognized: it is splayed out by RNA-protein interaction across an extensive L1ORF2 protein surface with adenine-specific contacts, with a necessary length of polyA characterized with more precision than in the original manuscript (revised Figure 2C).

5. Template selection: Moreover, the 22-nt poly(A) stretch used in the templates appears arbitrary and not consistent with the effect of p(A) length on retrotransposition of Alu sequences (which have a DNA encoded p(A) tract that can be experimentally controlled; see Dewannieux et al., Genomics, 2005).

We appreciated this comment and have examined more extensively the polyA tract length dependence of TPRT by purified L1ORF2 in the revised manuscript (Figure 2C), including RNAs with A-tract length comparable with L1-mediated Alu transgene insertion assayed in cells (Dewannieux et al., Genomics 2005). We observed that TPRT activity was high and approximately similar when using template RNAs with 75A, 50A, or 25A, but not 15A, 10A or 5A. Because the template with 25A gave high activity and a homogeneous product length, indicating a preferred position of TPRT initiation, we used this template RNA as default for TPRT reactions throughout the manuscript.

6. RNA hairpin in the template: The significance of the RNA hairpin-related experiments as a whole is unclear, given that a role in L1 biology has not been demonstrated, and somehow contradicts the observation that the 3' UTR is not required for retrotransposition (Moran et al., Cell, 1996). A stem-loop in L1 3' UTR was computationally predicted in Grechishnikova et al. (BMC Genomics, 2016), but again never shown to play any role in retrotransposition. Sequencing products obtained with the different mutants and template:primer variants would provide stronger evidence to support a potential role on defining the site of initiation, and should be compared with retrotransposition assays in cell culture with or without such a hairpin.

We understand the reviewer's concern and have considered and addressed it as follows. First, L1ORF2's native Alu RNA template is well-established to have RNA stem-loop structure 5' of its polyA tracts and this stem-loop fold is essential for Alu transposition (Weichenrieder et al., Nature 2000; Ahl et al., Mol Cell 2015; Dewannieux et al., Nat Gen 2003). Second, in our new cryo-EM structure of L1ORF2:Alu RNA complex, we observe the Alu stem-loop engaged with L1ORF2 in a very similar configuration as we see L1ORF2 engaging our synthetic RNA template (compare Figure 1E and 1F). Third, to provide biochemical evidence for the role of template RNA structure in TPRT initiation, we added the experiments shown in Figure 3F. In Figure 3F, we compare TPRT using Alu

template RNAs with the native hairpin (AJh), a “more perfect” version of the hairpin (AJhm), and a “more disrupted” version with fewer base-pairs (AJh-uf). All three template RNAs were used by L1ORF2 for TPRT, and although the template with reduced stem structure had very modest defect for TPRT efficiency, it had a major defect in precision of the site of TPRT initiation (Figure 3F). Finally, while we are interested in developing *in vivo* TPRT assays to explicitly compare retrotransposition frequency of L1ORF2 RNP protein and RNA variants, in both *cis* and *trans* RNP assembly contexts, these assays are beyond the scope of our current work.

7. *CTS role: The proposed role of the insertion helix on processivity should be tested by proper processivity assays (using trapping nucleic acids). An additional concern related to Fig. 4g is that the primer hairpin appears significantly shortened in presence of the enzyme with the helix deletion. As the 3' overhang is necessary for priming, it is very likely that the absence of product is due to a terminal degradation of the primer and its inability to anneal to the template. Finally, the hypothesized role of the CTS on stabilizing or promoting primer:template heteroduplex could be tested directly using quantitative molecular interaction methods and CTS mutants, which would provide more strength to the predictions made by the authors.*

We thank the reviewer for these suggestions. To address these points, we first purified new preparations of L1ORF2 Δ Insertion helix mutant with additional purification steps to remove any contaminating protein that may have been cleaving DNA. Then we performed processivity assays using trapping nucleic acids with an experimental scheme adapted from assays on Group II intron RT by the lab of Anna Pyle (Zhao et al. RNA 2017). This data is presented in Reviewer Figure 3. We utilized long template RNAs, L1 5'UTR full-length (908 nt), or 5'-truncated (700 nt), annealed at their 3' end to ³²P-labelled 30 nt primer. Annealed primer-template duplexes were incubated with equimolar (100 nM) L1ORF2 protein (wild-type or Δ Insertion helix), or M-MLV retroviral RT, prior to addition of 4 μ M unlabeled competitor primer-template duplex to trap any dissociated RT. Both wild-type and mutant L1ORF2 generated predominantly full-length cDNA product, in contrast to M-MLV RT, consistent with the higher processivity of retroelement versus retroviral RTs. As we were not able to measure a difference in processivity between the wild-type and Δ Insertion helix mutant of L1ORF2, we have removed this speculative claim from the revised manuscript. Finally, we have reworded the statement that may have led to misinterpretation; our structure does not support any direct role of the CTS domain in stabilizing primer-template heteroduplex in the RT core active site.

Other points:

8. *Does the so-called Zn-knuckle in the C-terminal region of ORF2p bind Zn?*

We observe a classic zinc-knuckle fold in the C-terminal region of L1ORF2 that aligns well with published zinc-knuckle structures. However, we were not able to observe a clear density for the zinc atom where it would be expected. Since the zinc-knuckle is located at the periphery of our EM map, where the resolution is lower than the protein core, we are unable to definitively state if Zn binds the putative Zn-knuckle fold.

9. *L1 poly(A) tail is likely bound in vivo by PABPC1 (Dai et al., MCB, 2012). Can the authors determine (using AlphaFold models) whether such a binding would prevent reverse transcription or not, and/or might require prior displacement of PABPC1?*

The structural studies of PABPC1 bound to polyA indicate that an extensive amount (~30 nt) of polyA is bound and contorted by the PABPC1 protein (Sawazaki et al., Scientific Reports 2018). From our structure and biochemical assays, we infer that the length of polyA required by L1ORF2 is shorter than the footprint of one PABPC1 molecule. PABPC1's mode of engagement is not expected to be compatible with the polyA threading into the RT core of L1ORF2. We anticipate that displacement of PABPC1 from a segment of polyA would be necessary for polyA:L1ORF2 interaction.

10. L. 66: *TPRT was biochemically reconstituted by Cost et al. (EMBO J., 2002)*

We have cited this work and have further included wording to distinguish its accomplishments from those in our revised work.

11. *Although there is no clear consensus on the naming of the different domains of ORF2p, the authors introduce new ones, such as AP-EN (a more common usage is EN or APE), CTS (instead of CTD), etc. Former studies have introduced different names for the EN linker, too (cryptic sequence and Z domain; cf Christian et al., NAR, 2016). At minima, I would suggest displaying these alternative names in Fig. 1a so that the reader can easily relate the present work to previous studies.*

The revised manuscript uses the nomenclature “EN” to refer to the L1ORF2 endonuclease domain instead of our previous terminology “AP-EN”. Also, when EN linker and NTE are introduced in the text, we relate them previous motif designations. The revised manuscript better defines CTS as C-terminal segment (CTS) domain as more widely accepted for L1ORF2 (Adney et al., Genetics 2019).

12. *The term ‘retrotransposase’ is not commonly used in the field (a simple PubMed search returns only four articles). If the authors want to introduce a new name, they should indicate this more explicitly and provide a clear definition. Otherwise, they should stick to ORF2p.*

We removed usage of the word “retrotransposase” from the revised manuscript and instead adopted the well-accepted nomenclature.

13. *L. 259: When referring to L1 target site, Jurka (PNAS, 1997) and Sultana et al. (Mol. Cell, 2019) should be cited, too. As for “primer pairing with the template”, please cite Kulpa et al. (NSMB, 2006), Monot et al. (PLoS Genet., 2013), and Kopera et al. (PNAS, 2011). It would be also worth mentioning and discussing how the structure of ORF2p may explain that terminal nucleotides can be mismatched but still elongated, unlike most DNA polymerases (these articles).*

We thank the reviewer for these citation suggestions. We added the first two citations about L1 target site, but not the latter two citations. We do not have data to support a hypothesis for how the structure of L1ORF2 enables elongation of terminal primer-template mismatch. Because we cannot do justice to this topic, we have avoided speculating on this subject.

14. *L. 268-269: It is unclear how “the structural and biochemical characterization in this work informs template design for productive trans assembly of RNPs from the L1 family, removing the requirement for co-translational RNP assembly”, particularly for gene editing purpose, as assembly was done in vitro and any template:primer pair with sufficient base-pair complementarity appears to be successfully extended under these experimental conditions.*

We have removed this statement from the revised manuscript.

Thawani et al. have solved the first exciting structure of a LINE-1 retroelement complex that is responsible for comprising ~50% of the human genome. They have captured the LINE-1 RT in the process of cDNA synthesis using an RNA template. They show that the polyA tail of the RNA template is recognized by accessory domains outside of the RT core. In addition, a C-terminal domain was found to be responsible for disrupting RNA secondary structure in the template to promote processivity. This work represents the first high-resolution structure of a retroelement to give insight into cDNA synthesis during TPRT in humans. This work is a major advancement for the field and provides an initial framework for repurposing these mobile genetic elements for genome editing.

- 1. The authors used a bacterial group IIC intron retroelement as their model for a phylogenetic comparison with the LINE-1 RT. This is a suboptimal comparison as IIC RTs are only found in bacteria. The L1 RT is more directly related to the group IIB intron retroelements, which are found in both bacteria and eukaryotes. Therefore, it is more likely that the IIB lineage evolved into LINE-1 elements, rather than the IIC class, which hit a dead end in prokaryotes. In this regard, the first structure of group II intron retroelement inserting into DNA was solved by Haack et al. (2019) (ref. below) of the IIB class, which predates the group IIC intron retroelement structure solved by Chung et al. (2022).*

We thank the reviewer for appreciating the impact of this work and for the suggestion above. In our revised manuscript we compare the L1ORF2 RNP structure with that of the suggested Group IIB intron RNP and present it in Extended Data Figure 10A. We observe that a part of the intron RNA and intron RNA:DNA duplex are positioned adjacent to the intron RT thumb domain, generally in a similar position to that seen for the L1ORF2 RNP template RNA stem. We have highlighted the evolutionary relationships in the revised manuscript.

- 2. Ext. Data Fig. 6: The authors developed an assay for testing the importance of the recognition of the polyA tract for initiating cDNA synthesis. They used a DNA duplex with a polyT overhang to anneal a polyA-containing RNA template with the overhang providing the 3'-OH to prime cDNA synthesis. They show that mutating the polyA sequence into a GC-rich sequence in the RNA template does not result in cDNA synthesis. What is concerning is that the DNA overhang region is also simultaneously changed to a complementary GC-rich sequence, yet no evidence was provided to show that these regions anneal with each other. The GC-rich sequences in both the RNA template and DNA overhang could potentially form secondary structures that preclude annealing of the two regions. This is especially a concern since no high temperature annealing step was performed. This is puzzling since this is typically done for in vitro annealing of nucleic acids.*

We appreciate this comment. To address the overhang-primed RT assays concerns raised by the reviewers, we replaced the original manuscript's OPRT assays with *bona fide* TPRT assays in the revised manuscript. For TPRT assays we assemble target-site DNA by annealing complementary oligonucleotides using a protocol of heating to 95°C and slow cooling to 4°C. We refold template RNAs prior to TPRT assays by heating to 70°C and snap cooling to 4°C. For the RT assays shown in the revised manuscript (Extended Data Figure 2B-C), we annealed primer to template RNA by heating to 75°C and slow cooling to 4°C. These protocols are included in the Methods section of the revised manuscript.

3. *Many of the major conclusions/experiments regarding the L1 RT need to include a control RT such as Superscript III throughout the manuscript. It is difficult to assess the validity of the assays without a control RT. For example, it is possible that many of the properties observed for the L1 RT would also be observed with any general RT with unusual primer/template combinations. This is also true for Ext. Data Fig. 7a. In this assay, the length of the polyA tail on the RNA template was varied. Shortening this polyA tail will bring the structured GC-rich RNA hairpin closer to the RT active site and likely result in steric hindrance resulting in a lack of RT activity. However, this could happen with any general RT for cDNA synthesis. Therefore, a positive control RT would provide evidence for the assertion that this only occurs for the L1 RT.*

This is an excellent point. We have included Moloney Murine Leukemia Virus (M-MLV) RT for comparison to L1ORF2 in the RT primer-extension assays of Extended Data Figure 2B-C. We have also used M-MLV RT as a control for the TPRT assay (Extended Data Figure 2A), in which it showed no activity. We replaced the original manuscript's OPRT assays, which rely on primer 3' end pairing with a template 3' end, with TPRT assays that do not. We also expanded the analysis of template RNA variants with nucleotide substitutions rather than deletions to further resolve any concerns (revised Fig. 2C).

4. *To my knowledge, the C-terminus segment (CTS) that separates the secondary structure in the RNA template is the first description of such a domain in an RT. It is possible that a similar domain may exist in all RTs as their function generally requires them to disrupt secondary structure of the RNA template to be threaded through the active site. It may not have been seen before because no one determined the structure of an RT actively involved in cDNA synthesis in the presence of a highly structured RNA template. The presence of a homologous domain in other RTs should be verified through a bioinformatics search. It would be great if the authors could comment on this.*

We thank the reviewer for this suggestion. We bioinformatically investigated whether other RTs have a CTS-like fold. First, we searched for domains potentially related to the CTS domain using amino acid sequences. While the LINE-1 family RTs from primate species show a conserved CTS domain sequence, we failed to detect homology in eukaryotic RTs from other non-LTR retroelements and other RT families (Reviewer Figure 4). For example, we observed 16% identity/30% similarity between a region of the *Bombyx mori* (silk moth) R2 non-LTR retroelement protein and human L1ORF2 CTS, which falls below the standard criterion for homology.

We also performed a 3D structure-based search for the CTS domain fold. With this approach we found two candidates for domain homology: (1) A partial "CTS-like" fold in the R2 non-LTR retroelement protein (PDB 8GH6), interrupted by the C-terminal endonuclease domain. The putative CTS-like domain lies on a protein surface that separates the two strands of DNA duplex near the nick site (Extended Data Figure 8A); (2) A partial CTS fold in the human telomerase reverse transcriptase (TERT) RT core (PDB 7BG9) in a domain that makes contacts with single-stranded and duplex regions of telomerase RNA (Extended Data Figure 8B). These candidate "homologous" domains only partially align with the CTS. Therefore, to the best of our knowledge, L1ORF2's CTS appears to be a unique feature of the LINE-1 family, and its role as strand-splitter for RNA secondary structure appears to be the first visualization of such a role. We have included this discussion in the revised manuscript.

Minor revisions:

1. p. 5, line 137: This sentence refers to residue 468 as an arginine; however it is shown in Ext. Data Fig. 5b as “K468”, which is a lysine.

We have corrected this in the revised version.

2. p. 6, line 190: This sentence states that Ile1121 forms a hydrophobic with G1 of the RNA hairpin structure. However, Fig. 4d depicts the side chain of Ile1121 pointing away from G1. Therefore, it does not seem to be forming a hydrophobic interaction with the nucleobase of G1.

We have corrected our previous mis-labeling of the displayed bases, which we have now labeled to show Ile1121's side chain oriented towards base G-1 in the figure.

Reviewer Figure 1

Second strand nick and synthesis assay

Reviewer Figure 1: Denaturing gel analysis of in vitro TPRT reaction assaying first strand cleavage and second strand cleavage. The reaction conditions were identical to all TPRT assays in the manuscript, including identical protein and nucleic acid concentrations and buffer conditions. AJh RNA was used as the template for the + RNA reactions.

The expected on-target, second strand nick at 31nt is distinctly missing from the labeled second strand assay.

Reviewer Figure 2

T to C mutation in cleavage site

Reviewer Figure 2: Denaturing gel analysis of in vitro TPRT reaction assaying T to C mutations in the EN cleavage site. The reaction conditions were identical to all TPRT assays in the manuscript, including identical protein and nucleic acid concentrations and buffer conditions. AJh RNA was used as the template for the + RNA reactions.

Both cleavage sites are efficiently nicked by wild-type L1ORF2 but the consensus cleavage site is more readily used to synthesize the TPRT product.

Reviewer Figure 3

Single cycle processivity assay with L1ORF2
wild-type and Δ Insertion helix mutant

Reviewer Figure 3: Single cycle processivity assay was adapted from Zhao et al., RNA 2017. M-MLV RT, L1ORF2 wild-type or Δ Insertion helix mutants were incubated at 37C for 5 minutes with the pre-assembled RNA:DNA duplex at 100 nM. L1 5'UTR RNA (908nt): 32 P-labeled 30nt primer (primer: caaagaccaaagtagataaaaccacaaag) Unlabelled RNA:DNA duplex trap was added at 4 μ M with dNTPs and the reaction was incubated at 37C for 10 minutes before PCI extraction and loading on the denaturing gel. The reaction conditions used were identical to the TPRT assays in the manuscript.

While M-MLV RT dissociates from the partially elongated cDNA, wild-type and Δ Insertion helix mutant L1ORF2 generate full-length cDNAs showing no difference in processivity.

Reviewer Figure 4

Reviewer Figure 4: Multiple sequence alignment of reverse transcriptases to search for homologs of the L1ORF2 C-terminal segment.

Reviewer Reports on the First Revision:

Referees' comments:

Referee #1:

Remarks to the Author:

The authors have responded nicely to the previous critiques by developing additional assays, conducting new experiments, and determining the structure of LINE-1 ORF2p bound to a synthetic Alu RNA-like template. The authors should be commended on a very nice study. I only have some minor points for the authors to consider.

(1) LINE-1 stands for Long INterspersed Element-1 (please omit Nuclear).

(2) The authors may wish to delete ref. 28, as it deals with another SINE (SVA), which is not discussed in the paper.

(3) The authors may wish to qualify the statement on line 76; it is unclear how they determined that the products from the L1 3' UTR containing RNA are predominantly truncated; did they sequence the products (perhaps I missed this point)?

(4) The authors may wish to cite PMID: 9671814 when discussing the Z-domain on line 113.

Referee #2:

Remarks to the Author:

The authors have extensively revised their manuscript, with several interesting new observations and a refocused narrative. Overall, the revised version provides new possible insights into L1 biology. The most important are: (i) the structure of L1 ORF2p with an Alu-like RNA template (a biologically relevant L1 substrate), and the use of this template in subsequent experiments; (ii) the development of a target-primed reverse transcription (TPRT) assay *in vitro* using purified components; and (iii) the identification of a preferred DNA target with characteristics similar to replicating DNA. If confirmed, these results are exciting and would significantly advance our understanding of L1 replication. However, the new experiments raised several issues and some of the answers to previous comments are not fully satisfactory.

Major points:

1. The use of an Alu-like RNA is an attempt to mimic a native template for L1 ORF2p. Nevertheless, RNA of active Alu elements is not free but bound by SRP9/14 (Benett et al., *Genome Res.*, 2018), which influences its folding, and probably the interaction surface between Alu and ORF2p. Therefore, under the experimental conditions of this work, it is unclear whether this Alu RNA behaves like any random RNA with the benefit of providing monodisperse particles, or whether it reflects the bona fide interaction of Alu RNA with ORF2p. Structures of Alu RNA in complex with SRP9/14 have already been published and contacts between Alu RNA and SRP9/14 are known (Ahl et al., *Mol. Cell*, 2015). Minimally, the authors could check whether the obtained structure is topologically compatible with SRP9/14 binding and how the Alu structure in the context of ORF2p differs from other forms. I understand that the Alu-ORF2p structure is limited in resolution, but it should be sufficient for this purpose. As presented, the Alu/ORF2p structure is not exploited, except to show that it resembles to some extent the structure obtained L1 with the synthetic template.

2. The new experiment shown in Fig. 2c to test the specific recognition of the polyA tract by ORF2p is not fully conclusive. Indeed, after the initial cleavage, the length of the T-tract that will act as a primer is 14 nt with a CC mismatch in the middle. Thus, neither the 25N nor the 20N5A templates truly reproduce this setting. For this specific question, a simple RT assay would be beneficial to effectively test templates ending with homopolymers other than (A)_n and randomized sequences and matching primers.

3. In the same line, the TPRT activity profile of the Δ ss mutant has led the authors to conclude that the mutated residues “appear to contribute to stable engagement of the single-stranded polyA tract with L1ORF2 accessory domains”. However, this mutant appears to exhibit a clear reduction in nicking activity, which may subsequently affect the second step of the reaction (extension). This issue is also connected to points 3 and 4.

4. An important aspect of the revised manuscript is the development and optimization of an in vitro TPRT assay. TPRT is a concerted process whereby the nicked DNA is transferred to the RT active site from the same ORF2p protein to primer reverse transcription. However, it remains to be demonstrated whether the extension products detected in the assay are bona fide TPRT products or reflect EN cleavage, dissociation of the short radiolabeled primer from the duplex due to the very short annealed region, and subsequent extension by another ORF2p molecule of the released primer. The second scenario would be compatible with some of the results shown in Fig. 4b,c, and Ext. Data Fig. 9: the 5-bp and 26-bp substrates show similar levels of nicked substrate at the TTTT/A site (red arrows) but only the 5-bp nicked DNA is extended. The T_m of the 5-bp nicked DNA is very low in contrast to the 26-bp version. Therefore, the 5-bp nicked DNA could dissociate from the rest of the duplex and be extended (OPRT), while the 3' OH of the 26-bp nicked DNA would remain embedded in the duplex and be inaccessible. In other words, the TPRT assays shown in the manuscript could simply be endonuclease assays with a readout by reverse transcriptase-mediated extension. Unambiguously demonstrating that the TPRT assay actually measures the efficiency of a TPRT reaction is essential to support many of the claims made in the revised version.

5. Related to the previous point, the strategy of quantification for the TPRT reaction is unclear. I understand from the method section that only the full-length products are quantified. Why exclude the higher products? Moreover, also quantifying individually each of the two steps of the reaction would help decipher which one is affected by the variable element in the reaction (e.g. step 1 = cleaved primer at the EN site + RT products; step 2 = RT products/step 1).

6. The notion that ORF2p may preferentially nick DNA substrates close to single-strand/double-strand transitions is very interesting. The authors postulated that this may be due to a steric clash that may arise with long dsDNA targets. If their hypothesis is true, the Δ CTS version of ORF2p used in some experiments should nick all substrates equally, as would the EN domain alone. Testing this would significantly reinforce the structure-guided model proposed to explain the substrate specificity of ORF2p.

7. As noted by the authors, the average TSD length deduced from the specificity of the ORF2p substrate is significantly shorter than what is observed for in vivo insertions. It is proposed that PCNA could 'fine-tune' the selection of the target site, but the proposed mechanism is unclear. The effect of PCNA was tested in vitro only for increased activity and not for allowing usage of substrates with more internal cleavage site. A caveat may be added to underline that targeting the replication fork may represent only one mechanism of L1 integration (among other(s) which could generate longer TSDs).

Minor points:

8. Accepted nomenclature is 'LINE-1 ORF2p' or 'L1 ORF2p' (with a space or dash) for the protein encoded by ORF2. I would suggest sticking to this convention throughout the manuscript and in the figures.

9. Referencing could be improved:

a. For the presence of a G-quadruplex in L1 3' UTR, please cite Sahakyan et al. (NSMB, 2017); the authors may want to discuss the potential discrepancy as this previous study suggested that the G-quadruplex promotes retrotransposition.

b. L. 114, ref. 40: I believe the NTE domain was rather recognized in various non-LTR retroelements in Malik HS et al. (Mol. Biol. Evol., 1999).

c. The RNA-binding properties of the CTS were identified many years ago in Piskareva et al. (FEBS Openbio., 2013), which should be given credit.

d. L. 288 and 303: L1 EN nicking the lagging-strand template during replication was proposed in both Flash et al. (Cell, 2019) and Sultana et al. (Mol. Cell, 2019). Both studies have also hypothesized that second-strand synthesis could involve the host DNA replication machinery without the need for a second L1-mediated endonuclease cleavage. The preferred substrate of ORF2p identified in this manuscript is consistent with this earlier hypothesis.

10. TPRT is often misspelled as TRPT throughout the manuscript.

11. Lack of a document with track changes made it difficult to assess the differences between the two versions of the manuscript.

Author Rebuttals to First Revision:

Response to Reviewers' Comments

Below we describe how we addressed the new comments made by the reviewers. We made changes in the manuscript text, quantified reaction products in additional figure panels, and added Extended Data Figure 3c and Extended Data Figure 10. We also provide supporting data in two Reviewer Figures accompanying this document.

We thank you for your help in reviewing this paper.

Reviewer 1

The authors have responded nicely to the previous critiques by developing additional assays, conducting new experiments, and determining the structure of LINE-1 ORF2p bound to a synthetic Alu RNA-like template. The authors should be commended on a very nice study. I only have some minor points for the authors to consider.

We appreciate the reviewer's recognition of the impact of our work.

- *LINE-1 stands for Long Interspersed Element-1 (please omit Nuclear).*

We have revised this as requested by the reviewer.

- *The authors may wish to delete ref. 28, as it deals with another SINE (SVA), which is not discussed in the paper.*

We have omitted this citation as suggested and instead cited previous reference Deininger et al., Genome Biol 2011 here (a review on Alu SINEs).

- *The authors may wish to qualify the statement on line 76; it is unclear how they determined that the products from the L1 3' UTR containing RNA are predominantly truncated; did they*

sequence the products (perhaps I missed this point)?

We reworded our statement to specify that the L1 3'UTR products migrated faster than expected (i.e. were shorter than expected for full-length cDNA). We also made the context for the statement clear. The text now reads: "TPRT of the L1 3'UTR RNA resulted in a lower amount of product synthesis, with products predominantly migrating faster than expected for full-length cDNA (Fig. 1C, lane L1). L1 3'UTR template lacking nt 1-78 that form G-quadruplex gave the expected cDNA length, which matched the length of the shorter products from the full-length L1 3'UTR template (Fig. 1C, lanes L1 and L1Δ)."

- *The authors may wish to cite PMID: 9671814 when discussing the Z-domain on line 113.*

This work is now cited in addition to the more recent publication.

The authors have extensively revised their manuscript, with several interesting new observations and a refocused narrative. Overall, the revised version provides new possible insights into L1 biology. The most important are: (i) the structure of L1 ORF2p with an Alu-like RNA template (a biologically relevant L1 substrate), and the use of this template in subsequent experiments; (ii) the development of a target-primed reverse transcription (TPRT) assay in vitro using purified components; and (iii) the identification of a preferred DNA target with characteristics similar to replicating DNA. If confirmed, these results are exciting and would significantly advance our understanding of L1 replication. However, the new experiments raised several issues and some of the answers to previous comments are not fully satisfactory.

We have performed new experiments, presented as new Reviewer Figures accompanying this document and as new Extended Data Figure 10, to address these questions and reinforce the novelty and significance of our work.

Major points:

5. The use of an Alu-like RNA is an attempt to mimic a native template for L1 ORF2p. Nevertheless, RNA of active Alu elements is not free but bound by SRP9/14 (Benett et al., Genome Res., 2018), which influences its folding, and probably the interaction surface between Alu and ORF2p. Therefore, under the experimental conditions of this work, it is unclear whether this Alu RNA behaves like any random RNA with the benefit of providing monodisperse particles, or whether it reflects the bona fide interaction of Alu RNA with ORF2p. Structures of Alu RNA in complex with SRP9/14 have already been published and contacts between Alu RNA and SRP9/14 are known (Ahl et al., Mol. Cell, 2015). Minimally, the authors could check whether the obtained structure is topologically compatible with SRP9/14 binding and how the Alu structure in the context of ORF2p differs from other forms. I understand that the Alu-ORF2p structure is limited in resolution, but it should be sufficient for this purpose. As presented, the Alu/ORF2p structure is not exploited, except to show that it resembles to some extent the structure obtained with the synthetic template.

The SRP9/14 proteins bind a different region at the 5' end of Alu RNA, which is distinct from the region L1ORF2p binds. We have added a schematic of these two distinct binding sites on the Alu RNA to Extended Data Figure 3A. Because the 5' region of the Alu RNA for SRP9/14 interaction does not bind to L1ORF2p, this region remains flexible and cannot be visualized in our L1ORF2p cryo-EM map. To further address the reviewer's comment, we have superimposed the Alu-SRP structure (Ahl et al., 2015) with our structure using the common Alu stem seen in both structures (Extended Data Figure 3c). We find that there is no topological incompatibility between SRP9/14 binding and L1ORF2p binding. We have now stated this finding in the text.

The new L1ORF2p-Alu RNP structure reinforces our conclusion, originally from our L1ORF2p structure with a synthetic RNA template, about L1ORF2p binding to a hairpin stem, as was requested by the reviewer in the first round of feedback. The new structure prompted us to use Alu RNA as L1ORF2p's template in the biochemical assays of the revised manuscript. Beyond this use, our L1ORF2p-Alu RNP structure is lower resolution and has identical features that can be discerned at its limited resolution as compared to our high resolution L1ORF2p-synthetic RNP structure. Therefore, we primarily used the highest resolution structure to examine the detailed protein-RNA interactions.

6. *The new experiment shown in Fig. 2c to test the specific recognition of the polyA tract by ORF2p is not fully conclusive. Indeed, after the initial cleavage, the length of the T-tract that will act as a primer is 14 nt with a CC mismatch in the middle. Thus, neither the 25N nor the 20N5A templates truly reproduce this setting. For this specific question, a simple RT assay would be beneficial to effectively test templates ending with homopolymers other than (A)_n and randomized sequences and matching primers.*

We emphasize that the length of the T-tract base paired to the RNA is not 14 nt, but instead is 5 nt resulting from the cleavage at the 5' TTTTT/AA 3' target site. We can confidently say this from several experiments. First, the long TTTTTTCCTTTTT tract would not anneal with 14 adenosines in template RNA to generate primer-template duplex. Further, the L1ORF2p core can only accommodate and stabilize a maximum of only 6 RNA-DNA bp as seen in our structure, not 14 bp. Second, the proposition that more than 5 bp form between cleavage-generated primer and the template polyA ignores our data, which clearly shows that L1ORF2p can use target sites without any T-rich 5' overhang sequence: the target site with sequence CGCGCCGAA in the 5' overhang followed by the target DNA duplex CCTTTTT/AA is productive for TPRT (Figure 4d). We designed the template RNAs with 3' single-stranded 25N and 20NA5 for use with nicked target-site primers ending CCTTTTT-3'. To reinforce our conclusion that these RNAs are not productive templates for TPRT, we performed an additional experiment using the AluJ half RNAs ending in 25A, 25N or 20NA5 in combination with the target site DNA having CGCGCCGAA in the 5' overhang (see Reviewer Figure 1). The 25N and 20NA5 template RNAs do not support TPRT, matching our findings from Figure 2c. Finally, we do not agree that RT assays would be the most appropriate for studying polyA tail engagement during L1ORF2p TPRT. Instead, our TPRT assays provide more definitive answers.

7. *In the same line, the TPRT activity profile of the Δ ss mutant has led the authors to conclude that the mutated residues “appear to contribute to stable engagement of the single-stranded polyA tract with L1ORF2 accessory domains”. However, this mutant appears to exhibit a clear reduction in nicking activity, which may subsequently affect the second step of the reaction (extension). This issue is also connected to points 3 and 4.*

We added quantification comparing the amounts of target site nicking and TPRT by wild-type L1ORF2p and Δ ss mutant (Figure 2g). The Δ ss mutant's nicking activity is reduced by 50% compared to the wild-type protein, but its TPRT activity is drastically reduced to 8% of that of wild-type L1ORF2p TPRT, supporting our prior conclusion. The revised text now states the following to clarify this point: “When assayed for TPRT activity, the L1ORF2p mutant for single-stranded RNA base interactions (Δ ss) showed distinctly reduced TPRT while retaining significant EN activity and RT activity when assayed by primer extension (Fig. 2g and Extended Data Fig. 2b-c). We suggest that these contacts contribute to a conformation of L1ORF2p poised for cDNA synthesis.”

8. *An important aspect of the revised manuscript is the development and optimization of an in vitro TPRT assay. TPRT is a concerted process whereby the nicked DNA is transferred to the RT active site from the same ORF2p protein to primer reverse transcription. However, it remains to be demonstrated whether the extension products detected in the assay are bona fide TPRT products or reflect EN cleavage, dissociation of the short radiolabeled primer from the duplex due to the very short annealed region, and subsequent extension by another ORF2p molecule of the released primer. The second scenario would be compatible with some of the results shown in Fig. 4b,c, and Ext. Data Fig. 9: the 5-bp and 26-bp substrates show similar levels of nicked substrate at the TTTTT/A site (red*

arrows) but only the 5-bp nicked DNA is extended. The T_m of the 5-bp nicked DNA is very low in contrast to the 26-bp version. Therefore, the 5-bp nicked DNA could dissociate from the rest of the duplex and be extended (OPRT), while the 3' OH of the 26-bp nicked DNA would remain embedded in the duplex and be inaccessible. In other words, the TPRT assays shown in the manuscript could simply be endonuclease assays with a readout by reverse transcriptase-mediated extension. Unambiguously demonstrating that the TPRT assay actually measures the efficiency of a TPRT reaction is essential to support many of the claims made in the revised version.

We want to first emphasize that the template RNA and target DNA specificities for L1ORF2p we have uncovered here stand on their own, independent of L1ORF2p's TPRT coordination, or lack of coordination between DNA cleavage and RT activities. However, we are confident that L1ORF2p DNA cleavage and RT activities are coordinated. In Reviewer Figure 2, we show a comparison of DNA cleavage and TPRT activities of wild-type L1ORF2p versus an equimolar mixture of EN-dead L1ORF2p and RT-dead L1ORF2p. While the amount of cleaved DNA is similar between the wild-type protein and the mixture of EN-dead and RT-dead proteins, very little cDNA synthesis occurred in the latter reaction. This would not be the case if DNA primer dissociated from EN-dead protein. Instead, this result indicates that the vast majority of TPRT product is generated by coordinating DNA cleavage and cDNA synthesis. Finally, please note that the 5-bp and 26-bp substrates do not show similar levels of nicked substrate at the target TTTT/A site. Please compare the cleavage product indicated by red arrow in Figure 4b or Extended Data Figure 9a to observe that there is drastically lower level of cleavage of the 26-bp target site.

However, we agree with the reviewer that cleavage internally in duplex DNA may not support efficient TPRT, because L1ORF2p is not able to transfer the primer 3' end from the EN active site to the RT active site. Duplex melting must occur for the primer 3' end to base-pair with template, and L1ORF2p may not contain a domain that can perform this role. Furthermore, because we show a highly stimulatory role for L1ORF2p recognition of a 5' single-stranded overhang from the duplex target site, a longer upstream duplex could compromise an L1ORF2p interaction with single-stranded DNA that helps transfer nicked primer to the RT active site. We discuss this in the manuscript by stating, "The L1ORF2p target-site architecture, where first-strand cleavage occurs at a limited length of duplex away from a single-strand/duplex transition on the 5'-overhang strand, produces a nick only ~10 bp away from the 5' overhang. This ~10 bp of duplex is prone to dissociation, eliminating the need for second-strand nicking (Fig. 4f)."

Finally, visualizing the coordination of EN cleavage and RT activities by L1ORF2p would likely involve single-molecule biophysical assays. While such extensive experiments are beyond the scope of our current work, we agree that it will be of interest to pursue them in future studies.

9. Related to the previous point, the strategy of quantification for the TPRT reaction is unclear. I understand from the method section that only the full-length products are quantified. Why exclude the higher products? Moreover, also quantifying individually each of the two steps of the reaction would help decipher which one is affected by the variable element in the reaction (e.g. step 1 = cleaved primer at the EN site + RT products; step 2 = RT products/step 1).

We would like to clarify that for most RNAs, larger than full-length cDNA product levels are much less abundant, with low signal above background. Also, some longer products run close to the gel wells where protein-nucleic acid aggregates also migrate. Because the longer than full-length cDNA products result from template-jumping, their amount is proportional to the full-length cDNA product in each lane, hence their exclusion does not influence our relative quantification.

For most Figure panels, the amount of nicked DNA product is either similar across lanes (e.g. comparing use of different template RNAs) or cannot be quantified because there is not a distinct nicked product band at the correct size (when comparing across target sites). For specific panels that the reviewer requested in points 3 and 4, we added quantification (Figs. 2g, 3d). For these two panels, comparing the relative amount of nicked product and relative amount of TPRT product provides meaningful information and also reveals any change in efficiency of nicked primer elongation. We provided the quantification in that manner.

10. *The notion that ORF2p may preferentially nick DNA substrates close to single-strand/double-strand transitions is very interesting. The authors postulated that this may be due to a steric clash that may arise with long dsDNA targets. If their hypothesis is true, the Δ CTS version of ORF2p used in some experiments should nick all substrates equally, as would the EN domain alone. Testing this would significantly reinforce the structure-guided model proposed to explain the substrate specificity of ORF2p.*

We have tested the DNA cleavage by the Δ CTS mutant of L1ORF2p using target sites with different duplex lengths upstream of the nick site and have added those results to the manuscript as new Extended Data Figure 10. We observed a strikingly different pattern of cleavage with the Δ CTS mutant, as predicted by our model. DNA substrates with 13-bp, 18-bp and 26-bp duplex upstream of the target site are not nicked at the target site by wild-type L1ORF2p, but Δ CTS L1ORF2p does nick the target site in these duplexes and does so selectively to produce a distinct cleavage product at the expected length (Extended Data Fig. 10). We expanded Fig. 4c to add a 13-bp duplex upstream of the target site for direct comparison with Extended Data Fig. 10 and to reinforce that wild-type L1ORF2p cannot efficiently cleave when >10-bp are present upstream of the cleavage site. Notably, because the level of EN nicking by the Δ CTS mutant is not equal for all the target sites we tested, we suggest that other features of target DNA might influence cleavage activity. In the main text, we state, “Consistent with what would be expected from the structure, deletion of L1ORF2p’s CTS domain (Δ CTS mutant) enabled nicking of DNA substrates with an upstream duplex region greater than 13 bp (Extended Data Fig. 10). Nonetheless, the Δ CTS mutant did not nick all target sites equally (Extended Data Fig. 10), indicating that there are other determinants of efficient nicking beyond the minimal consensus TTTT/AA.” In summary, the wild-type L1ORF2p and the Δ CTS mutant demonstrate distinct EN cleavage patterns, consistent with our model.

11. *As noted by the authors, the average TSD length deduced from the specificity of the ORF2p substrate is significantly shorter than what is observed for in vivo insertions. It is proposed that PCNA could 'fine-tune' the selection of the target site, but the proposed mechanism is unclear. The effect of PCNA was tested in vitro only for increased activity and not for allowing usage of substrates with more internal cleavage site. A caveat may be added to underline that targeting the replication fork may represent only one mechanism of L1 integration (among other(s) which could generate longer TSDs).*

We have revised this sentence to now state, “The target-site DNA architecture also accounts for sequence duplication surrounding the new L1 insertion, although observed target-site duplication lengths^{8,29} would also depend on other factors, e.g. the extent of unpairing of upstream duplex by incursion of Replication Protein A from the adjacent single-stranded DNA”.

Minor points:

12. *Accepted nomenclature is 'LINE-1 ORF2p' or 'L1 ORF2p' (with a space or dash) for the protein encoded by ORF2. I would suggest sticking to this convention throughout the manuscript and in the figures.*

We have changed this in the manuscript and figures as suggested.

13. *Referencing could be improved:*

- *For the presence of a G-quadruplex in L1 3' UTR, please cite Sahakyan et al. (NSMB, 2017); the authors may want to discuss the potential discrepancy as this previous study suggested that the G- quadruplex promotes retrotransposition.*

We have cited this reference.

- *L. 114, ref. 40: I believe the NTE domain was rather recognized in various non-LTR retroelements in Malik HS et al. (Mol. Biol. Evol., 1999).*

This reference is cited on line 114.

- *The RNA-binding properties of the CTS were identified many years ago in Piskareva et al. (FEBS Openbio., 2013), which should be given credit.*

We have added this reference when discussing the properties of the CTS.

- *L. 288 and 303: L1 EN nicking the lagging-strand template during replication was proposed in both Flash et al. (Cell, 2019) and Sultana et al. (Mol. Cell, 2019). Both studies have also hypothesized that second-strand synthesis could involve the host DNA replication machinery without the need for a second L1-mediated endonuclease cleavage. The preferred substrate of ORF2p identified in this manuscript is consistent with this earlier hypothesis.*

We want to clarify that our work introduces the completely unique idea that no machinery may be needed for second strand cleavage - not ORF2p, nor host machinery - which is what we discuss. Our work does not address which polymerase does second-strand synthesis. For nicking the lagging strand template, Flasch et al. suggested it occurs as a preference, whereas Sultana et al. concluded that there is no preference for that strand when frequency of the target-site motif is considered. Therefore, we have not included this reference in that specific discussion sentence, but Sultana et al (Mol Cell, 2019) is cited in our manuscript in a different place.

14. *TPRT is often misspelled as TRPT throughout the manuscript.*

The three instances of this error were corrected.

15. *Lack of a document with track changes made it difficult to assess the differences between the two versions of the manuscript.*

Our previous revision resulted in extensive manuscript re-writing and tracked changes at that stage was not informative for us or the reviewers. We are providing this second revision with all changes tracked.

Reviewer Figure 1

TPRT with polyA mutant RNAs with non-T overhang target DNA

Target DNA: **CGCGCCGAA**CCCTTTTAAAGGACGCATGATGCGGAAACAATGCATCAC
GGAAAAATTCCTGCGTACTACGCCTTTGTTACGTAGTG

Template RNA
3' tail: - 25A 25N 20N5A

L1ORF2p: - + + +

Reviewer Figure 1: Denaturing gel analysis of in vitro TPRT reaction with target DNA containing non-T overhang and polyA mutant RNAs. The reaction conditions were identical to all TPRT assays in the manuscript, including identical protein and nucleic acid concentrations and buffer conditions. AJh RNA ending in 25A, 25N, or 20N5A were used as the template for the + RNA reactions as marked.

In agreement with Figure 2C's findings, the AJh 25A, but not AJh 25N and AJh 20N5A result in TPRT. Loss of TPRT with the AJh 20N5A RNA reinforces that polyA recognition by the L1ORF2p is important for TPRT.

Reviewer Reports on the Second Revision:

Referees' comments:

Referee #2:

Remarks to the Author:

The authors have addressed all my previous concerns. Congratulations for this remarkable piece of work!

I noted below a few minor points to the discretion of the authors:

1. In the new Ext. Data Fig. 10, as no internal WT ORF2p control is included, quantifying cleavage activity could help the reader to compare with results shown in Fig. 4 for the WT enzyme.
2. The reviewer Fig. 2 convincingly shows that EN and RT activities are indeed coordinated. Why not include this figure in the manuscript as extended data?
3. The authors still use "L1ORF2p" instead of "L1 ORF2p" throughout the manuscript (my previous point #8).

Author Rebuttals to Second Revision:

The authors have addressed all my previous concerns. Congratulations for this remarkable piece of work! I noted below a few minor points to the discretion of the authors:

1. In the new Ext. Data Fig. 10, as no internal WT ORF2p control is included, quantifying cleavage activity could help the reader to compare with results shown in Fig. 4 for the WT enzyme.

Extended Data Fig. 10 is now Extended Data Fig. 8. The lowest on-target cleavage activities do not generate a product resolved from the background of off-target cleavage, so we would not consider quantification reliable. Also, for compliance with Nature editorial policies, we refrained from comparing cleavage product intensities between set of reactions, since the reaction products were resolved on more than one gel.

2. The reviewer Fig. 2 convincingly shows that EN and RT activities are indeed coordinated. Why not include this figure in the manuscript as extended data?

We appreciate this idea behind this request, but we will refrain from including the data for the RT- dead and EN-dead mixing experiment because we have so far tested only one target site. [REDACTED]

3. The authors still use "L1ORF2p" instead of "L1 ORF2p" throughout the manuscript (my previous point #8).

We have modified L1ORF2p to L1 ORF2p through the manuscript including the figure files.